# ETV4 is a mechanical transducer linking cell crowding dynamics to lineage specification

Seungbok Yang[1,4], Mahdi Golkaram[2,4], Seyoun Oh[1], Yujeong Oh [1], Yoonjae Cho[1], Jeehyun Yoe[1], Sungeun Ju[1], Matthew A. Lalli[3], Seung-Yeol Park [1], Yoontae Lee [1] & Jiwon Jang [1]✉

Dynamic changes in mechanical microenvironments, such as cell crowding, regulate lineage fates as well as cell proliferation. Although regulatory mechanisms for contact inhibition of proliferation have been extensively studied, it remains unclear how cell crowding induces lineage specification. Here we found that a well-known oncogene, ETS variant transcription factor 4 (ETV4), serves as a molecular transducer that links mechanical microenvironments and gene expression. In a growing epithelium of human embryonic stem cells, cell crowding dynamics is translated into ETV4 expression, serving as a pre-pattern for future lineage fates. A switch-like ETV4 inactivation by cell crowding derepresses the potential for neuroectoderm differentiation in human embryonic stem cell epithelia. Mechanistically, cell crowding inactivates the integrin–actomyosin pathway and blocks the endocytosis of fibroblast growth factor receptors (FGFRs). The disrupted FGFR endocytosis induces a marked decrease in ETV4 protein stability through ERK inactivation. Mathematical modelling demonstrates that the dynamics of cell density in a growing human embryonic stem cell epithelium precisely determines the spatiotemporal ETV4 expression pattern and, consequently, the timing and geometry of lineage development. Our findings suggest that cell crowding dynamics in a stem cell epithelium drives spatiotemporal lineage specification using ETV4 as a key mechanical transducer.

Gastrulation is an early developmental event to derive the three embryonic germ layers. In amniotes, the pre-gastrulation embryo goes through an evolutionarily conserved morphogenetic process where it forms a single-layered epithelial sheet known as the epiblast[1]. The evolutionary conservation of the single-layered epithelium suggests that this morphological structure is the prerequisite for forming the three embryonic germ layers[1]. This epithelium stably persists for about one week in primate post-implantation embryos, during which it undergoes 10- to 20-fold size expansion and acquires mature differentiation potential[2,3]. However, the direct contribution of epithelial expansion to lineage determination remains uncertain.

Self-organization is a cellular process that spontaneously creates complex structures with no particular pre-pattern[4,5]. In human embryonic stem cell (hESC)-derived gastruloid models based on micropatterning technology, a differentiating hESC colony forms an ordered structure of germ layers along the radial axis with ectoderm in the centre and mesendoderm in the periphery[6,7]. In past decades, a significant emphasis has been placed on diffusible signalling factors to explain self-organization[8]. Accordingly, the gradients of BMP4 and Activin–Nodal signalling were suggested to drive the spatial derivation of multiple germ layers within an hESC colony[6,7]. Nonetheless, considering the evolutionary conservation of epithelial structures in

[1]Department of Life Sciences, Pohang University of Science and Technology, Pohang, Republic of Korea. [2]Department of Mechanical Engineering, University of California, Santa Barbara, Santa Barbara, CA, USA. [3]Seaver Autism Center for Research and Treatment at Mount Sinai, New York, NY, USA. [4]These authors contributed equally: Seungbok Yang, Mahdi Golkaram. ✉e-mail: jiwonjang@postech.ac.kr

pre-gastrulating epiblasts, we hypothesized that unique morphological features related to the epithelium are involved in lineage specification.

In a single-layered epithelium, active proliferation in a confined environment induces local cell crowding that inhibits epithelial cell proliferation and induces cell extrusion[9]. YAP/TAZ and PIEZO1 were identified as crucial mediators linking cell crowding to cellular responses[10–15]. Furthermore, growing evidence suggests that cell crowding plays an essential role in cellular differentiation in the epidermis and developing zebrafish hearts[16,17]. Together with the critical roles of YAP/TAZ and PIEZO1 in stem cell differentiation[18–23], these findings suggest local cell crowding as a key cellular mechanism through which a new cell fate is generated within a seemingly homogeneous stem cell population. Nonetheless, given the diversity of mechanical stimuli encountered by stem cells during embryonic development, it remains crucial to identify additional factors that transduce mechanical signals to gene expression.

In this work, we identified an ultrasensitive mechanical transducer, ETV4, and its upstream mechanotransduction pathway in hESCs. This discovery unveils the mechanism by which epithelial crowding regulates spatiotemporal lineage derivation.

## Results

### Regulation of ETV4 expression by mechanical cues

To identify transcriptional regulators that link cell crowding to transcriptional responses in hESCs, we manipulated local cell crowding by limiting the cell adhesive area and performed RNA sequencing (RNA-seq) (Fig. 1a, Extended Data Fig. 1b, and Supplementary Table 1). Cell crowding was verified by measuring the average area of cells covering the substrate (hereafter referred to as cell area; Extended Data Fig. 1a). Gene ontology (GO) analysis with total differentially expressed genes (DEGs) ($|log_2(fold change)| > 0.5$, adjP < 0.05) revealed the top 10 GO terms related to embryonic development and differentiation (Extended Data Fig. 1c)[24], supporting the relevance of cell crowding in developmental processes. Contractile actomyosin bundles represent a reliable indicator of cellular mechanical stress levels. Gene set enrichment analysis (GSEA) revealed a significant reduction in genes related to actomyosin in crowded hESCs (Extended Data Fig. 1d)[25], confirming decreased mechanical stress. These results were validated by immunostaining for phosphorylated myosin light chain (pMLC) that marks contractile actomyosin bundles (Extended Data Fig. 1e).

To infer transcriptional regulators, we applied a recently developed analytic tool called Lisa (Landscape In Silico Deletion Analysis) that uses chromatic profile data[26]. Lisa analysis with top 200 downregulated DEGs in crowded hESCs (ordered by fold change) revealed 40 transcriptional regulators with significant *P* values (*P* < 0.01) and high expression in hESCs (fragments per kilobase of transcript per million mapped reads (FPKM) > 10; Fig. 1b and Supplementary Table 2). TEAD2 and SRF, previously known to be related to mechanotransduction, are included in the list. Protein function annotation with the selected transcription factors (TFs) using InterPro revealed seven protein domains with significant *P* values (adjP < 0.05). Among those identified, we found two terms related to PEA3-type ETS-domain TFs (Fig. 1b).

The PEA3 family consists of ETV1 (also known as ER81), ETV4 (also known as PEA3) and ETV5 (also known as ERM)[27]. Immunostaining validated the high expression of all PEA3 family TFs in hESCs and cell-crowding-induced reduction in protein expression (Fig. 1c–e). Among the PEA3 family TFs, we focused on ETV4 for its highest expression level in hESCs (Extended Data Fig. 1f). To test if cell-crowding-induced ETV4 downregulation is a reversible phenotype, we used a scratch assay, where rapid reactivation of ETV4 expression was observed in the cells adjacent to the scratches along with increased cell area (Fig. 1f). The cell-density-mediated regulation of ETV4 expression was also confirmed in other epithelial cell lines (Fig. 1g and Extended Data Fig. 1g). Direct manipulation of cell area by a cell stretching system revealed that a 15% decrease in cell area was sufficient

to diminish the nuclear expression of ETV4 (Fig. 1h,i and Extended Data Fig. 1h–j).

Substrate stiffness is another physiologically relevant stimulus that regulates cellular mechanical stress[28]. Reduced mechanical stress by soft substrates was confirmed by immunostaining for pMLC and YAP/TAZ (Extended Data Fig. 1k,l). Compared with plastic (-1 Gpa), the nuclear expression of ETV4 proteins was significantly reduced when cells were placed on soft substrates (Fig. 1j,k). Replating hESCs to plastic resumed with high ETV4 expression (Fig. 1l). Overall, we have identified ETV4 as a TF whose expression is regulated by various mechanical cues.

### Regulation of ETV4 expression by cell crowding dynamics

Like the in vivo epiblast, hESCs intrinsically grow, forming a single-layered epithelial colony in a culture dish (Fig. 2a). As hESC colonies expanded, we observed dynamic spatiotemporal changes in cell density. In small colonies (<0.5 mm²), individual cells exhibited a relatively large cell area, indicating a lesser degree of crowding (Fig. 2b). Conversely, in larger colonies (>2 mm²), cells located in the centre were smaller, while those at the periphery retained a larger cell area (Fig. 2b and Extended Data Fig. 2a). A gradual decrease in cell area was observed from the periphery to the centre of large hESC colonies (Fig. 2c). Because cell area exhibited a strong correlation with nucleus size (Extended Data Fig. 2b), we conducted real-time measurements of single nucleus sizes in an hESC line expressing a nuclear reporter, H2B-GFP. As the hESC colony expanded, an increase in single-cell variation in nucleus size was observed, with cells in the centre becoming smaller (Fig. 2d). These results collectively underscore the dynamic regulation of local cell crowding during epithelial expansion.

Given the mechanosensitive regulation of ETV4 expression, we investigated the ETV4 expression patterns in H9 and H1 hESC colonies. ETV4 was homogeneously expressed in the nucleus of cells from small hESC colonies (<0.5 mm²; Fig. 2e and Extended Data Fig. 2c). Strikingly, large hESC colonies (>2 mm²) showed zonation of ETV4 expression with a reduction in the crowded centre (Fig. 2e and Extended Data Fig. 2c). However, the expression of core pluripotency genes (*OCT4*, *NANOG* and *SOX2*) remained high across the whole colony (Fig. 2e and Extended Data Fig. 2c,d). These results were confirmed in hESCs cultured either in a different medium or on a different coating extracellular matrix (ECM; Extended Data Fig. 2e,f). Clonal colonies derived from single hESCs also showed the zonation of ETV4 expression (Extended Data Fig. 2g). Consistent with the zonation pattern, sharp reduction in ETV4 expression was observed with increased cell density (Fig. 2f). The zonation pattern of ETV4 predominantly emerged when the colony diameter exceeded 1,000 μm, and the size of the ETV4-low area increased proportionally to the whole colony size (Fig. 2g). Reversible ETV4 activation in the centre of large colonies was confirmed by a scratch assay (Fig. 2h).

Because cell density influences various cellular processes, we first measured the cell shape index (CSI) to evaluate cell circularity at the boundary where ETV4 expression sharply transitions. High ETV4-expressing cells exhibited significantly larger cell areas than low ETV4-expressing cells, while both types of cells had similar CSIs (Extended Data Fig. 3a). These results suggest that ETV4 expression is associated with cell area rather than cell geometry. Next, we studied the potential effect of cell density on cellular metabolism. GSEA, based on the RNA-seq data from crowded hESCs, revealed no significant difference in the expression of glycolysis- and hypoxia-related genes (Extended Data Fig. 3b). To directly measure hypoxia, we used the hypoxyprobe system, in which pimonidazole hydrochloride forms protein adducts in hypoxic cells (Extended Data Fig. 3c)[29]. Consistent with the GSEA results, there was no evident induction of hypoxia in the crowded centre of large hESC colonies (Extended Data Fig. 3c). Furthermore, glucose uptake measurements demonstrated similar uptake rates between the periphery and the centre of large hESC colonies (Extended Data Fig. 3d,e). Although we cannot completely

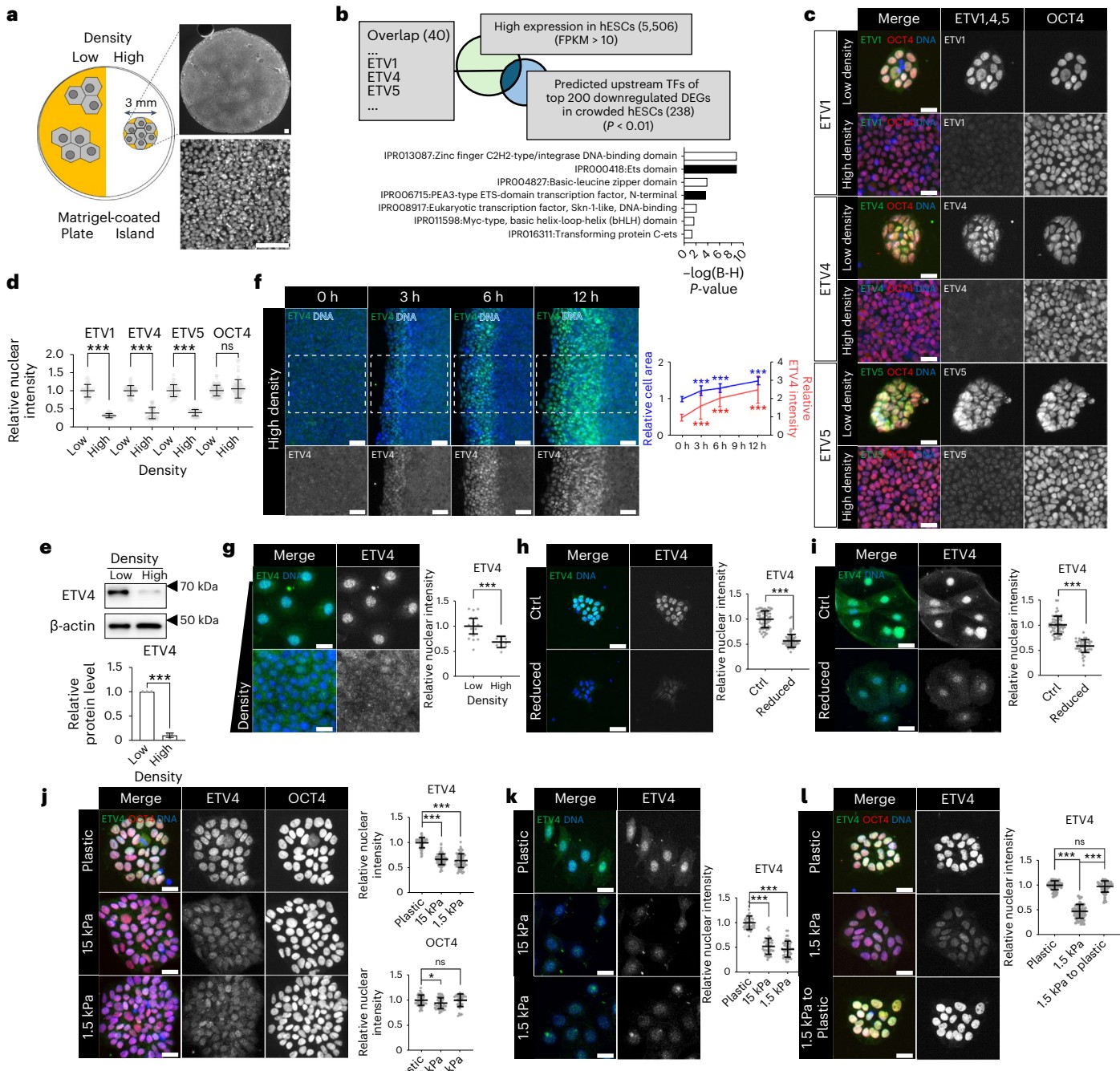

**Fig. 1 | Mechanical microenvironments regulate ETV4 expression. a**, Schematic representation of high-density culture. Cell adhesive areas were reduced by matrigel-coated islands. **b**, Significant terms from InterPro protein domain analysis of 40 predicted upstream regulators of top 200 downregulated DEGs in crowded hESCs with high expression (>10 FPKM). A full list of the 40 proteins can be found in Supplementary Table 2. B-H, Benjamini–Hochberg. **c**,**d**, Representative images (**c**) and quantifications (**d**) of immunofluorescence assay for ETV1,4,5 and OCT4 in different densities of H9 hESCs. *n* = 40 cells for ETV1, *n* = 55 cells for ETV4, *n* = 45 cells for ETV5, *n* = 77 cells for OCT4. **e**, Western blots of ETV4 in different densities of H9 hESCs. *n* = 3 independent experiments. **f**, Immunofluorescence assay for ETV4 in scratched H9 hESCs. *n* = 40 cells for 0 h, *n* = 50 cells for 3, 6 and 12 h. Cell area was measured by dividing the total surface area by the number of cells. *n* = 9 regions for relative cell area. **g**, Immunofluorescence assay for ETV4 in MCF-7 cells. *n* = 30 cells

for low density and *n* = 41 cells for high density. **h**, Immunofluorescence assay for ETV4 in H9 colonies on the cell stretching system. *n* = 60 cells. **i**, Immunofluorescence assay for ETV4 in MCF-7 cells on the cell stretching system. *n* = 59 cells for control and *n* = 46 reduced cells. **j**, Immunofluorescence assay for ETV4 and OCT4 in H9 hESCs on PDMS layers with different stiffness. ETV4: *n* = 80 cells for plastic and 15 kPa, *n* = 70 cells for 1.5 kPa; OCT4: *n* = 50 cells for plastic, *n* = 40 cells for 15 kPa and 1.5 kPa. **k**, Immunofluorescence assay for ETV4 in MCF-7 cells on PDMS layers with different stiffness. n = 45 cells. **l**, Immunofluorescence assay for ETV4 in hESCs after replating. *n* = 50 cells. *n* is number of cells (**d**,**f**,**g**,**h**,**i**,**j**,**k**,**l**) or regions (**f**) pooled from three independent experiments. Two-sided Student's *t*-test, ***P < 0.001, **P < 0.01, *P < 0.05; ns, not significant. Exact P values are presented in Supplementary Table 9. Scale bars: 25 μm (**c**,**g**,**h**,**i**,**j**,**k**,**l**), 50 μm (**f**), 100 μm (**a**). Numerical source data and unprocessed gels are available as Source data.

rule out the potential effect of cellular metabolism, these data indicate that cell-density-mediated ETV4 expression primarily depends on cell area.

Interestingly, crowded cells in the centre of large hESC colonies maintained active nuclear expression of YAP proteins despite the sharp inactivation of ETV4 (Fig. 2i and Extended Data Fig. 3f,g). We

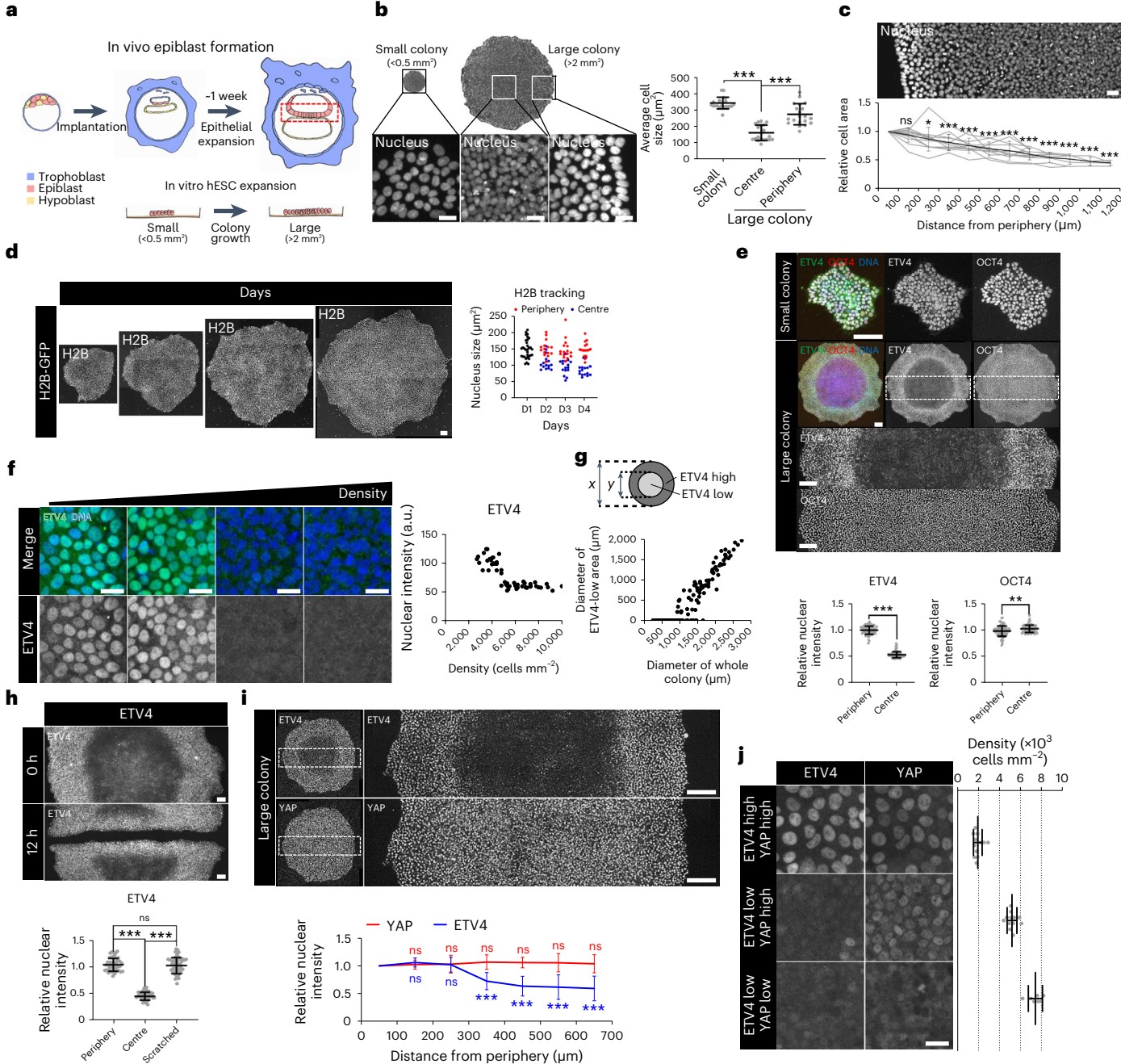

**Fig. 2 | ETV4 expression is spatiotemporally regulated by cell crowding in a growing hESC epithelium. a**, Schematic representation of in vivo epiblast formation and in vitro hESC expansion. **b,c**, Average cell area measured by dividing the total surface area by the number of cells. *n* = 20 regions (**b**) and *n* = 12 colonies (**c**). **d**, Time-course tracking assay for H2B-GFP in H9 colonies. *n* = 32 cells from three independent colony-tracking assays for H2B-GFP. **e**, Immunofluorescence assay for ETV4 and OCT4 in small and large H9 colonies. *n* = 80 cells. **f**, Immunofluorescence assay for ETV4 in different densities of hESCs. *n* = 52 cells. **g**, Quantitative analysis of the relationship between the diameters of ETV4-low areas and whole colonies in H9 hESCs. *n* = 168 colonies. **h**, Immunofluorescence assay for ETV4 in scratched H9 colonies. *n* = 50 cells.

**i**, Immunofluorescence assay for ETV4 and YAP in large H9 colonies. Nuclear intensities for ETV4 and YAP signals were measured in single cells. *n* = 7 colonies. **j**, Cell density measurements in ETV4high/YAPhigh, ETV4low/YAPhigh and ETV4low/YAPlow regions of hESC colonies. *n* = 12 regions for ETV4high/YAPhigh and ETV4low/YAPhigh; *n* = 7 regions for ETV4low/YAPlow. *n* is number of cells (**d,e,f,h**), colonies (**c,g,i**) or regions (**b**) pooled from three independent experiments; or number of regions (**j**) pooled from four independent experiments. Two-sided Student's *t*-test, ***P < 0.001, **P < 0.01, *P < 0.05; ns, not significant. Exact P values are presented in Supplementary Table 9. Scale bars: 25 μm (**b,c,d,f,j**), 50 μm (**h**), 100 μm (**e**) and 200 μm (**i**). Numerical source data are available as Source data.

employed two shRNAs to target YAP (Extended Data Fig. 3h) and subsequently validated the accuracy of the YAP immunostaining results (Extended Data Fig. 3i). To gain deeper insights, we seeded hESCs at varying densities. Although nuclear expression of both ETV4 and YAP proteins was seen in a low density (~2,000 cells mm⁻²), inactivation of ETV4, but not YAP, took place in a medium density (~5,000 cells mm⁻²;

Fig. 2j). At a high density (~7,500 cells mm⁻²), nuclear expression of both ETV4 and YAP proteins was reduced (Fig. 2j). The density ranges we used in this study are equivalent to those from other studies based on 2D gastruloid models[6]. These findings confirm that ETV4 and YAP respond differently to changes in cell density in hESCs. Given the crucial role of YAP/TAZ in epithelial proliferation[10], a uniformly high

YAP/TAZ level (Fig. 2i and Extended Data Fig. 3f,g) conforms to active cell division and high levels of the proliferation marker Ki67 and 5-ethynyl-2′-deoxyuridine (EdU) incorporation in the crowded centre (Extended Data Fig. 3j–l). YAP inhibition significantly diminished hESC colony growth (Extended Data Fig. 3m). Overall, these results demonstrate spatiotemporal regulation of ETV4 expression by cell crowding dynamics in a growing hESC epithelium.

### Derepression of the neuroectoderm fate by ETV4 inactivation

Pluripotent epiblasts undergo dynamic changes in differentiation potential as the epithelial disc undergoes size expansion before gastrulation[2,30]. Consistently, growing evidence shows that the size of in vitro hESC colonies influences the differentiation propensity[7,31,32], suggesting a direct role of epithelial expansion in lineage specification. Indeed, hESCs showed a biased differentiation toward mesendoderm (ME) when they were differentiated in a small colony. Under a robust neuroectoderm (NE)-directed differentiation condition called dual SMAD inhibition[33], small colonies of hESCs were unable to produce NE cells expressing PAX6, which is a necessary and sufficient NE marker gene in humans (Fig. 3a and Extended Data Fig. 4a)[34]. However, ME-directed differentiation by BMP4 and FGF2 efficiently turned small hESC colonies into Brachyury+ ME cells (Fig. 3a and Extended Data Fig. 4a)[35,36]. In stark contrast, large hESC colonies produced both NE and ME lineage cells with clear spatial separation. PAX6+ NE cells predominantly emerged in the centre under dual SMAD inhibition, whereas ME cells were derived in the periphery region in the presence of BMP4 and FGF2 (Fig. 3a and Extended Data Fig. 4a). These results follow previous findings based on micropatterning technology[6,7]. The NE differentiation potential of large colonies was confirmed when hESCs were spontaneously differentiated by FGF2 and TGF-β deprivation (Extended Data Fig. 4b).

Based on the above results, we hypothesized that the NE fate, initially repressed in small colonies, can be derepressed in the centre as cell crowding occurs during colony expansion. Indeed, high cell density dramatically promoted NE differentiation at the expense of ME derivation (Fig. 3b), consistent with the findings of a previous report[33]. Because ETV4 expression is downregulated in the crowded centre of large colonies, ETV4 could play a key role in suppressing the NE fate. The pre-patterns of ETV4 inactivation in undifferentiated hESC colonies precisely matched the size of the PAX6+ NE area in differentiated colonies (Fig. 3c). ETV4 expression was decreased in NE cells but not in ME cells, supporting the suppressive role of ETV4 in NE derivation (Extended Data Fig. 4c). Accordingly, ETV4 overexpression completely blocked the emergence of PAX6+ NE cells in large colonies (Fig. 3d and Extended Data Fig. 4d). Furthermore, ETV4 knockdown (KD) enabled small colonies to differentiate into NE cells (Fig. 3e and Extended Data Fig. 4e,f), which was blocked by ectopic ETV4 expression (Extended Data Fig. 4g). By contrast, ETV4 depletion impeded Brachyury+ ME cell differentiation (Extended Data Fig. 4h). Finally, cell crowding-mediated NE promotion was blocked by ETV4 overexpression (Extended Data Fig. 4i). These results demonstrate that ETV4 downregulation by cell crowding underlies NE lineage derepression.

To investigate the transcriptome-wide effect of ETV4 in hESCs, we performed RNA-seq after ETV4 KD (Fig. 3f and Supplementary Table 3). GO analysis of total DEGs ($|\log_2(\text{fold change})| > 0.5$, adjP < 0.05) showed significant enrichment in embryonic development and morphogenesis in top 10 GO terms (Extended Data Fig. 5a), supporting the key role of ETV4 in early lineage determination. Moreover, DEGs upregulated by ETV4 KD included genes related to neural differentiation with nervous system development in TOP10 GO terms (Fig. 3f and Extended Data Fig. 5b), confirming the repressive role of ETV4 in the NE derivation. To pinpoint the molecular mechanisms of ETV4, we focused on downregulated DEGs because ETV4 primarily acts as a transcriptional activator[37,38]. Downregulated DEGs included many genes related to ECM remodelling, such as matrix metalloproteinases (MMPs; Fig. 3f), with ECM organization in top 10 GO terms (Extended Data Fig. 5c). Recently,

it was reported that N-cadherin marks cells in the periphery of hESC colonies[39]. A re-analysis of published single-cell RNA-seq (scRNA-seq) data from sorted peripheral N-cadherin+ cells confirmed the elevated expression of a well-established ETV4 target gene, *DUSP6* (Extended Data Fig. 5d,e)[40]. Furthermore, genes related to ECM remodelling, such as MMPs, were significantly upregulated in N-cadherin+ peripheral cells with ECM in top 10 GO terms (Extended Data Fig. 5e,f and Supplementary Table 4). These results support the role of ETV4 in ECM remodelling in the periphery region of hESC colonies.

ETV4 is a known direct upstream regulator of MMPs in cancer[37,41–43]. ETV4 KD or overexpression altered the expression of MMPs in hESCs (Extended Data Fig. 5g–i). The treatment of pan MMP inhibitors (GM6001 and BB94) was sufficient to derepress the NE fate in small hESC colonies (Fig. 3g). By contrast, MMP inhibition disrupted ME differentiation (Fig. 3g). Interestingly, membrane-type MMPs (MT-MMPs) such as MMP14 showed higher expression in hESCs than other MMPs (Extended Data Fig. 5j). The overexpression of MMP14 phenocopied ETV4 overexpression (Extended Data Fig. 5k,l) and blocked NE derepression by ETV4 KD (Fig. 3h). Overall, these results suggest that ETV4 is an NE repressor linking cell crowding dynamics to lineage specification.

### Spatiotemporal ETV4 expression regulated by ERK

The MAPK signalling pathway is a well-known regulator of PEA3 family TFs in cancer[44]. To investigate this molecular link, we took advantage of the kinase translocation reporter (KTR) system[45]. When the kinase of interest is active, fluorescently-tagged substrates are phosphorylated and localized in the cytoplasm (Fig. 4a). The KTRs for ERK, p38 and JNK were validated by specific inhibitors (Extended Data Fig. 6a). For p38, individual cells displayed highly variable kinase activities with no discernible difference between the centre and periphery of large colonies (Extended Data Fig. 6b), whereas JNK activities were low in most cells (Extended Data Fig. 6c). However, the ERK-KTR showed clear cytoplasmic localization in most cells of small colonies (Fig. 4a). As a colony grew, a sharp reduction in ERK activity was observed in the crowded centre (Fig. 4b and Extended Data Fig. 6d,e). Induction of cell crowding was sufficient to inactivate ERK (Extended Data Fig. 6f). Live cell imaging of ERK activity revealed that the ERK activity pattern closely resembled the expression pattern of ETV4 (Fig. 4c).

ERK inhibition by chemical inhibitors (PD0325901 and U0126) induced a rapid decrease in ETV4 protein abundance without affecting the mRNA level (Fig. 4d and Extended Data Fig. 6g,h). Proteasome inhibition blocked the decrease, suggesting that ERK regulates ETV4 protein stability (Fig. 4d). ERK inhibition dramatically reduced the half-life of ETV4 proteins from 4.15 h to 0.35 h (Fig. 4e). COP1 is a critical E3 ligase for the degradation of PEA3 family TFs[46,47]. In large hESC colonies, COP1 was primarily localized in the nucleus with homogenous expression (Extended Data Fig. 7a). COP1 KD slightly increased the basal level of ETV4 protein (Extended Data Fig. 7b–d) and nullified the effect of ERK inhibition on ETV4 expression (Extended Data Fig. 7d). COP1 depletion also blocked ETV4 downregulation in the crowded centre of large hESC colonies and suppressed NE differentiation (Extended Data Fig. 7e,f).

Finally, ERK inhibition derepressed the NE fate and inhibited ME differentiation in small hESC colonies (Fig. 4f). NE derepression by ERK inhibition was completely blocked by ETV4 overexpression (Fig. 4g). ERK activation by constitutively active KRAS[G12V] increased ETV4 protein levels and suppressed NE differentiation (Extended Data Fig. 7g–i). While ERK signalling is known to have a broad impact on numerous regulatory factors, our findings suggest ETV4 as a primary target of ERK in the context of lineage specification.

### Cell-crowding-mediated regulation of receptor endocytosis

The FGF and TGF-β signalling pathways play crucial roles in maintaining pluripotency[48–51]. Short-term treatment of A83-01 (TGF-β inhibitor) showed no effect on ERK activity (Fig. 5a); however, SU-5402 (FGF

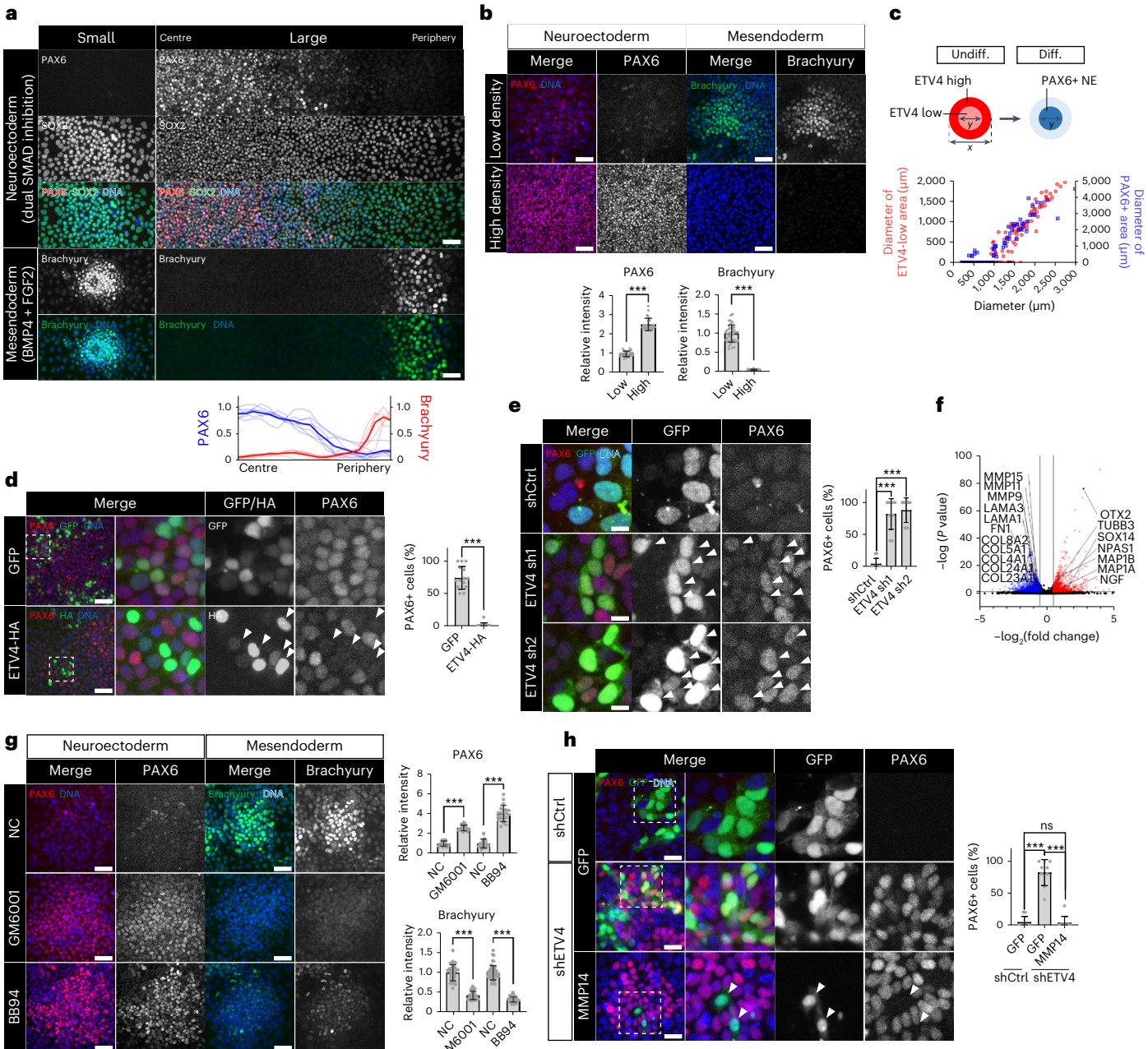

**Fig. 3 | Cell-crowding-induced ETV4 inactivation derepresses the NE fate.**
**a**, Immunofluorescence assay for PAX6, SOX2, Brachyury in differentiated H9 hESC colonies. *n* = 7 colonies. **b**, Immunofluorescence assay for PAX6 and Brachyury in H9 hESCs cells differentiated to either NE or ME cells. *n* = 25 regions for NE, *n* = 40 regions for ME. **c**, Quantitative analysis of the relationship between the diameters of ETV4-low areas in undifferentiated H9 hESC colonies and the diameters of PAX6+ areas in NE-differentiated colonies. *n* = 168 colonies for ETV4 and *n* = 92 colonies for PAX6. **d**, Immunofluorescence assay for PAX6 in large H9 colonies transduced with lentiviral vectors expressing ETV4-HA and differentiated to NE cells for 5 days. *n* = 15 regions. **e**, Immunofluorescence assay for PAX6 in small H9 colonies transduced with lentiviral vectors expressing ETV4 shRNAs together with GFP and differentiated to NE cells for 5 days. *n* = 10 regions. **f**, Volcano plot showing DEGs in H9 hESCs after ETV4 KD. The red and blue dots indicate

upregulated and downregulated genes, respectively, with cutoff values for DEGs: log₂(fold change) < −0.5 or > 0.5, adjP < 0.05. Full list of DEGs can be found in Supplementary Table 3. **g**, Immunofluorescence assay for PAX6 and Brachyury in small H9 colonies differentiated to either NE or ME cells with pan MMP inhibitors, GM6001 (5 μM) and BB94 (2 μM). NE differentiation: *n* = 21 regions for GM6001, *n* = 20 regions for BB94; ME differentiation: *n* = 38 regions for GM6001, *n* = 52 regions for BB94. **h**, Immunofluorescence assay for PAX6 in shETV4-expressing H9 hESCs transduced with lentiviral vectors expressing MMP14 together with GFP and differentiated to NE cells for 5 days. *n* = 9 regions. *n* is number of colonies (**a**,**c**) or regions (**b**,**d**,**e**,**g**,**h**) pooled from three independent experiments. Two-sided Student's *t*-test, ***P < 0.001, **P < 0.01, *P < 0.05; ns, not significant. Exact P values are presented in Supplementary Table 9. Scale bars: 10 μm (**e**), 25 μm (**h**), 50 μm (**a**,**d**,**g**), 100 μm (**b**). Numerical source data are available in Source data.

inhibitor) reduced ERK activity and the level of ETV4 proteins without altering mRNA expression (Fig. 5b,c). We used shRNAs targeting FGFR1 owing to the high expression in hESCs (Extended Data Fig. 8a–c). FGFR1 KD decreased ERK activity and ETV4 expression (Extended Data

Fig. 8d,e), suggesting FGFR1-mediated signalling as an upstream pathway of ERK and ETV4.

An emerging body of evidence shows that receptor localization contributes to regulating downstream signalling[52,53]. For example,

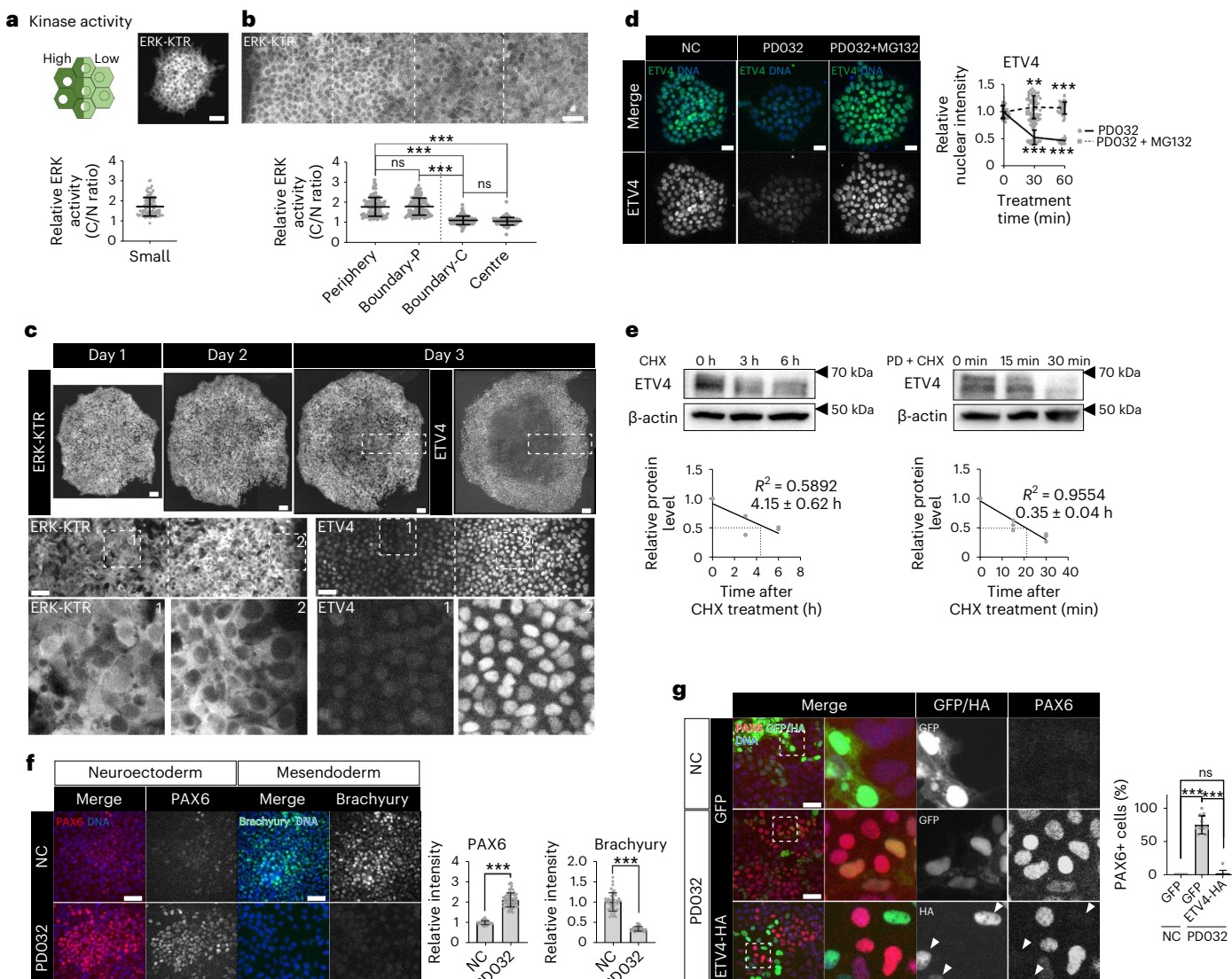

**Fig. 4 | ERK regulates the spatiotemporal ETV4 expression. a,b,** ERK activity measured by the ratio of the cytoplasmic over nuclear intensities of ERK-KTR (C/N ratio) in single cells within small (**a**) and large (**b**) H9 colonies. *n* = 95 cells for small; *n* = 85 cells for periphery and centre; *n* = 90 cells for boundary-P (peripheral side near the boundary where ERK acivity transitions) and boundary-C (central side near the boundary). **c,** Representative time-course images from three independent colony-tracking assays for ERK-KTR in H9 colonies, followed by immunofluorescence assay for ETV4. **d,** Immunofluorescence assay for ETV4 in H9 hESCs treated with PD0325901 (1 µM) and MG132 (10 µM). *n* = 60 cells. **e,** Quantification of ETV4 protein stability in H9 hESCs in the presence of PD0325901 (1 µM). For CHX, *n* = 3 (0 h), 3 (3 h) and 2 (6 h) independent experiments. For PD + CHX, n = 3 independent experiments.

PD, PD0325901; CHX, cycloheximide. **f,** Immunofluorescence assay for PAX6 and Brachyury in small H9 colonies differentiated to either NE or ME cells with PD0325901 (1 µM). NE differentiation: *n* = 75 regions for NC, *n* = 69 regions for PD0325901; ME differentiation: *n* = 42 regions. **g,** Immunofluorescence assay for PAX6 in H9 hESCs transduced with lentiviral vectors expressing ETV4-HA and differentiated to NE cells for 5 days with PD0325901 (1 µM). *n* = 11 regions. *n* is number of cells (**a,b,d**) or regions (**f,g**) pooled from three independent experiments. Two-sided Student's *t*-test, ***P < 0.001, **P < 0.01, *P < 0.05; ns, not significant. Exact *P* values are presented in Supplementary Table 9. Scale bars: 25 µm (**d**), 50 µm (**a,b,f,g**), 100 µm (**c**). Numerical source data, unprocessed gels and additional microscope images are available as Source data.

it was recently reported that lateral localization of TGF-β receptors impeded cellular responses to apically applied ligands[6,54]. However, we observed that cell-crowding-induced ETV4 inactivation occurred in hESC culture on a transwell system, where ligands are accessible to both apical and basolateral sides of cells (Extended Data Fig. 8f). In the case of receptor tyrosine kinases, endocytosis is functionally related to the downstream signalling activation[52]. Activated EGFR and FGFR accumulate in endosomes that serve as a signalling platform[55], and blocking receptor endocytosis inhibits the downstream signalling[56–58]. Recently, it was reported that membrane tension influenced FGFR endocytosis and downstream ERK activity in mouse ESCs, implicating mechanical regulation of receptor endocytosis[59,60]. To test this idea,

we generated stable hESC lines expressing GFP reporters for various endosomes and lysosomes[61]. In small hESC colonies, a significant number of FGFR1-containing vesicles colocalized with RAB5A⁺ early endosomes (Fig. 5d), a finding that was confirmed using another early endosome marker, EEA1 (Fig. 5e). FGFR1-containing vesicles were also observed in late endosomes (RAB7A), recycling endosomes (RAB11B) and lysosomes (LAMP1; Fig. 5d). These results suggest that FGFR1 proteins are under active endocytosis, followed by recycling or degradation. Strikingly, cells in large hESC colonies showed distinct subcellular localizations of FGFR1 proteins depending on their positions. These proteins were predominantly present in endosome vesicles in the periphery of large colonies (Fig. 5e and Extended Data Fig. 9a). In

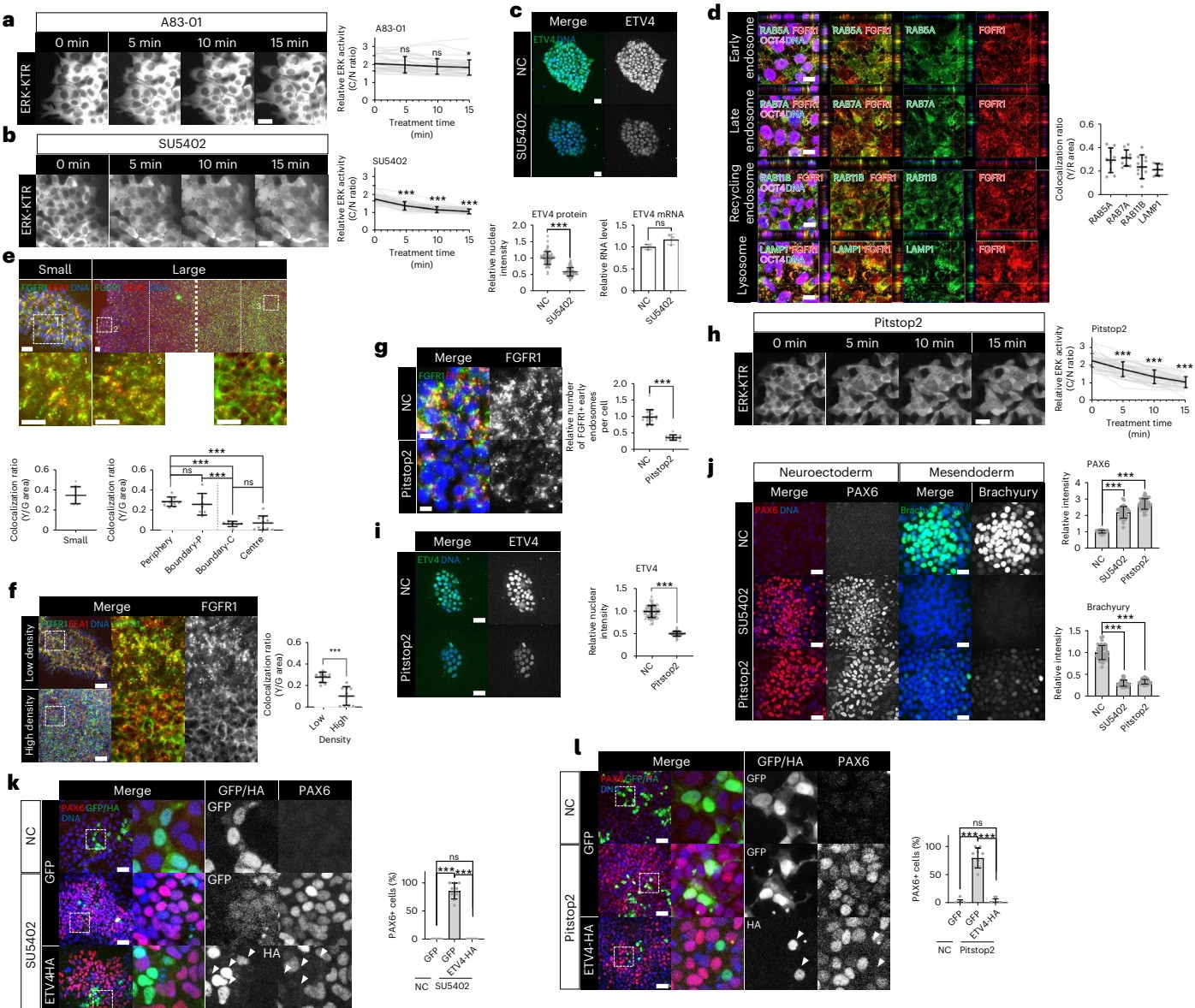

**Fig. 5 | Cell crowding controls the endocytosis of FGF receptors. a,b**, Time-course images for ERK-KTR in H9 hESCs treated with A83-01 (5 µM; **a**) or SU5402 (10 µM; **b**). ERK activity was measured by the ratio of the cytoplasmic over nuclear intensities. $n = 35$ cells. **c**, Immunofluorescence assay and qPCR analysis for ETV4 in H9 hESCs treated with SU5402 (10 µM, 1 h). $n = 77$ cells for immunofluorescence assay and $n = 3$ samples for qPCR. **d**, Immunofluorescence assay for FGFR1 in H9 hESCs expressing various reporters. Colocalization was measured by the yellow/red (Y/R) ratio. $n = 9$ regions for RAB5A and LAMP1, $n = 10$ regions for RAB7A and RAB11B. **e**, Immunofluorescence assay for FGFR1 and EEA1 in small and large H9 colonies. Colocalization was measured by the yellow/green (Y/G) ratio. $n = 8$ regions for small, n = 10 regions for periphery and centre, $n = 7$ regions for boundary-P and boundary-C. **f**, Immunofluorescence assay for FGFR1 and EEA1 in H9 hESCs in low- and high-density cultures (24 h after seeding). Colocalization was measured by the Y/G ratio. $n = 9$ regions for low density and $n = 10$ regions for high density. **g**, Immunofluorescence assay for FGFR1 and EEA1 in H9 hESCs

treated with Pitstop2 (50 µM, 30 min). $n = 15$ regions for NC and $n = 13$ regions for Pitstop2. **h**, Time-course images for ERK-KTR in H9 hESCs treated with Pitstop2 (50 µM). $n = 40$ cells. **i**, Immunofluorescence assay for ETV4 in H9 hESCs treated with Pitstop2 (50 µM, 30 min). $n = 120$ cells. **j**, Immunofluorescence assays for PAX6 and Brachyury in small H9 colonies differentiated to either NE or ME cells with SU5402 (10 µM) or Pitstop2 (20 µM). $n = 40$ regions. **k**, Immunofluorescence assay for PAX6 in small H9 colonies transduced with lentiviral vectors expressing ETV4-HA and differentiated to NE cells for 5 days with SU5402 (10 µM). $n = 9$ regions. **l**, Immunofluorescence assay for PAX6 in small H9 colonies transduced with lentiviral vectors expressing ETV4-HA and differentiated to NE cells for 5 days with Pitstop2 (20 µM). $n = 9$ regions. $n$ is number of cells (**a,b,c,h,i**) or regions (**d,e,f,g,j,k,l**) pooled from three independent experiments. Two-sided Student's $t$-test, ***$P < 0.001$, **$P < 0.01$, *$P < 0.05$; ns, not significant. Exact $P$ values are presented in Supplementary Table 9. Scale bars: 10 µm (**d,g**), 25 µm (**a,b,c,e,h,i,j**), 50 µm (**f,k,l**). Numerical source data are available as Source data.

stark contrast, membrane localization of FGFR1 proteins was observed in the crowded centre, suggesting impaired endocytosis (Fig. 5e and Extended Data Fig. 9a, b). FGFR1 proteins were localized in both apical and basolateral membranes in the crowded centre (Extended Data Fig. 9a). The transition of FGFR1 localization occurred abruptly along the radial axis of large hESC colonies (Fig. 5e). To further validate these

results, we applied a DNA aptamer (TD0) previously developed as an agonist of FGFR1 (Extended Data Fig. 9c)[62]. Fluorophore-tagged aptamers were found in FGFR1-containing early endosomes in the periphery of large hESC colonies (Extended Data Fig. 9d). By contrast, few aptamer-containing early endosomes were detected in the centre (Extended Data Fig. 9d). Furthermore, induction of cell crowding was

sufficient to impair FGFR1 endocytosis with loss of EEA1 colocalization (Fig. 5f). These results show that FGFR1 endocytosis is tightly regulated by cell crowding.

Next, we used Pitstop2 (a clathrin inhibitor) and Dynasore (a dynamin inhibitor) to suppress endocytosis (Extended Data Fig. 9e)[63,64]. Endocytosis inhibition dramatically decreased the number of FGFR1-containing endosomes, ERK activity and ETV4 expression (Fig. 5g–i and Extended Data Fig. 9f–h), which phenocopied the effects of FGFR inhibition (Fig. 5b,c). Constitutively active KRAS[G12V] blocked the effect of endocytosis inhibition on ETV4 expression (Extended Data Fig. 9i), suggesting that endocytosis regulates ETV4 through the FGFR–RAS signalling.

Consistent with previous research[65,66], we found that FGFR inhibition was sufficient to derepress the NE fate within small hESC colonies (Fig. 5j and Extended Data Fig. 9j), while suppressing the ME derivation (Fig. 5j). Notably, we observed a similar phenotypic outcome upon the inhibition of endocytosis (Fig. 5j). Furthermore, ectopic ETV4 expression entirely counteracted the NE-promoting effects induced by FGFR and endocytosis inhibitors (Fig. 5k,l). Overall, our findings suggest that cell-crowding-mediated blockade of FGFR1 endocytosis leads to the inactivation of both ERK and ETV4.

### Regulation of receptor endocytosis by integrin–actomyosin

To figure out molecular mechanisms bridging cell crowding to FGFR1 endocytosis, we revisited the RNA-seq data performed in crowded hESCs. Downregulated DEGs in crowded hESCs showed significant enrichment of genes related to cell–ECM interactions such as integrins (Fig. 6a and Extended Data Figs. 1b and 10a). Published scRNA-seq data from N-cadherin[+] peripheral cells also showed focal adhesion in top 10 KEGG pathways (Extended Data Fig. 10b)[39]. These results concord with the fact that cell crowding leads to a smaller surface area through which cells interact with the ECM.

Consistent with the transcriptomic results, a high level of integrin β1 was observed in paxillin[+] focal adhesions in small colonies and the periphery of large colonies (Fig. 6b). By contrast, crowded cells in the centre showed perinuclear localization of integrin β1 with loss of paxillin colocalization (Fig. 6b). Focal adhesion kinase (FAK) is a critical component of integrin signalling[67]. Immunostaining of phosphorylated FAK (pFAK, active FAK) revealed the sharp downregulation in response to increased cell density in hESCs (Fig. 6c). Furthermore, FAK inhibition suppressed FGFR1 endocytosis, ERK activity and ETV4 expression (Extended Data Fig. 10c–f). These findings suggest that integrin signalling bridges cell crowding to FGFR1 endocytosis.

Although integrin signalling has the potential to impact a wide array of cellular processes, we focused on its role in regulating cytoplasmic actin filaments, given the significance of actin dynamics in receptor endocytosis[68–70]. Consistent with the patterns of integrin β1[+] focal adhesions, small hESC colonies showed well-established pMLC[+] bundles, while the zonation of pMLC staining appeared in large hESC colonies with a sharp reduction in the centre (Fig. 6d,e). Disassembly

of pMLC[+] bundles was observed by integrin inhibition (RGDS peptide), FAK inhibition or cell crowding (Extended Data Figs. 1e and 10g,h). We used either blebbistatin (myosin inhibitor) or YM (Y-27632, RhoA kinase inhibitor; ML-7, myosin light chain kinase inhibitor) to inhibit actomyosin (Extended Data Fig. 10i,j)[71]. Transient inhibition of actomyosin activity decreased the number of FGFR1-containing endosome vesicles and inactivated ERK and ETV4 (Fig. 6f–h and Extended Data Fig. 10d,k,l). Importantly, the transient activation of actomyosin by doxycycline-inducible expression of constitutively active RhoA[Q63L] partially re-activated FGFR1 endocytosis in the crowded centre of large hESC colonies (Fig. 6i and Extended Data Fig. 10m). These findings suggest that actomyosin, in addition to serving as a marker of cellular mechanical stress, plays a crucial role in regulating receptor endocytosis in response to mechanical stimuli. However, it is essential to acknowledge that other cellular changes induced by cell crowding, such as the reduction in apical surface area for ligand interaction and alterations in cell geometry, may also contribute to regulating receptor endocytosis.

Finally, actomyosin inhibition derepressed the NE fate in small hESC colonies while preventing ME differentiation (Fig. 6j). The NE derepression by actomyosin inhibition was blocked by ETV4 over-expression (Fig. 6k). Overall, these results suggest that the integrin–actomyosin pathway serves as a crucial link connecting cell crowding to FGFR1 endocytosis, ETV4 expression and NE specification (Fig. 6l).

### Dynamic ETV4 expression captured by mathematical modelling

A critical question that remains to be answered in this study is how the gradient of cell crowding from the periphery to the centre produces a sharp boundary in ETV4 expression. To address this question, we propose a mathematical framework that describes a cell crowding model. Based on real-time imaging data of H2B-GFP hESC colony growth, we obtained an analytical solution to the governing reaction-diffusion partial differential equation describing cell crowding dynamics (Methods). This equation assumes a cell population density varying in time and space due to the flow of cells (diffusion) from high- to low-density regions (outward flow) as well as cell division (reaction). A similar approach has been used to describe bacterial colony growth[72]. Our model accurately predicted the colony population and colony size (diameter; Fig. 7a). Moreover, the model predicted an increase in cell density over time and higher cell crowding at the colony centre (Fig. 7b).

Cell crowding inversely correlates with ECM-accessible area, linking ECM ligand availability to cell density. Moreover, it is well established that integrin–ECM binding kinetics follow a cooperative interaction[73,74]. Such an interaction enforces a system with an on/off switch-like behaviour. We used the Hill equation to model integrin–ECM interactions[75]. The equation captures the biomolecular interaction that exhibits the cooperativity between two binding molecules. Our model predicts a switch for integrin activity that occurs at different distances from the colony's centre as the colony grows over time (Fig. 7c).

**Fig. 6 | The integrin–actomyosin pathway regulates FGFR endocytosis. a**, GSEA with the RNA-seq data from H9 hESCs in low- and high-density cultures. GSEA was performed with three different gene sets: cell–substrate junction (left), cell–ECM interaction (middle) and integrin adhesome (right). **b**, Immunofluorescence assay for Integrin β1 and Paxillin in small and large H9 colonies. $n = 9$ regions for small, $n = 13$ regions for periphery, and $n = 10$ regions for centre. **c**, Immunofluorescence assay for pFAK and OCT4 in different densities of H9 hESCs. $n = 69$ regions. **d,e**, Immunofluorescence assay for pMLC in small (**d**) and large (**e**) H9 colonies. $n = 30$ regions. **f**, Immunofluorescence assay for FGFR1 and EEA1 in small H9 colonies treated with Blebbistatin (50 μM, 30 min). $n = 10$ regions. **g**, Time-course images for ERK-KTR in H9 hESCs treated with Blebbistatin (50 μM). ERK activity was measured by the ratio of the cytoplasmic over nuclear intensities (C/N ratio). $n = 30$ cells. **h**, Immunofluorescence assay for ETV4 in H9 hESCs treated with Blebbistatin (50 μM, 1 h). $n = 241$ cells for NC and $n = 230$ cells for Blebbistatin.

**i**, Immunofluorescence assay for FGFR1 and EEA1 in large H9 colonies transduced with lentiviral vectors expressing constitutively active RhoA[Q63L]-GFP in a doxycycline-dependent manner. $n = 25$ regions. **j**, Immunofluorescence assay for PAX6 and Brachyury in small H9 colonies differentiated to either NE or ME cells with Blebbistatin (1 μM). NE differentiation: $n = 37$ regions, ME differentiation: $n = 45$ regions for NC and $n = 38$ regions for Blebbistatin. **k**, Immunofluorescence assay for PAX6 in H9 hESCs transduced with lentiviral vectors expressing ETV4-HA and differentiated to NE cells for 5 days with Blebbistatin (1 μM). $n = 9$ regions. **l**, Graphic summary of the mechanotransduction pathway regulating ETV4 expression and lineage fates. $n$ is number of cells (**g,h**) or regions (**b,c,e,f,i,j,k**) pooled from three independent experiments. Two-sided Student's $t$-test, ***$P < 0.001$, **$P < 0.01$, *$P < 0.05$; ns, not significant. Exact $P$ values are presented in Supplementary Table 9. Scale bars: 10 μm (**b,i**), 25 μm (**c,d,e,f,g**), 50 μm (**h,j,k**). Numerical source data are available as Source data.

The ultrasensitive transition of integrin activity appears when the core cell density reaches a critical density of 5,000 cells mm$^{-2}$ (Fig. 7d), which matches well with the pFAK and ETV4 experimental data (Figs. 2f, 6c and 7e). Furthermore, our model recapitulated the experimental data with the ETV4 low-expression region appearing in the centre when the colony size reaches 1,000–1,500 μm in diameter (Fig. 7f,g). The critical cell density can only be observed after $t > t_{cr}$ (where $t$ represents time

and $t_{cr}$ represents the critical time when the ETV4 transition boundary appears for the first time), and the radius at which the ETV4 transition occurs varies as a function of time (Fig. 7h). Although our model successfully demonstrated the ultrasensitive transition of integrin activity and ETV4 expression based on the ECM availability and the cooperativity in integrin–ECM interaction, it is essential to acknowledge that cell shape itself may also contribute to regulating integrin signalling or the

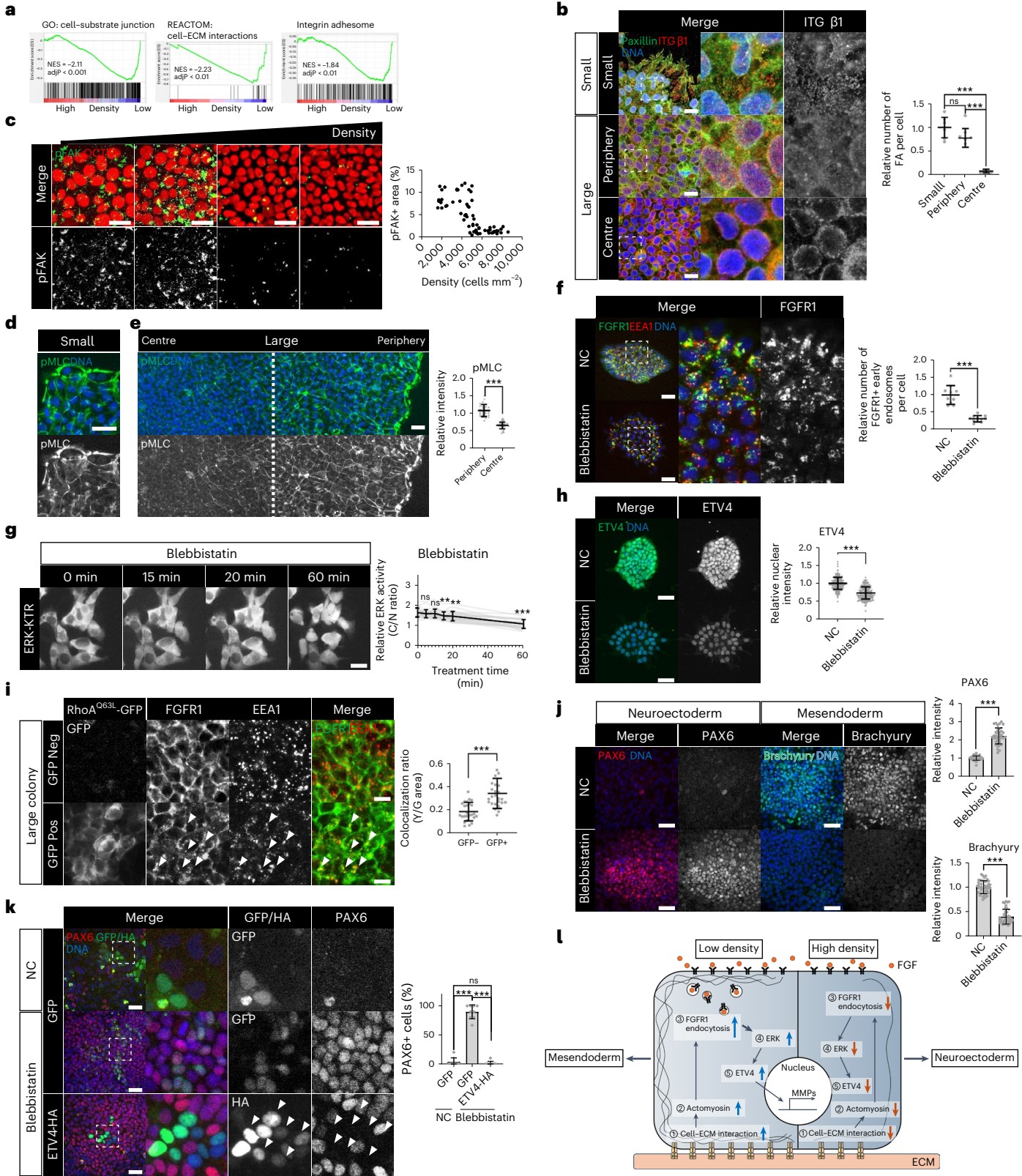

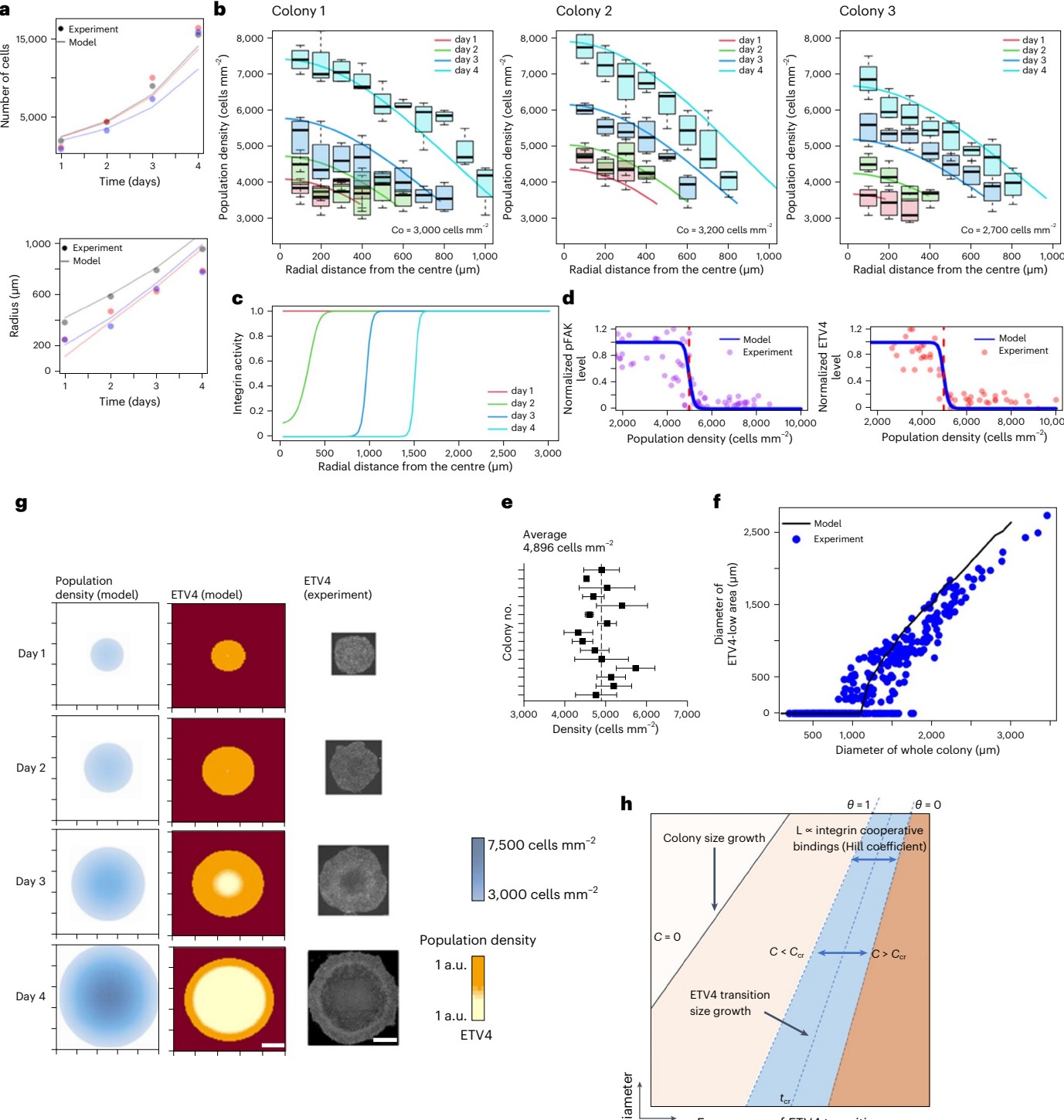

**Fig. 7 | Mathematical modelling links cell crowding to ETV4 ultrasensitivity.**
**a**, Model (solid lines) for hESC colony growth that accurately predicts colony growth and proliferation dynamic. Different colours represent three different colonies. **b**, Comparison between model and observed spatiotemporal change in cell population density. Solid lines show model prediction and boxplots represent the range of experimentally measured cell population densities at certain distance on days 1, 2, 3 and 4. The three panels illustrate three different colonies with an initial average cell population density $C_0$. The black lines in the middle of the boxes are the median values for each group. The vertical size of the boxes illustrates the interquartile range. Whiskers represent 1.5 × interquartile range. $n = 4$ regions within a colony for each distance. **c,d**, Model prediction for integrin activity (**c**), pFAK and ETV4 (**d**) transition that occurs at a critical cell

population density of -5,000 cells mm⁻². Simulation generated for an example colony with an initial radius of 250 μm and an initial average cell population density 3,000 cells mm⁻². **e**, Critical cell densities for ETV4 inactivation measured across H9 colonies. $n = 3$ regions within a colony. **f**, Comparison between model and experimental measurement for ETV4 transition (Fig. 2g). Simulation generated for an example colony with an initial radius of 250 μm and an initial average cell population density 3,000 cells mm⁻². $n = 168$ colonies pooled from three independent experiments. **g**, Model for spatiotemporal change of population density and radial distribution of ETV4. Scale bar, 600 μm. Simulation generated for an example colony with an initial radius of 250 μm and an initial average cell population density 3,000 cells mm⁻². **h**, ETV4 transition map as a function of space and time.

downstream pathways independently of ECM availability[76]. Overall, our model bridges the simple growth of stem cell epithelium to the timing and location of ETV4 inactivation and NE derepression.

## Discussion

An outstanding question in developmental biology is how multiple lineages arise from a single-layered epithelium of seemingly homogeneous stem cells. During the expansion of hESCs, active proliferation prompts local cell crowding in the centre of epithelial colonies. A gradual change in cell crowding induces the abrupt inactivation of integrin signalling based on the cooperativity of the integrin–ECM interaction. The blunted integrin–actomyosin pathway impedes FGF receptor endocytosis, thereby derepressing the NE fate from ETV4 (Fig. 6l). Importantly, ETV4, formerly known as an oncogene, serves as a mechanotransducer linking cell crowding dynamics to lineage specification in a stem cell epithelium.

Previous studies have employed hESC-derived gastruloid models grown on micropatterned plates to study molecular mechanisms for embryonic germ layer derivation[6,7,77,78]. Although a differentiation signal, BMP4, affects all cells in a culture plate, distinct lineages emerge in a defined position along the radial axis of hESC colonies. Reaction–diffusion models have been used to explain lineage patterning, where asymmetric gradients of signal activators and inhibitors determine the positional identity of differentiating cells[79,80]. In hESCs, BMP4 directly stimulates the expression of NOGGIN, a BMP antagonist, providing a reaction–diffusion mechanism for pattern formation[6,7]. In addition, cell-density-dependent relocalization of TGF-β receptors to the basolateral side restricts signalling activation, contributing to gastruloid formation[6]. In this study, we identified ETV4 as a mechanotransducer for lineage determination. ETV4 responds to cell crowding dynamics evoked by epithelial expansion, generating a pre-pattern with a sharp boundary for future lineage fates. Our findings offer an independent patterning mechanism, directly connecting mechanical microenvironments to lineage specification.

In an in vitro ESC differentiation, NE derivation is deemed the default pathway because minimal medium conditions without additional signalling molecules are sufficient to direct ESCs toward the NE lineage[81]. However, we found that derepression from ETV4 is essential for successful NE derivation. This finding suggests that, at first, hESCs are not inclined towards NE differentiation. The withdrawal of signalling molecules such as FGF acts as a signal to inactivate ETV4 and thereby induces derepression of the NE fate in hESCs. These results provide a new perspective in understanding lineage fate determination during early hESC differentiation.

Core pluripotency genes are essential for the self-renewal of hESCs. At the same time, these genes play critical roles in regulating lineage specification[82]. Upon differentiation, OCT4 and NANOG promote the ME lineage, while SOX2 drives NE derivation[83,84]. These findings suggest that the core pluripotency genes act as the uppermost regulators of stem cell self-renewal and differentiation. In this study, however, we propose that ETV4 impacts lineage fate determination earlier than core pluripotency genes. In large hESC colonies, ETV4 protein expression sharply decreases in the crowded centre before the expression of any core pluripotency genes alters, highlighting ETV4 as the earliest determinant of lineage fates in hESCs.

Epithelial cell crowding activates contact inhibition, mediated by YAP/TAZ and PIEZO1, to suppress proliferation and maintain tissue homeostasis[10,13,15,85,86]. However, the mechanism by which cell crowding decouples lineage fate determination from contact inhibition of proliferation is unclear. Here, we identified that ETV4 acts as an on/off switch with ultrasensitive dependence on cell crowding. Sharp ETV4 inactivation by cell crowding derepresses the NE fate. At the same time, sustained YAP activity safeguards stem cell proliferation in a high-density environment. In such a setting, hESCs can utilize spatiotemporal heterogeneity in cell density to derive multiple lineages while maintaining active proliferation. Beyond stem cell differentiation, ETV4 is a critical oncogene with elevated expression in multiple cancers[44,87]. Therefore, our discovery of ETV4 as a mechanical transducer provides new insights into the mechanisms underlying tumour progression and suppression.

## Online content

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

## Methods

### Maintenance of cell lines

H9 (WA09) and H1 (WA01) hESCs were purchased from WiCell and cultured on Matrigel (Corning) either in mTeSR1 (Stem Cell Technologies) or TeSR-E8 (Stem Cell Technologies) medium. H9 and H1 hESCs were authenticated by short tandem repeat analysis. hESCs were maintained at 37 °C with 5% $CO_2$ and were passaged every 3–4 days by ReLeSR (Stem Cell Technologies). This work was approved by the Human Stem Cell Research Oversight Committee at Pohang University of Science and Technology (PIRB-2021-R035). HEK293T cells (CRL-11268, ATCC) and MCF-7 cells (HTB-22, ATCC) were cultured in DMEM supplemented with 10% fetal bovine serum (FBS). ARPE19 cells (CRL-2302, ATCC) were cultured in DMEM/F12 with 10% FBS. Cells were maintained at 37 °C with 5% $CO_2$ and were passaged every 2–3 days by trypsin (Welgene). To manipulate substrate stiffness, H9 hESCs and MCF7 cells were plated and cultured on 35 mm imaging dishes with 15 and 1.5 kPa PDMS layers (ibidi). To facilitate cell attachment, we applied the dishes to a thin Matrigel coating (1% Matrigel diluted in basal media). The stiffness values reported in our study represent the properties of the underlying plastic and hydrogel substrates.

### Differentiation of hESCs

For spontaneous differentiation, hESCs on Matrigel were induced to differentiate with basal hESC culture media (DMEM/F12, 15% knockout serum replacement, MEM nonessential amino acid solution, and 0.1 mM 2-mercaptoethanol) without FGF2 and TGF-β. For neuroectoderm differentiation, hESCs were differentiated on Matrigel with hESC culture media (DMEM/F12, 5% knockout serum replacement, MEM nonessential amino acid solution, and 0.1 mM 2-mercaptoethanol) containing SB431542 (PeproTech, 10 μM) and Dorsomorphin (Tocris, 2 μM). For mesendoderm differentiation, hESCs were differentiated in mTeSR1 medium (containing FGF2) supplemented with recombinant human BMP4 (R&D Systems, 5 ng ml$^{-1}$).

### High-density culture of hESCs

Small drops of Matrigel (2 μl) were deposited on a chamber slide glass (Thermo) to create a controlled pattern of the adhesive surface. The prepared micropattern chamber slides were sealed with parafilm and incubated for 1 h at room temperature before use. hESCs were dissociated by ReLeSR (Stem Cell Technologies) and resuspended in 1 ml of mTeSR1 or TeSR-E8 medium. The entire resuspended solution was seeded on a micropattern chamber slide. After cell adhesion, excess cells were removed by washing, and adherent cells were left to grow for ~2–3 days to cover the whole patterned area. The cell densities used in this study, ranging from 2,000 to 10,000 cells mm$^{-2}$, match the parameters established in prior research utilizing 2D gastruloid models[6].

### Assessment of hypoxic cells

Hypoxic cells were detected using a Hypoxyprobe kit (Hypoxyprobe) according to the manufacturer's instructions. hESCs cultured under a hypoxic condition (5% $O_2$, 5% $CO_2$ and 90% $N_2$) were used for validation. Cultured cells were treated with pimonidazole to a final concentration of 400 μM at 37 °C for 2 h. Hypoxic cells were then detected by fluorescence imaging.

### Glucose uptake assay

Glucose uptake was assessed using a glucose uptake assay kit (Dojindo) following the manufacturer's instructions. To validate the assay, hESCs were incubated at 4 °C for 15 min to inhibit glucose uptake. Cultured cells were treated with the Glucose Uptake Probe to a final concentration of 5 μM at 37 °C for 15 min. The cells were washed three times with cold washing solution and replenished with cold washing solution at 4 °C. Glucose uptake was subsequently visualized by fluorescence imaging.

### Lentiviral preparation and concentration

Lentiviral vector plasmids were transfected into HEK293T cells with second-generation packaging vectors psPAX2 (Addgene, 12260) and pMD2.G (Addgene, 12259; Supplementary Table 6). Transfected HEK293T cells were cultured at 37 °C with 5% $CO_2$ for 2 days. Supernatants were filtered using 0.45 μm filters (Corning) and concentrated overnight at 4 °C using Lenti-X Concentrator (Takara Bio). Concentrated lentiviruses were resuspended in PBS and stored at −80 °C.

### Generation of hESC reporter lines

Concentrated lentiviral particles for KTRs and endosome/lysosome reporters were transduced to H9 hESCs grown on Matrigel with TeSR-E8. Two days after transduction, 2 μg ml$^{-1}$ puromycin (Gibco) was added to culture medium. Puromycin selection was performed for at least 4 days to generate stable hESC reporter lines.

### Immunofluorescence staining

All samples were fixed using 3.7% methanol-free formaldehyde (Thermo) for 15 min at room temperature. After fixation, samples were washed 3 times with PBS and permeabilized with 0.25% Triton X-100 (SIGMA)-supplemented PBS for 10 min at room temperature. Samples were blocked with 10% FBS in PBS for 1 h at room temperature. The primary antibodies diluted in the blocking solution were treated overnight at 4 °C (Supplementary Table 5). After washing three times with PBS, an appropriate Alexa Fluor dye-conjugated secondary antibody was used to treat samples for 1 h at room temperature. Nuclear staining was performed on samples for 2 min with Hoechst 33342 (Thermo). Images were captured by a fluorescence microscope (Leica DMi8) or confocal microscope (ZEISS LSM800). Sample images were prepared in imageJ 1.53 software (Fiji), and statistical analysis was performed using the GraphPad Prism 9.1.0 software (GraphPad Software).

### Cell shape analysis

The CSI is calculated based on measurements obtained from ZO1 staining images, where each cell's boundary is outlined, and its area and perimeter are measured. To calculate the CSI, we used the following formula: $CSI = 4\pi \times area/perimeter^2$.

### EdU incorporation assay

To label actively proliferating cells, we used the Click-iT EdU Imaging Kit (Thermo) following the manufacturer's instructions. hESCs were treated with EdU to a final concentration of 5 μM at 37 °C for 6 h. After EdU labelling, hESCs were washed with PBS and fixed using 3.7% methanol-free formaldehyde (Thermo) for 15 min at room temperature. EdU was detected by the Click-iT reaction protocol.

### Crystal violet staining

Cells were washed once with PBS and stained at room temperature for 1 min with 1 ml of crystal violet staining solution (SIGMA). After staining, samples were washed three times with PBS, and dried at room temperature for 15 min.

### Quantitative real-time PCR

Total RNA was isolated using QIAzol Lysis Reagent (QIAGEN) and reversely transcribed with SuperiorScript III Master Mix (Enzynomics). Quantitative RT-PCR analysis was performed on the CFX Connect Real-Time PCR Detection System (BIO-RAD) with TOPreal qPCR 2X PreMIX (Enzynomics) (Supplementary Table 7). GAPDH was used as a normalization control. Results were plotted using the GraphPad Prism 9.1.0 software (GraphPad Software).

### Western blot

Cells were lysed with RIPA lysis buffer supplemented with Protease Inhibitor Cocktail Kit5 (Quartett). Protein concentration was measured using the BSA Protein Assay kit (Thermo). The same amount of

protein sample was separated on SDS-PAGE and then transferred to a nitrocellulose membrane (BIO-RAD). Membranes were blocked with PBST (0.1% Tween 20 in PBS) containing 5% skim milk; then immunoblotting was performed overnight at 4 °C with primary antibodies (Supplementary Table 5). The membranes were stained with an appropriate HRP-conjugated secondary antibody for 1 h at room temperature. Signals were detected with Amersham imager 680 (Amersham), and relative signal intensity was quantified by Multi gauge 3.0 (FUJIFILM).

## Cell stretching experiment

A manual cell stretching system STB-100 (Togetherbio) was used to control cell area directly. Stretch chambers (STB-CH-4W) were coated with Matrigel. Cells were seeded in a chamber that was stretched by 20% for 2 days. The chamber was then gently released to reduce cell area and incubated for 24 h before further experiments.

## Cell scratch experiment

Cells were scraped using P10 pipet tips, washed with 1 ml of culture medium to get rid of the debris, and then fed with 2 ml of culture medium.

## RNA sequencing analysis

Using the Illumina TruSeq Stranded mRNA Sample Preparation Kit (Illumina), 1 μg of the total RNA of the sample was prepared to create a library. The poly-A-containing mRNA molecules were purified using poly-T-attached magnetic beads. The mRNAs were copied into the cDNA library using SuperScript II reverse transcriptase (Invitrogen). Following the qPCR Quantification Protocol Guide (KAPA BIOSYSTEMS), the library was quantified using the KAPA Library Quantification Kit for Illumina Sequencing Platform and qualified using TapeStation D1000 ScreenTape (Agilent Technologies). Then, the indexed library was submitted to Illumina NovaSeq (Illumina), and pair-end (2 × 100 bp) sequencing was performed. The raw reads from the Illumina NovaSeq were trimmed using TRIMMOMATIC and mapped to the *Homo sapiens* transcript reference (GRCh37) using HISAT v2.1.0. Transcript assembly of known transcripts was processed by StringTie v2.1.3b. The expression abundance of genes was calculated as read counts or FPKM values per sample and filtered through statistical hypothesis testing for additional analysis such as DEGs.

## Differential gene expression analysis

We used DESeq2[88] (https://bioconductor.org/packages/DESeq2/) to perform differential gene expression analysis using bulk RNA-seq data. To identify DEGs from scRNA-seq data, we used DEsingle[89] implementation in R/Bioconductor (https://bioconductor.org/packages/DEsingle). *P* values were adjusted for multiple comparisons using the FDR approach.

## Cellular crowding model

A reaction–diffusion equation is used to model colony growth dynamics as

$$\frac{\partial C(r,t)}{\partial t} = D\left(\frac{\partial^2 C(r,t)}{\partial r^2} + \frac{1}{r}\frac{\partial C(r,t)}{\partial r}\right) + \alpha C(r,t) \tag{1}$$

by capturing spatial variation in colony cell population density through diffusion as well as cell proliferation (first and second term in equation (1)) where $C$ defines cell population density and $r$ is defined as the radial distance from the centre of the colony. $D$ and $\alpha$ represent diffusion coefficient and cellular division rate and can be measured experimentally. For example, hESC division rate was inferred from the range of experimentally observed cellular doubling rate using $\alpha = \ln(2)/$(doubling rate) from previous reports[90]. Diffusion coefficient was also estimated using a least squares fit to Fig. 7a. Spatially homogenous solution to equation (1) represents colony population exponential

growth (that is the number of cells in the colony) with the division rate $\alpha$. With the following initial and boundary conditions

$$C(r, t = 0) = \begin{cases} C_o, r < r_0 \\ 0, r \geq r_0 \end{cases} \tag{2}$$

$$C(r = 0, t) < \infty, C(r = 1, t) = 0 \tag{3}$$

solution to equation (1) can be found as

$$C(r,t) = \sum_{k=1}^{\infty} A_k e^{-(D\lambda_k^2 - \alpha)t^*} J_0(\lambda_k r^*) \tag{4}$$

where $A_k = \frac{2C_0 r_0 J_1(\lambda_k r_0)}{\lambda_k J_1^2(\lambda_k)}, J_0(\lambda_k) = 0, k \in \mathbb{N}$ $t^*$ and $r^*$ are scaled time and radius.

With $J_m(r)$ as the Bessel function of the first kind and $\lambda_k$ zeros of $J_0(r)$, equation (4) predicts cellular crowding at any location in time. Since equation (4) predicts a smooth radial distribution of cell population density, we defined the colony radius as the radius where cell population density reaches a constant as estimated experimentally based on Fig. 7b. To simplify the governing equation describing cell colony growth, we neglected active cell–cell interaction and cell surface adhesion, which could introduce several additional parameters to our model. Moreover, the diffusion equation assumes a continuous $C(r,t)$, an assumption that is no longer valid around the boundary where cell population distribution becomes discrete and, therefore, we applied the boundary condition in equation (3) to ensure that below a threshold, cell population density is set to 0. Both assumptions have been used in studies modelling colonial bacterial growth[72].

Based on cooperativity of an integrin–ECM interaction, we used Hill equation to model integrin–ligand interaction as

$$\theta = \frac{L^n}{K_d + L^n} \tag{5}$$

where $\theta$, $L$, $K_d$ and $n$ represent integrin activity, ligand concentration, integrin–ligand dissociation constant, and the Hill coefficient, respectively. $L$ is limited by cellular crowding such that in a high population density, ligand availability reduces due to the smaller contact between cell surface and ECM. Therefore, $L$ is inversely proportional to cellular crowding and can be expressed as

$$L = \frac{L_0}{C(r,t)} \tag{6}$$

where $L_0$ is a constant. Using equations (5) and (6), we derived integrin activity as a function of cellular crowding as

$$\theta = \frac{1}{1 + (\beta C(r,t))^n} \tag{7}$$

where $\beta$ is a constant and can be found experimentally. Furthermore, integrin activates FAK through tyrosine-phosphorylation (pFAK), which in turn upregulates ETV4 expression or in other words:

$$\theta \longrightarrow \text{pFAK} \longrightarrow \text{ETV4} \tag{8}$$

The Hill coefficient determines the degree of cooperativity and determines ETV4 transition width (that is, the sharpness of the sigmoid shape of $\theta$) such that $n \to \infty$ converges to a step function. However, the location of the transition (that is, ETV4/integrin transition diameter) is independent of $n$. At equilibrium, pFAK and ETV4 expression levels are directly correlated with $\theta$ with reaction equilibrium constant as the correlation coefficient. Normalization of pFAK and ETV4 expression was done using min–max normalization (min = 0, max = 1.2) to scale

pFAK/ETV4 expression to be compared to model prediction in Fig. 7d. Together, this mathematical model can accurately predict both cell population dynamic and the spatiotemporal dynamic of ETV4 expression. Model parameters are measured experimentally and presented in Supplementary Table 8. $\beta$ is obtained by least square fitting equation (7) into experimental measurement of cell population density and ETV4 expression profile. The estimate and the permutation error for $\beta$ are reported in Supplementary Table 8.

## Statistics and reproducibility

Statistical analyses were performed using GraphPad Prism 9.1.0 software (GraphPad Software) and ImageJ 1.53 software (Fiji). No statistical method was used to pre-determine sample size, but our sample sizes are similar to those reported in previous publications[91,92]. No data were excluded from the analyses. The experiments were not randomized, and the investigators were not blinded to allocation during experiments and outcome assessment. Data distribution was assumed to be normal, but this was not formally tested. All differences were compared using a two-tailed Student's *t*-test. Error bars represent mean ± s.d. For image quantifications, precise numbers of quantified cells or colonies are provided in the figures or legends. Exact *P* values are available in Supplementary Table 9.

## Reporting summary

Further information on research design is available in the Nature Portfolio Reporting Summary linked to this article.

## Data availability

The RNA-seq data generated from this study have been deposited in the Gene Expression Omnibus (GSE183702). Published bulk and single cell RNA-seq that were re-analysed here are available from the Gene Expression Omnibus (GSE69982 and GSE126022). Source data are provided with this study. All other data supporting the findings of this study are available from the corresponding author on reasonable request.

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

## Acknowledgements

We thank S. T. Baek at Pohang University of Science and Technology for providing a lentiviral vector expressing KRAS^G12V and W. Kim at Chungnam National University for insightful comments. This work was supported by the National Research Foundation of Korea grant funded by the Korea government (NRF-2020M3A9D8038184, NRF-2021R1A4A1031754, NRF-2022R1F1A1063619 and RS-2023-00221112 to J.J.), by the Korea Institute for Advancement of Technology and the Ministry of Trade, Industry and Energy of the Republic of Korea (P0021109 to J.J.) and BK21 FOUR.

## Author Contributions

J.J. was responsible for conceptualization. S.Y. and M.G. were responsible for methodology. S.O., Y.O., Y.C., J.Y. and S.J. were responsible for validation. M.G. and M.A.L. were responsible for formal analyses. S.Y., M.G., S.O., Y.O., Y.C., J.Y. and S.J. were responsible for investigation. S.Y., M.G. and J.J. wrote the original draft. S.Y., M.G., S.O., Y.O., Y.C., J.Y., M.A.L., S-Y.P., Y.L. and J.J. were responsible for review and editing. J.J. was responsible for supervision and funding acquisition.

## Competing interests

The authors declare no competing interests.

## Additional information

**Extended data** is available for this paper at https://doi.org/10.1038/s41556-024-01415-w.

**Correspondence and requests for materials** should be addressed to Jiwon Jang.

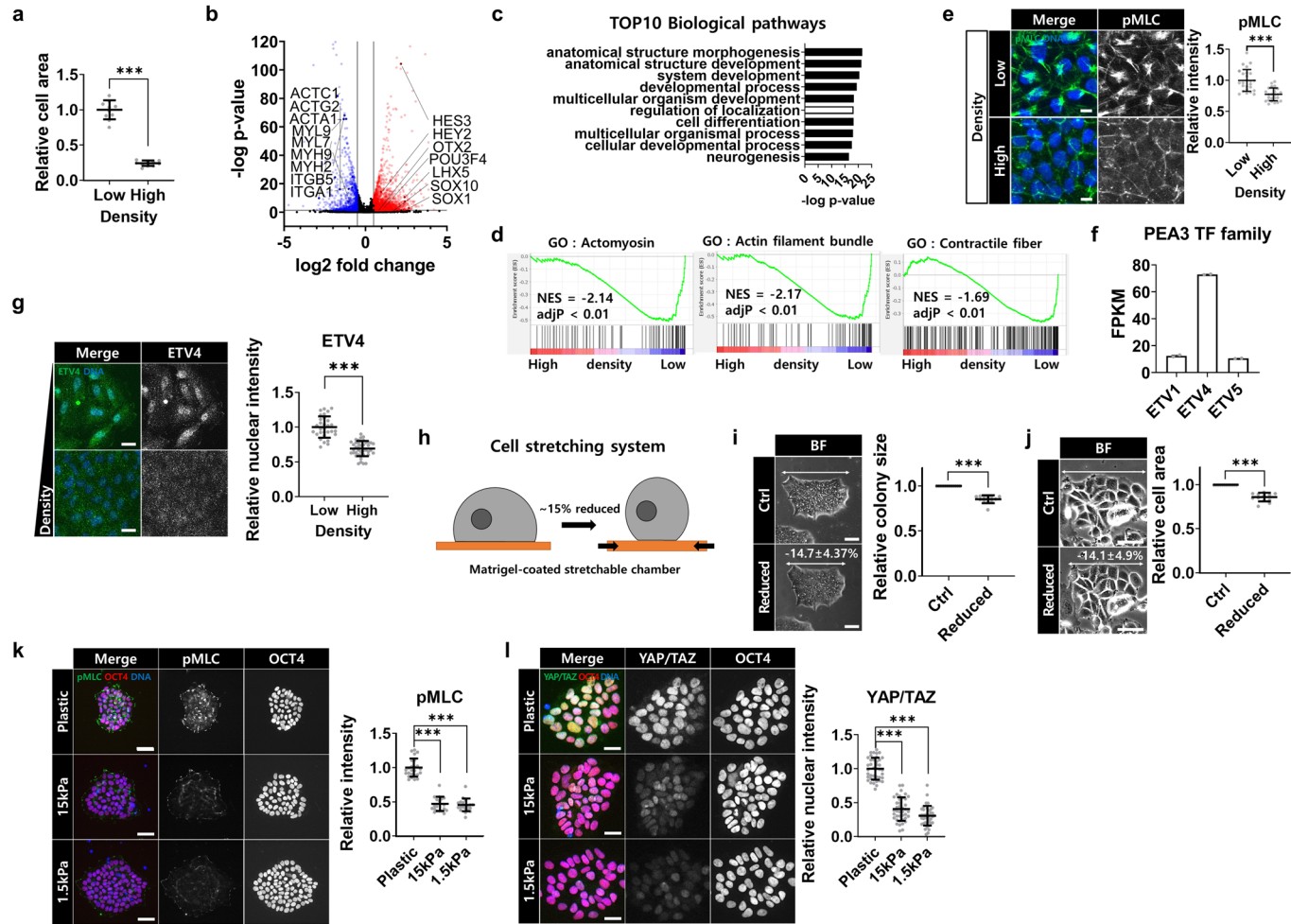

Extended Data Fig. 1 | **The expression of ETV4 is regulated by mechanical microenvironments. a**, Relative cell area of H9 hESCs in low and high density cultures. n=10 regions pooled from two independent experiments. **b**, Volcano plot showing DEGs from H9 hESCs in high density culture. The red and blue dots indicate up- and down-regulated genes, respectively, with cutoff values for DEGs: log$_2$FC < −0.5 or > 0.5, adjP < 0.05. Full list of DEGs can be found in Supplementary Table 1. **c**, The top 10 biological pathways ranked by p-values from GO analysis with total DEGs **d**, Gene set enrichment analysis (GSEA) with the RNA-seq data from H9 hESCs in low and high density cultures. GSEA was performed with 3 different gene sets: Actomyosin, Actin filament bundle, Contractile fiber from Molecular Signatures Database (MSigDB). **e**, Immunofluorescence assay for pMLC in H9 hESCs under low and high density cultures. n=25 regions. **f**, FPKM of PEA3 family transcription factors in H9 hESCs (GSE183702). n=2 samples pooled from two independent experiments. **g**, Immunofluorescence assay for ETV4 in

ARPE19 cells under low and high density cultures. n=30 cells for low density, n=41 cells for high density. **h**, Schematic representation of the cell stretching system. **i**, Colony size measurements in H9 hESCs on the cell stretching system. n=12 colonies. **j**, Cell area measurements in MCF-7 cells on the cell stretching system. n=12 colonies. **k**, Immunofluorescence assay for pMLC and OCT4 in H9 hESCs on PDMS layers with different stiffnesses. n=25 colonies for plastic, n=20 colonies for 15 kPa and 1.5 kPa pooled from two independent experiments. **l**, Immunofluorescence assay for YAP/TAZ and OCT4 in H9 hESCs on PDMS layers with different stiffnesses. n=40 cells for YAP/TAZ pooled from two independent experiments. n = number of cells (g), regions (e), or colonies (i and j) pooled from three independent experiments. Two-sided Student's t-test, ***P < 0.001, **P < 0.01, *P < 0.05. Exact P values are presented in Supplementary Table 9. Scale bars: 10 μm (e,j), 25 μm (g and l), 50 μm (i and k). Numerical source data are available in Source data.

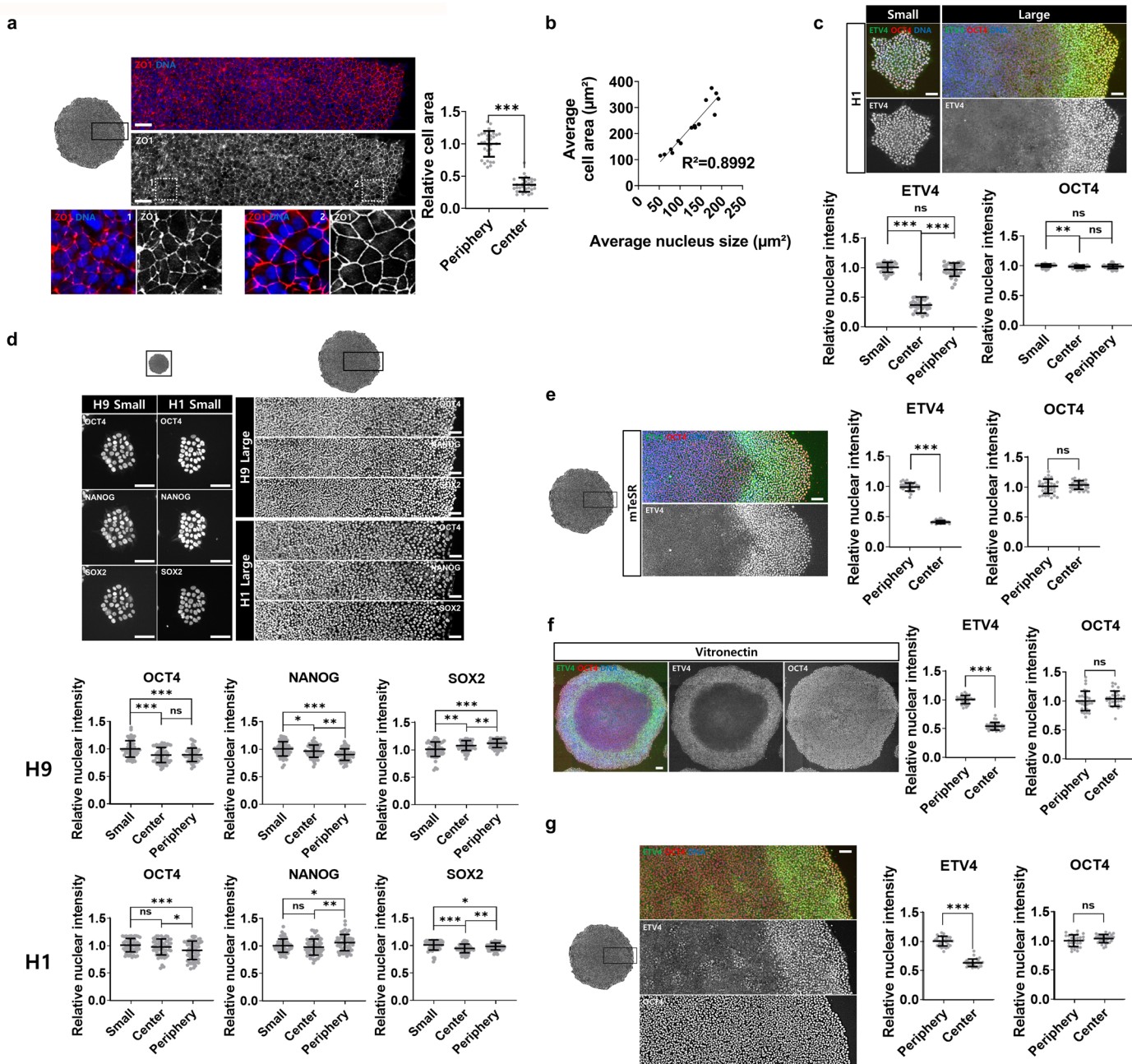

**Extended Data Fig. 2 | Cell crowding regulates ETV4 expression in a growing hESC colony. a**, Immunofluorescence assay for ZO1 in large H9 colonies. n=30 regions. **b**, Correlation of nucleus size with cell area in H9 hESCs. n=15 cells. **c,d**, Immunofluorescence assay for OCT4, NANOG, SOX2, and ETV4 in H1 and H9 colonies. n=30 cells for (c), n=60 cells for (d). **e,f**, Immunofluorescence assay for OCT4 and ETV4 in large H9 colonies in mTeSR1 on Matrigel-coated plates (e) or in TeSR-E8 on vitronectin-coated plates (f). n=30 cells. **g**, Immunofluorescence

assay for OCT4 and ETV4 in large H9 colonies derived from single cells. n=30 cells. n = number of cells (b,c,d,e,f, and g) or regions (a) pooled from three independent experiments. Two-sided Student's t-test, ***P < 0.001, **P < 0.01, *P < 0.05. Exact P values are presented in Supplementary Table 9. Scale bars: 25 μm (a), 50 μm (c,d,e,f, and g). Numerical source data are available in Source data.

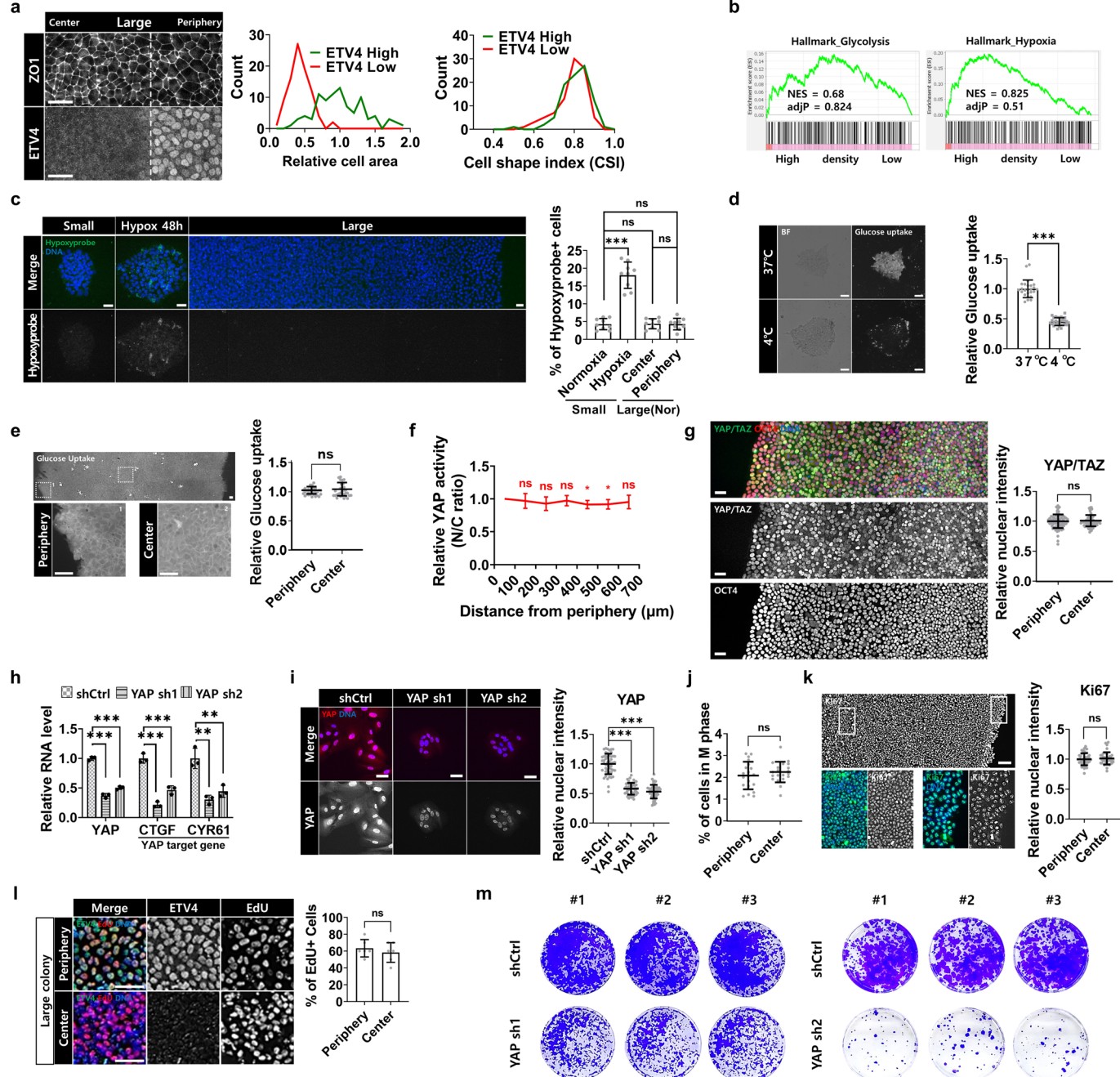

**Extended Data Fig. 3 | ETV4 and YAP exhibit distinct sensitivities to changes in cell density in hESCs. a**, The statistical analysis of cell shape index and relative cell area of ETV4-high and ETV4-low cells in large H9 colonies. n=90 cells. **b**, Gene set enrichment analysis (GSEA) with the RNA-seq data from H9 hESCs in low and high density cultures. GSEA was performed with 2 different gene sets: Hallmark_Glycolysis, Hallmark_Hypoxia from Molecular Signatures Database (MSigDB). **c**, Immunofluorescence assay for Hypoxyprobe in small and large H9 colonies. n=9 regions. **d,e**, Glucose uptake assay in small and large H9 colonies. n=25 regions for 37°C (d), n=31 regions for 4°C (d), and n=30 regions for (e). **f**, Immunofluorescence assay for YAP in large H9 colonies. YAP activity was measured by the ratio of the nuclear over cytoplasmic intensities (N/C ratio). n=5 colonies. **g**, Immunofluorescence assay for YAP/TAZ and OCT4 in large H9 colonies. n=80 cells pooled from two independent experiments. **h,i**, Validation of lentiviral vectors expressing YAP shRNAs in H9 hESCs by qPCR (h) and immunostaining (i). n=3 samples for qPCR (h) and n=50 cells for immunostaining (i). **j**, Quantification of cells in M phase in large H9 colonies. n=20 regions. **k**, Immunofluorescence assay for Ki67 in large H9 colonies. n=40 cells pooled from two independent experiments. **l**, Immunofluorescence assay for ETV4 and EdU in large H9 colonies. n=6 regions. **m**, Crystal violet staining in H9 hESCs transduced with lentiviral vectors expressing YAP shRNAs. n = number of cells (a and i), colonies (f), or regions (c,d,e,j, and l) pooled from three independent experiments. Two-sided Student's t-test, ***P < 0.001, **P < 0.01, *P < 0.05. Exact P values are presented in Supplementary Table 9. Scale bars: 25 μm (a,c,d,e,g, and i), 50 μm (l), 100 μm (k). Numerical source data are available in Source data.

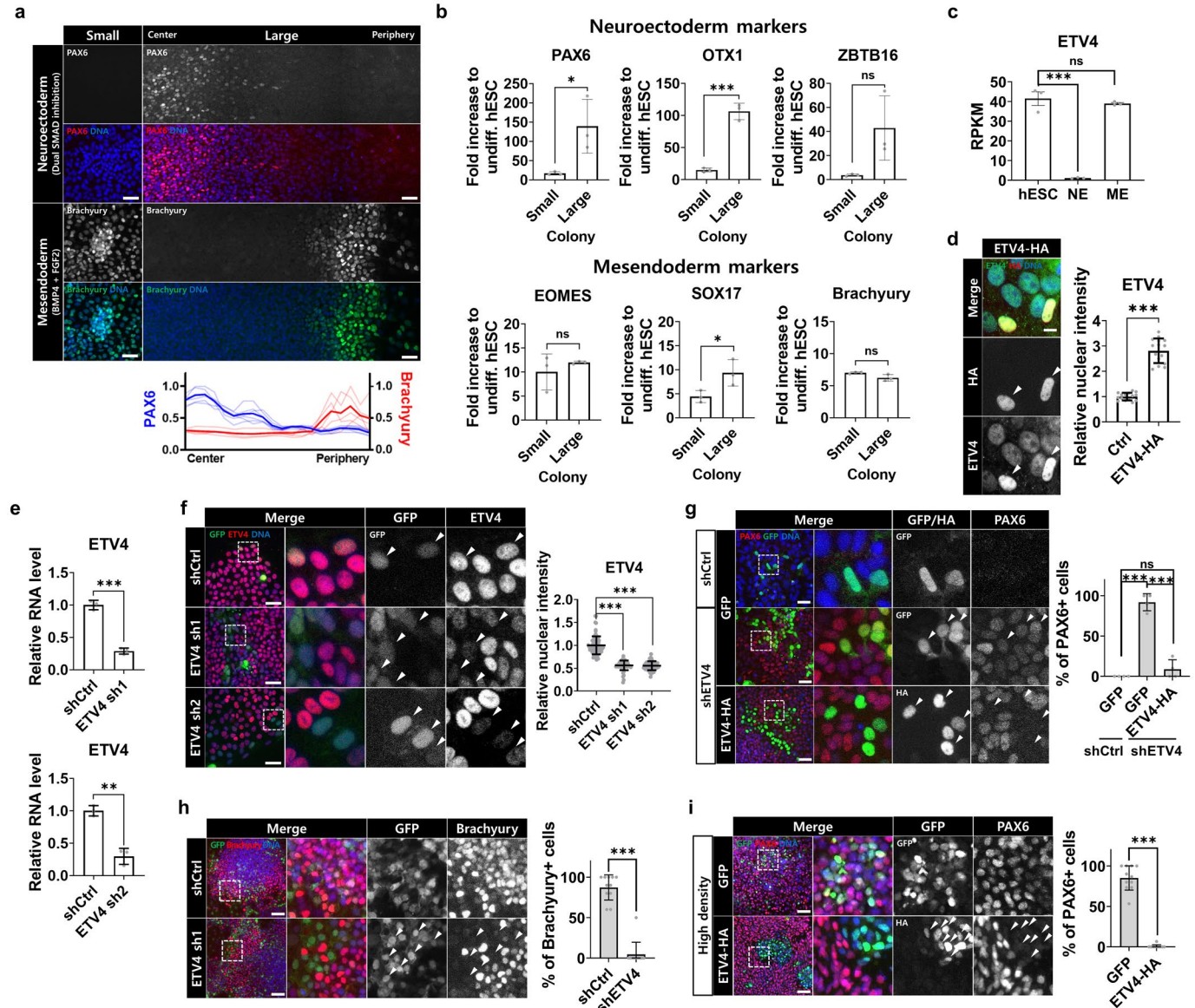

**Extended Data Fig. 4 | ETV4 links cell crowding to NE derepression.**
**a**, Immunofluorescence assay for PAX6 and Brachyury in differentiated H1 colonies. n=6 colonies. **b**, qPCR analysis for lineage markers in H9 hESCs differentiated by FGF2 and TGF-β deprivation. n=3 samples pooled from three independent experiments. **c**, RPKM of ETV4 in undifferentiated H9 hESCs, and H9-derived NE and ME cells (GSE69982). n=3 samples. **d**, Validation of lentiviral vectors expressing ETV4-HA in H9 hESCs by immunostaining. n=15 cells. **e,f**, Validation of lentiviral vectors expressing ETV4 shRNAs together with GFP in H9 hESCs by qPCR (e) and immunostaining (f). n=3 samples for qPCR (e) from three independent experiments, n=75 cells for shCtrl, n=55 cells for ETV4 sh1, and n=50 cells for ETV4 sh2 for immunostaining (f) pooled from two independent experiments. **g**, Immunofluorescence assay for PAX6 in shETV4-expressing H9

hESCs transduced with lentiviral vectors expressing ETV4-HA together with GFP and differentiated to NE cells for 5 days. n=4 regions. **h**, Immunofluorescence assay for Brachyury in H9 hESCs transduced with lentiviral vectors expressing ETV4 shRNA and differentiated to ME cells for 3 days. n=11 regions. **i**, Immunofluorescence assay for PAX6 in high density H9 hESCs transduced with lentiviral vectors expressing ETV4-HA and differentiated to NE cells for 5 days. n=11 regions. n = number of cells (d), colonies (a), or regions (g,h, and i) pooled from three independent experiments. Two-sided Student's t-test, ***P < 0.001, **P < 0.01, *P < 0.05. Exact P values are presented in Supplementary Table 9. Scale bars: 10 μm (d), 25 μm (f, and g), 50 μm (a), 100 μm (h and i). Source numerical data are available in Source data.

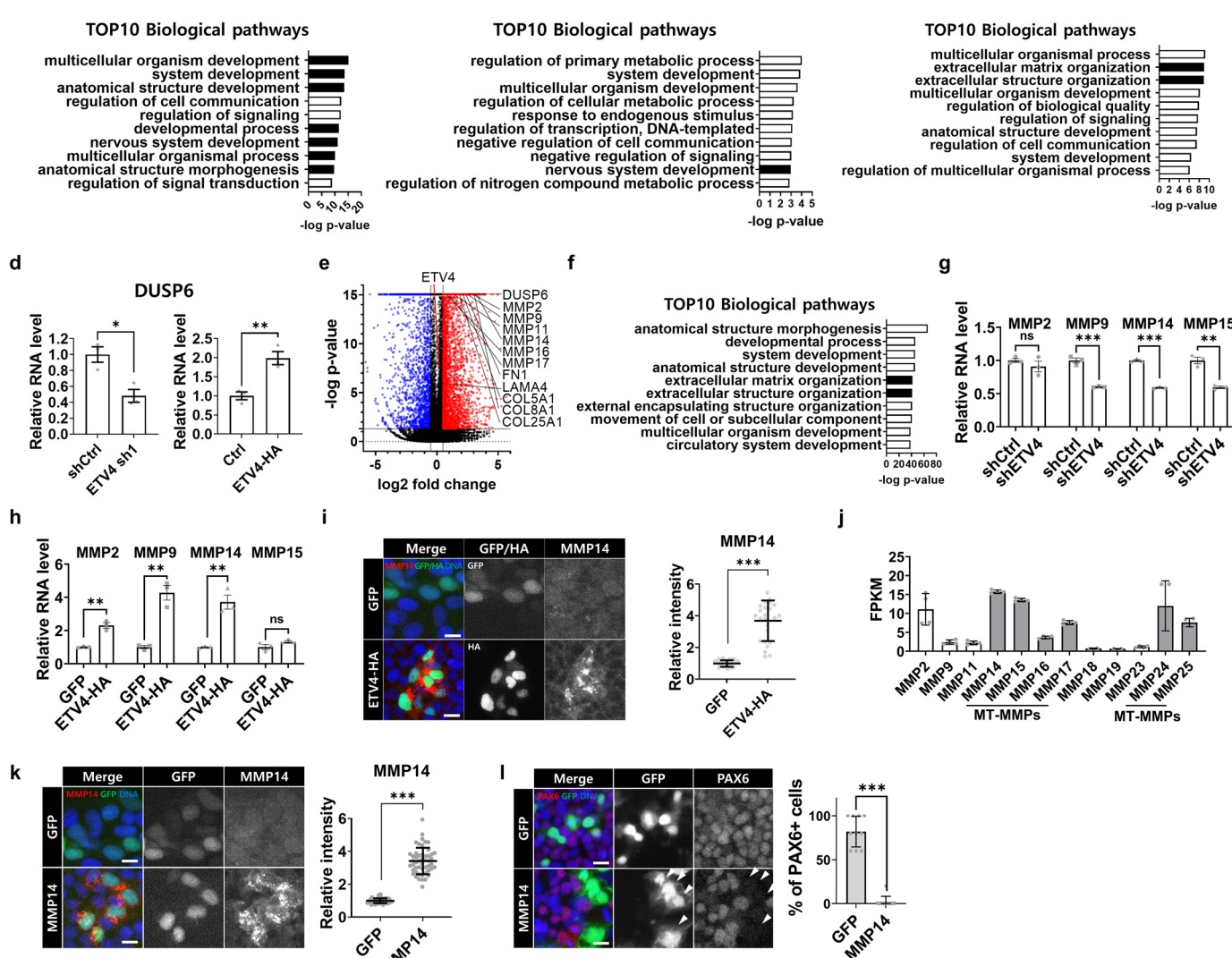

**Extended Data Fig. 5 | ETV4 inhibits NE differentiation by MMPs. a,b,c** The top 10 biological pathways ranked by p-values from GO analysis with total DEGs (a), upregulated DEGs (b), and downregulated DEGs (c) after ETV4 KD. **d**, qPCR analysis for DUSP6 in H9 hESCs transduced with lentiviral vectors expressing ETV4 shRNA or ETV4-HA. n=3 samples pooled from three independent experiments. **e**, Volcano plot showing DEGs in H1 hESCs sorted by N-cadherin expression. The red and blue dots indicate up- and down-regulated genes, respectively, with cutoff values for DEGs: $log_2FC < -0.5$ or $> 0.5$, adjP < 0.05 (GSE126022). Full list of DEGs can be found in Supplementary Table 4. **f**, The top 10 biological pathways ranked by p-values from GO analysis with upregulated DEGs in N-cadherin⁺ H1 hESCs. **g,h**, qPCR analysis for MMP2, MMP9, MMP14, and MMP15 in H9 hESCs transduced with lentiviral vectors expressing ETV4 shRNA

(g) or ETV4-HA (h). n=3 samples pooled from three independent experiments. **i**, Immunofluorescence assay for MMP14 in H9 hESCs transduced with lentiviral vectors expressing ETV4-HA. n=25 cells. **j**, FPKM of MMPs in undifferentiated H9 hESCs (GSE183702). n=4 samples pooled from four independent experiments. **k**, Immunofluorescence assay for MMP14 in H9 hESCs transduced with lentiviral vectors expressing MMP14 together with GFP. n=50 cells. **l**, Immunofluorescence assay for PAX6 in H9 hESCs transduced with lentiviral vectors expressing MMP14 together with GFP and differentiated to NE cells for 5 days. n=10 regions. n = number of cells (i and k) or regions (l) pooled from three independent experiments. Two-sided Student's t-test, ***P < 0.001, **P < 0.01, *P < 0.05. Exact P values are presented in Supplementary Table 9. Scale bars: 10 μm (i,k, and l). Numerical source data are available in Source data.

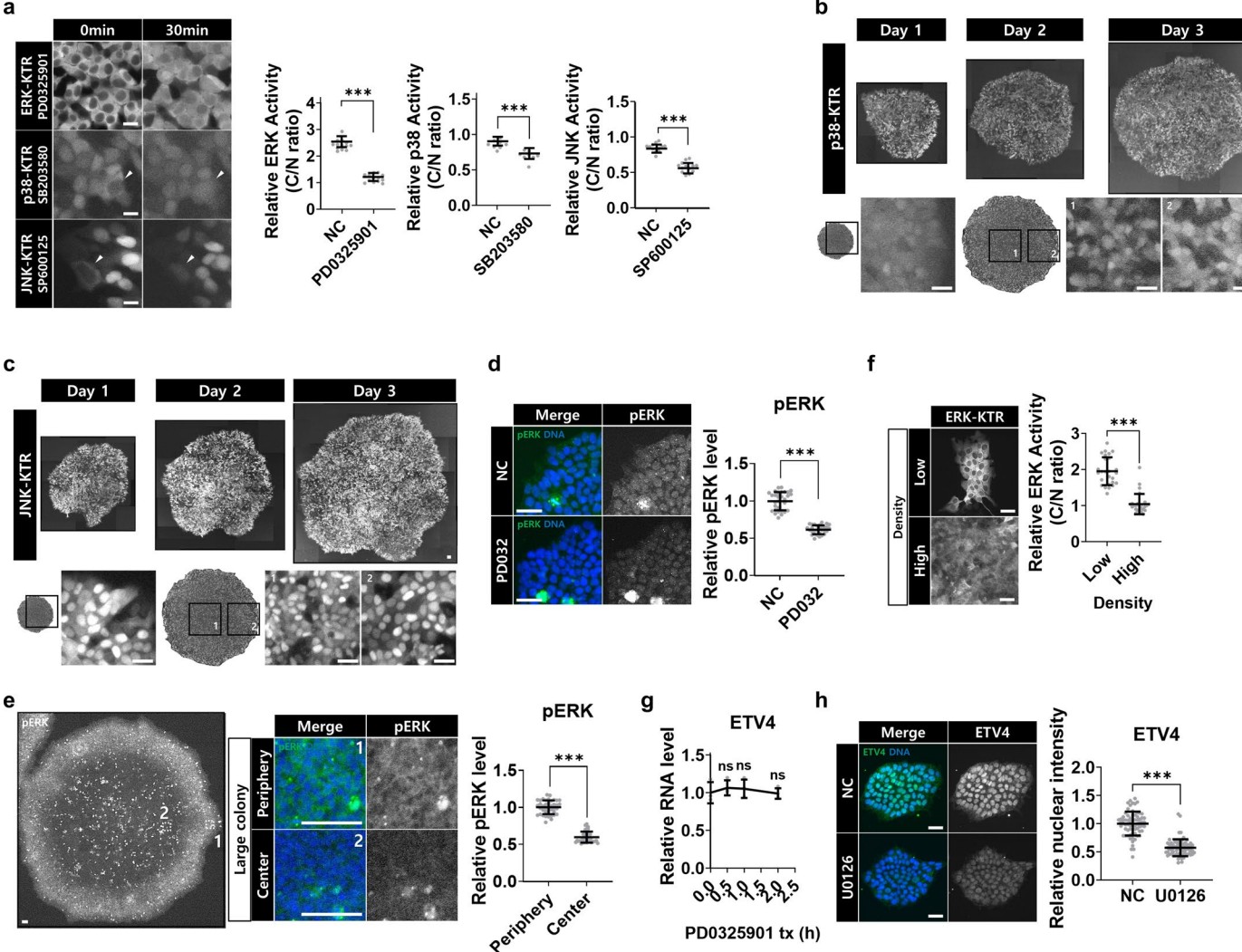

**Extended Data Fig. 6 | ERK regulates the protein stability of ETV4.**
 a, Validation of KTRs treated with specific inhibitors. n=13 cells for ERK-KTR and JNK-KTR, n=8 cells for p38-KTR pooled from two independent experiments. b,c, Representative time-course images from three independent colony-tracking assays for p38-KTR (b) and JNK-KTR (c) in H9 colonies. d, Immunofluorescence assay for pERK in H9 cells treated with PD0325901 (1 µM, 1h). n=30 cells. e, Immunofluorescence assay for pERK in large H9 colonies. Mitotic activation of ERK was seen in brightly stained cells. n=40 cells. f, ERK activity measured by the ratio of the cytoplasmic over nuclear intensities (C/N ratio) in single

H9 hESCs expressing ERK-KTR under low and high density cultures (24h after seeding). n=26 cells. g, qPCR analysis for ETV4 in H9 hESCs treated with PD0325901 (1 µM). n=3 samples pooled from three independent experiments. h, Immunofluorescence assay for ETV4 in H9 hESCs treated with U0126 (10 µM, 1h). n=70 cells pooled from two independent experiments. n = number of cells (d,e,f) pooled from three independent experiments. Two-sided Student's t-test, ***P < 0.001, **P < 0.01, *P < 0.05. Exact P values are presented in Supplementary Table 9. Scale bars: 10 µm (a), 25 µm (b,c,d,f, and h), 100 µm (e). Numerical source data and additional microscope images are available in Source data.

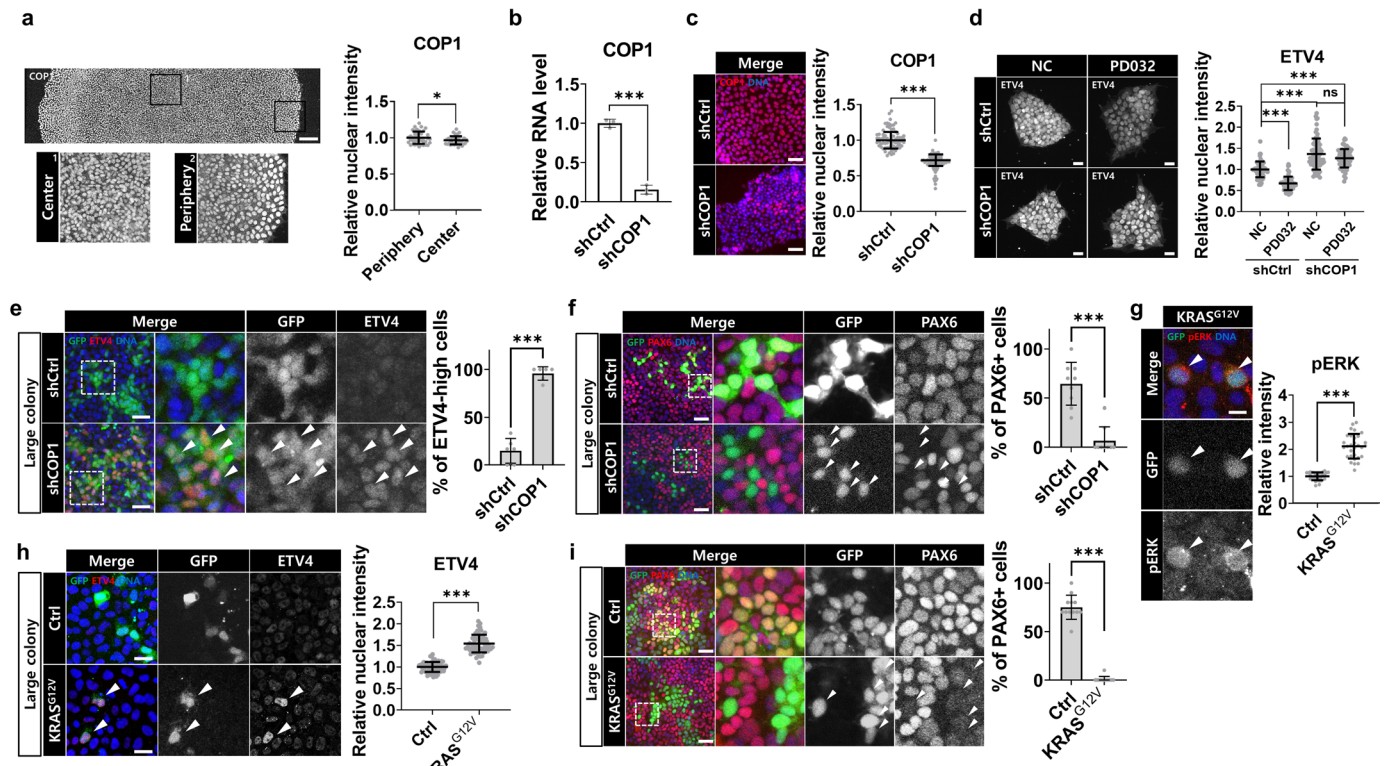

**Extended Data Fig. 7 | COP1 is involved in the regulation of ETV4 protein stability. a**, Immunofluorescence assay for COP1 in large H9 colonies. n=40 cells. **b,c**, Validation of lentiviral vectors expressing COP1 shRNA in H9 hESCs by qPCR (b) and immunostaining (c). n=3 samples for qPCR (b) from three independent experiments, n=60 cells for immunostaining (c) pooled from two independent experiments. **d**, Immunofluorescence assay for ETV4 in H9 hESCs transduced with lentiviral vectors expressing COP1 shRNA and treated with PD0325901 (1 µM, 30 min). n=64 cells. **e**, Immunofluorescence assay for ETV4 in large H9 colonies transduced with lentiviral vectors expressing COP1 shRNA together with GFP. n=6 regions pooled from two independent experiments. **f**, Immunofluorescence assay for PAX6 in H9 hESCs transduced with lentiviral vectors expressing COP1 shRNA together with GFP and differentiated to NE cells for 5 days. n=9 regions.

**g**, Immunofluorescence assay for pERK in H9 hESCs transduced with lentiviral vectors expressing KRAS[G12V] together with GFP. n=30 cells pooled from two independent experiments. **h**, Immunofluorescence assay for ETV4 in H9 hESCs transduced with lentiviral vectors expressing KRAS[G12V] together with GFP. n=63 cells pooled from two independent experiments. **i**, Immunofluorescence assay for PAX6 in H9 hESCs transduced with lentiviral vectors expressing constitutively-active KRAS[G12V] together with GFP and differentiated to NE cells for 5 days. n=12 regions. n = number of cells (a and d) or regions (f and i) pooled from three independent experiments. Two-sided Student's t-test, ***P < 0.001, **P < 0.01, *P < 0.05. Exact P values are presented in Supplementary Table 9. Scale bars: 10 µm (g), 25 µm (c,d,e, and h), 50 µm (f and i), 100 µm (a). Numerical source data are available in Source data.

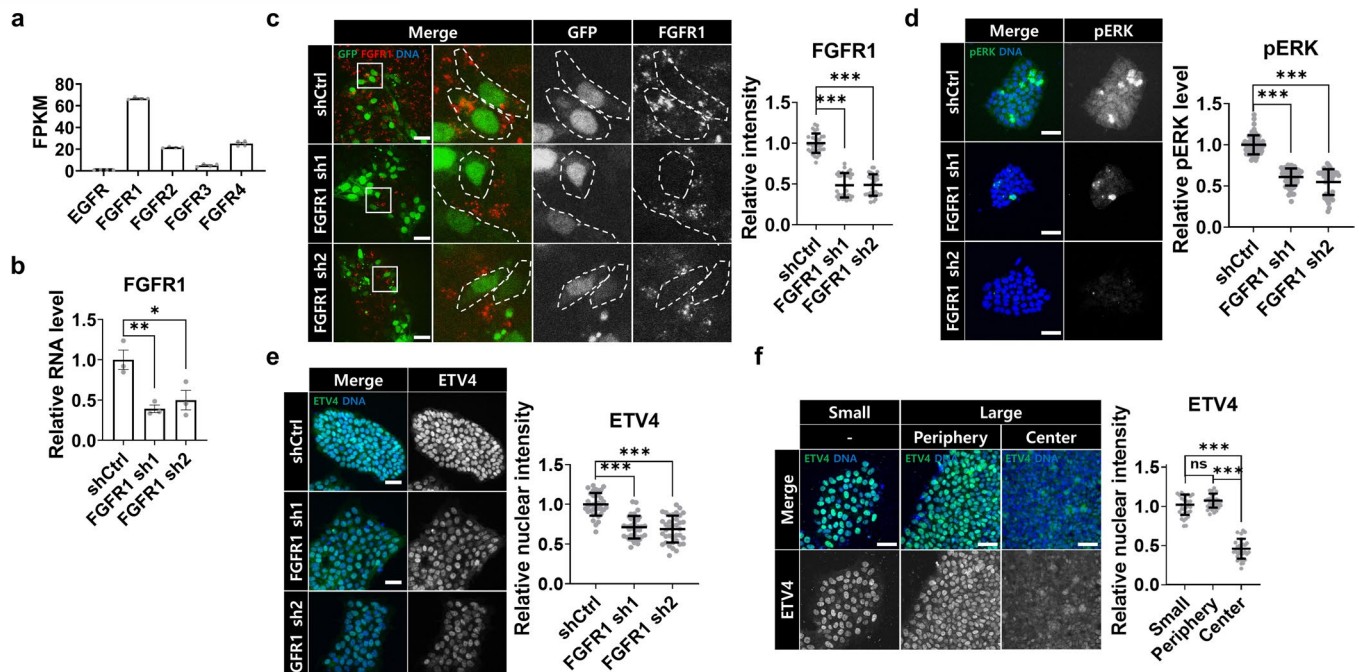

**Extended Data Fig. 8 | FGF signalling regulates ERK and ETV4. a**, FPKM of EGFR and FGFRs in undifferentiated H9 hESCs (GSE183702). n=4 samples pooled from four independent experiments. **b,c**, Validation of lentiviral vectors expressing FGFR1 shRNAs in H9 hESCs by qPCR (b) and immunostaining (c). n=3 samples for qPCR (b) and n=33 cells for immunostaining (c). **d**, Immunofluorescence assay for pERK in H9 hESCs transduced with lentiviral vectors expressing FGFR1 shRNAs together with GFP. n=50 cells pooled from two independent experiments. **e**, Immunofluorescence assay for ETV4 in H9 hESCs transduced with lentiviral vectors expressing FGFR1 shRNAs together with GFP. n=37 cells pooled from two independent experiments. **f**, Immunofluorescence assay for ETV4 in small and large H9 colonies cultured in a transwell system. n=30 cells. n = number of cells (c and f) pooled from three independent experiments. Two-sided Student's t-test, ***P < 0.001, **P < 0.01, *P < 0.05. Exact P values are presented in Supplementary Table 9. Scale bars: 25 μm (c,d,e, and f). Numerical source data are available in Source data.

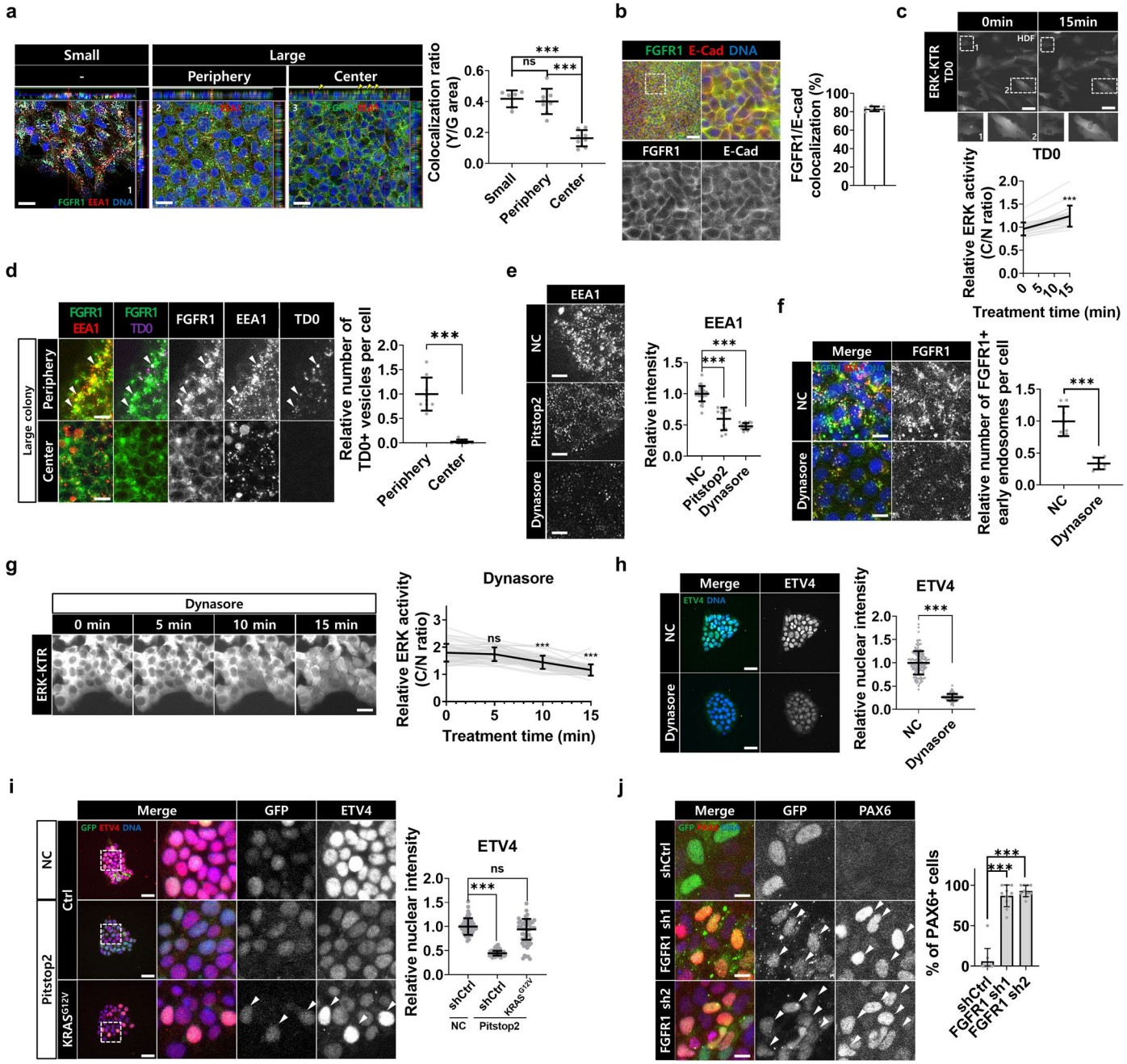

**Extended Data Fig. 9 | Cell crowding blocks FGFR endocytosis.**
**a**, Immunofluorescence assay for FGFR1 and EEA1 in small and large H9 colonies. Arrow heads indicate apical localization of FGFR1 proteins. n=6 regions for Small, n=8 regions for Large. **b**, Immunofluorescence assay for FGFR1 and E-cadherin in the centre of large H9 colonies. n=8 regions. **c**, Validation of TD0 (500 nM) in human dermal fibroblasts expressing ERK-KTR. n=22 cells pooled from two independent experiments. **d**, Immunofluorescence assay for FGFR1 and EEA1 in large H9 colonies treated with TD0 (500 nM, 1h). n=9 regions pooled from two independent experiments. **e**, Immunofluorescence assay for EEA1 in H9 hESCs treated with Pitstop2 (50 μM, 1h) or Dynasore (100 μM, 1h). n=28 regions for NC, n=12 regions for Pitstop2, n=16 regions for Dynasore pooled from two independent experiments. **f**, Immunofluorescence assay for FGFR1 and EEA1 in H9 hESCs treated with Dynasore (100 μM, 1h). n=8 regions pooled from two independent experiments. **g**, Time-course

images for ERK-KTR in H9 hESCs treated with Dynasore (100 μM). ERK activity measured by the ratio of the cytoplasmic over nuclear intensities (C/N ratio). n=46 cells. **h**, Immunofluorescence assay for ETV4 in H9 hESCs treated with Dynasore (100 μM, 1h). n=140 cells pooled from two independent experiments. **i**, Immunofluorescence assay for ETV4 in H9 hESCs transduced with lentiviral vectors expressing KRAS[G12V] together with GFP and treated with Pitstop2 (50 μM, 30 min). n=45 cells. **j**, Immunofluorescence assay for PAX6 in H9 hESCs transduced with lentiviral vectors expressing FGFR1 shRNAs together with GFP and differentiated to NE cells for 5 days. n=10 regions. n = number of cells (g and i) or regions (a,b, and j) pooled from three independent experiments. Two-sided Student's t-test, ***P < 0.001, **P < 0.01, *P < 0.05. Exact P values are presented in Supplementary Table 9. Scale bars: 10 μm (e,f, and j), 15 μm (a), 25 μm (d,g,h, and i), 50 μm (b), 100 μm (c). Numerical source data are available in Source data.

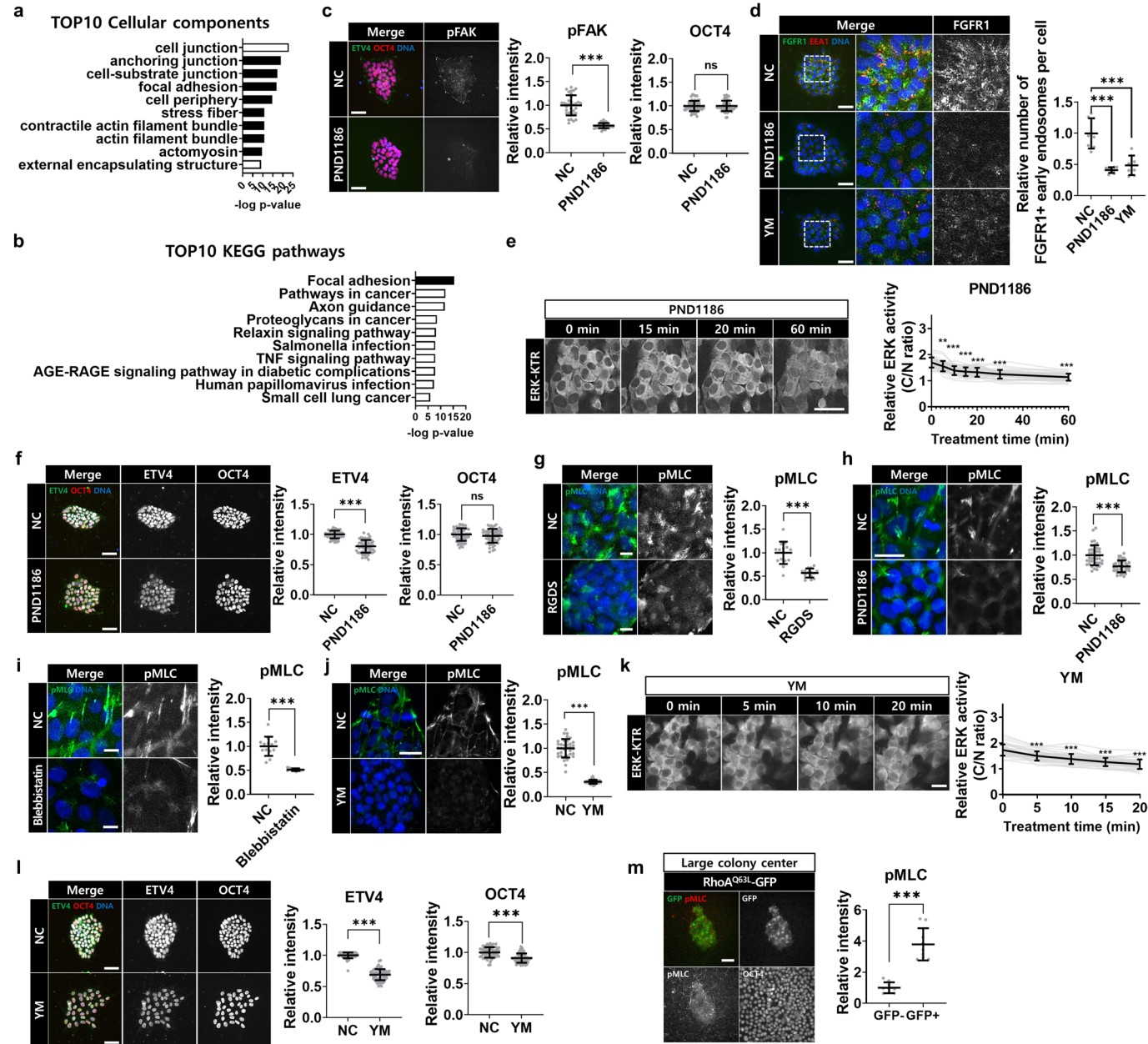

**Extended Data Fig. 10 | Cell crowding inhibits the integrin-actomyosin pathway. a**, The top 10 cellular components ranked by p-values from GO analysis with downregulated DEGs in high density H9 hESCs. **b**, The top 10 KEGG pathways ranked by p-values from GO analysis with upregulated DEGs in N-cadherin⁺ H1 hESCs. **c**, Immunofluorescence assay for pFAK and OCT4 in H9 hESCs treated with PND1186 (2 μM, 1h). n=35 cells for pFAK, n=60 cells for OCT4 pooled from two independent experiments. **d**, Immunofluorescence assay for FGFR1 and EEA1 in H9 hESCs treated with PND1186 (2 μM, 1h) or YM (10 μM, 1h). n=8 regions for NC and YM, n=9 regions for PND1186. **e**, Time-course images for ERK-KTR in H9 hESCs treated with PND1186 (2 μM). n=40 cells. **f**, Immunofluorescence assay for ETV4 and OCT4 in H9 hESCs treated with PND1186 (2 μM, 1h). n=60 cells. **g**, Immunofluorescence assay for pMLC in H9 hESCs treated with RGDS (500 μM, 24h). n=15 regions pooled from two independent experiments. **h**, Immunofluorescence assay for pMLC in H9 hESCs treated with PND1186 (2 μM, 1h). n=40 regions pooled from two independent experiments.

**i**, Immunofluorescence assay for pMLC in H9 hESCs treated with Blebbistatin (50 μM, 30 min). n=13 regions for NC and n=10 regions for Blebbistatin pooled from two independent experiments. **j**, Immunofluorescence assay for pMLC in H9 hESCs treated with YM (10 μM, 1h). n=35 regions pooled from two independent experiments. **k**, Time-course images for ERK-KTR in H9 hESCs treated with YM (10 μM). n=39 cells. **l**, Immunofluorescence assay for ETV4 and OCT4 in H9 hESCs treated with YM (10 μM, 1h). n=60 cells. **m**, Immunofluorescence assay for pMLC and OCT4 in the centre of large H9 colonies transduced with lentiviral vectors expressing constitutively-active RhoA^Q63L-GFP in a doxycycline-dependent manner. n=9 regions. n = number of cells (e,f,k, and l) or regions (d and m) pooled from three independent experiments. Two-sided Student's t-test, ***P < 0.001, **P < 0.01, *P < 0.05. Exact P values are presented in Supplementary Table 9. Scale bars: 10 μm (g, and i), 20 μm (h and j), 25 μm (c,d,e,f,k, and l), 50 μm (m). Numerical source data are available in Source data.

# Reporting Summary

## Statistics

For all statistical analyses, confirm that the following items are present in the figure legend, table legend, main text, or Methods section.

| n/a | Confirmed | |
|---|---|---|
| ☐ | ☒ | The exact sample size (*n*) for each experimental group/condition, given as a discrete number and unit of measurement |
| ☐ | ☒ | A statement on whether measurements were taken from distinct samples or whether the same sample was measured repeatedly |
| ☐ | ☒ | The statistical test(s) used AND whether they are one- or two-sided *Only common tests should be described solely by name; describe more complex techniques in the Methods section.* |
| ☒ | ☐ | A description of all covariates tested |
| ☒ | ☐ | A description of any assumptions or corrections, such as tests of normality and adjustment for multiple comparisons |
| ☐ | ☒ | A full description of the statistical parameters including central tendency (e.g. means) or other basic estimates (e.g. regression coefficient) AND variation (e.g. standard deviation) or associated estimates of uncertainty (e.g. confidence intervals) |
| ☐ | ☒ | For null hypothesis testing, the test statistic (e.g. *F*, *t*, *r*) with confidence intervals, effect sizes, degrees of freedom and *P* value noted *Give P values as exact values whenever suitable.* |
| ☒ | ☐ | For Bayesian analysis, information on the choice of priors and Markov chain Monte Carlo settings |
| ☒ | ☐ | For hierarchical and complex designs, identification of the appropriate level for tests and full reporting of outcomes |
| ☒ | ☐ | Estimates of effect sizes (e.g. Cohen's *d*, Pearson's *r*), indicating how they were calculated |

*Our web collection on statistics for biologists contains articles on many of the points above.*

## Software and code

Policy information about availability of computer code

| Data collection | Confocal images were taken by ZEISS LSM800 confocal microscope (ZEISS). Realtime PCR result were collected by CFX Connect Real-Time PCR Detection System (BIO-RAD). Western blot results were visualized with Amersham imager 680 (Amersham). |
|---|---|
| Data analysis | Sequencing data sets were processed and analyzed using the following tools: HISAT v2.1.0 StringTie v2.1.3b gProfiler ve109_eg56_p17_773ec798 GSEA v4.3.2 R v3.6.3 R package DESeq2 v1.38.3 R package DEsingle v1.18.1 Images were processed and analyzed using the following tools: ImageJ v1.53 Statistical analysis was performed using the following tools: GraphPad Prism v9.1.0 gel analysis was performed using the following tools: Multiguage v3.0 |

For manuscripts utilizing custom algorithms or software that are central to the research but not yet described in published literature, software must be made available to editors and reviewers. We strongly encourage code deposition in a community repository (e.g. GitHub). See the Nature Portfolio guidelines for submitting code & software for further information.

## Data

Policy information about availability of data

All manuscripts must include a data availability statement. This statement should provide the following information, where applicable:

- Accession codes, unique identifiers, or web links for publicly available datasets
- A description of any restrictions on data availability
- For clinical datasets or third party data, please ensure that the statement adheres to our policy

Data are available in the main text, supplementary materials, and Gene Expression Omnibus (GSE183702)

## Human research participants

Policy information about studies involving human research participants and Sex and Gender in Research.

| | |
|---|---|
| Reporting on sex and gender | H1 (male) and H9 (female) hESCs cell lines were used to cover all sex in this study. |
| Population characteristics | N/A |
| Recruitment | N/A |
| Ethics oversight | This work was approved by the Human Stem Cell Research Oversight Committee at Pohang University of Science and Technology (PIRB-2021-R035) |

Note that full information on the approval of the study protocol must also be provided in the manuscript.

# Field-specific reporting

Please select the one below that is the best fit for your research. If you are not sure, read the appropriate sections before making your selection.

☒ Life sciences          ☐ Behavioural & social sciences          ☐ Ecological, evolutionary & environmental sciences

For a reference copy of the document with all sections, see nature.com/documents/nr-reporting-summary-flat.pdf

# Life sciences study design

All studies must disclose on these points even when the disclosure is negative.

| | |
|---|---|
| Sample size | Preliminary experiments were performed when possible to determine requirements for sample size. Sample size sufficiency was determined by preliminary data or discussion. For statistical significance, the sample size was always independently performed three or more times(except for a few supplementary data with two independent experiments). |
| Data exclusions | No data were excluded from the analysis. |
| Replication | All experiments were replicated or performed independently for at least three times(except for a few supplementary data with two independent experiments). |
| Randomization | Randomization is not applicable to our study since all experiments were conducted on cultured cells. Any variations observed between treatment groups are not attributed to sampling bias. |
| Blinding | In general, all investigators were blind when they execute and gain data. During the experiments the investigators needed to know the media composition to maintain or induce differentiation of embryonic/pluripotent stem cells. Data analysis steps were blinded when available. |

# Reporting for specific materials, systems and methods

We require information from authors about some types of materials, experimental systems and methods used in many studies. Here, indicate whether each material, system or method listed is relevant to your study. If you are not sure if a list item applies to your research, read the appropriate section before selecting a response.

## Materials & experimental systems

| n/a | Involved in the study |
|---|---|
| ☐ | ☒ Antibodies |
| ☐ | ☒ Eukaryotic cell lines |
| ☒ | ☐ Palaeontology and archaeology |
| ☒ | ☐ Animals and other organisms |
| ☒ | ☐ Clinical data |
| ☒ | ☐ Dual use research of concern |

## Methods

| n/a | Involved in the study |
|---|---|
| ☒ | ☐ ChIP-seq |
| ☒ | ☐ Flow cytometry |
| ☒ | ☐ MRI-based neuroimaging |

## Antibodies

| Antibodies used | Primary antibodies:<br>Mouse anti-Pax-6, Santa Cruz, SC-81649 (PAX6, monoclonal)<br>Mouse anti-Oct-3/4 (C-10), Santa Cruz, SC-5279 (C10, monoclonal)<br>Rabbit anti-SOX2, Millipore, AB5603 (Polyclonal)<br>Rat anti-beta1 integrin (CD29), BD, 553715 (9EG7, monoclonal)<br>Rabbit anti-ETV1, NOVUSBIO, NBP2-57731 (Polyclonal)<br>Rabbit anti-ETV4, Proteintech, 10684-1-AP (Polyclonal)<br>Rabbit anti-ETV5, Proteintech, 13011-1-AP (Polyclonal)<br>Rabbit anti-Ki-67 (D3B5), Cell Signaling Technology, 9129S (D3B5, monoclonal)<br>Rabbit anti-pFAK (D20B1), Cell Signaling Technology, 8556S (D20B1, monoclonal)<br>Rabbit anti-pERK [P-p44/42 MAPK(T202/Y204)], Cell Signaling Technology, 4370S (D13.14.4E, monoclonal)<br>Rabbit anti-pAkt, Cell Signaling Technology, 9271T (Polyclonal)<br>Rabbit anti-FGFR1 (D8E4), Cell Signaling Technology, 9740S (D8E4, monoclonal)<br>Rabbit anti-HA-Tag (C29F4), Cell Signaling Technology, 3724S (C29F4, monoclonal)<br>Rat anti-HA-Tag (3F10), Roche, 12158167001 (3F10, monoclonal)<br>Rabbit anti-pMLC, Cell Signaling Technology, 3674S (Polyclonal)<br>Goat anti-Nanog, R&D, AF1997 (Polyclonal)<br>Goat anti-Brachyury, R&D, AF2085 (Polyclonal)<br>Mouse anti-EEA1, BD, 610456 (14, monoclonal)<br>Rabbit anti-COP1, BETHYL, A300-894A (Polyclonal)<br>Rabbit anti-paxillin, NOVUSBIO, NBP2-57097 (Polyclonal)<br>Mouse anti-ZO1, ThermoFisher Scientific, 33-9100 (1A12, monoclonal)<br>Mouse anti-E-Cad, Cell Signaling Technology, 14472S (4A2, monoclonal)<br>Rabbit anti-MMP14, NOVUSBIO, NBP2-67415 (3-F7, monoclonal)<br>Rabbit anti-GFP, ThermoFisher Scientific, A11122 (Polyclonal)<br>Mouse anti-YAP, Santa Cruz, SC-101199 (63.7, monoclonal)<br>Rabbit anti-YAP/TAZ, Cell Signaling Technology, 8418S (D24E4, monoclonal)<br>mouse anti-β-actin, Santa Cruz, SC-47778 (C4, monoclonal)<br><br>Secondary antibodies:<br>anti-Rabbit IgG (H+L) Secondary Antibody, HRP, ThermoFisher Scientific, 31460<br>anti-Mouse IgG (H+L) Secondary Antibody, HRP, ThermoFisher Scientific, 31430<br>anti-Goat IgG (H+L) Cross-Adsorbed Secondary Antibody, Alexa Fluor 488, ThermoFisher Scientific, A-11055<br>anti-Mouse IgG (H+L) Highly Cross-Adsorbed Secondary Antibody, Alexa Fluor Plus 555, ThermoFisher Scientific, A-32773<br>anti-Mouse IgG (H+L) Highly Cross-Adsorbed Secondary Antibody, Alexa Fluor 647, ThermoFisher Scientific, A-31571<br>anti-Rabbit IgG (H+L) Highly Cross-Adsorbed Secondary Antibody, Alexa Fluor 488, ThermoFisher Scientific, A-21206<br>anti-Rabbit IgG (H+L) Highly Cross-Adsorbed Secondary Antibody, Alexa Fluor 555, ThermoFisher Scientific, A-31572<br>anti-Rat IgG (H+L) Cross-Adsorbed Secondary Antibody, Alexa Fluor 555, ThermoFisher Scientific, A-21434 |
|---|---|
| Validation | - Mouse anti-Pax-6, Santa Cruz, SC-81649: The antibody guarantee covers the use of the antibody for WB, IP, IHC and IF applications. Species reactivity: Human, Mouse, Rat, Avian<br><br>- Mouse anti-Oct-3/4 (C-10), Santa Cruz, SC-5279: hESCs differentiation led to a reduced fluorescence signal shown by immunofluorescent staining (Data not included). The antibody guarantee covers the use of the antibody for WB and IF applications. No cross-reactivity may occur with Oct-3/4 isoform B. Species reactivity: Mouse, Rat and Human<br><br>- Rabbit anti-SOX2, Millipore, AB5603: hESCs differentiation led to a reduced fluorescence signal shown by immunofluorescent staining (Data not included). The antibody guarantee covers the use of the antibody for WB and IF applications. Species reactivity: Human, Mouse.<br><br>- Rat anti-beta1 integrin (CD29), BD, 553715: The antibody guarantee covers the use of the antibody for WB, IP, IHC and IF applications. Species reactivity: Mouse<br><br>- Rabbit anti-ETV1, NOVUSBIO, NBP2-57731: The antibody guarantee covers the use of the antibody for ICC and IF applications. Species reactivity: Mouse, Rat<br><br>- Rabbit anti-ETV4, Proteintech, 10684-1-AP: ETV4 Knockdown using two independent shRNAs led to a reduced fluorescence signal shown by immunofluorescent staining (Extended Data Fig. 4f). The antibody guarantee covers the use of the antibody for WB and IF applications. Species reactivity: Human, Mouse, Rat |

- Rabbit anti-ETV5, Proteintech, 13011-1-AP: The antibody guarantee covers the use of the antibody for WB and IF applications. Species reactivity: Human, Mouse

- Rabbit anti-Ki-67 (D3B5), Cell Signaling Technology, 9129S: The antibody guarantee covers the use of the antibody for ICC and Flow cytometry applications. Species reactivity: Human, Mouse, Rat

- Rabbit anti-pFAK (D20B1), Cell Signaling Technology, 8556S: FAK inhibition using chemical(PND1186) led to a reduced fluorescence signal shown by immunofluorescent staining (Extended Data Fig. 9c).The antibody guarantee covers the use of the antibody for WB and IP applications. Species reactivity: Human

- Rabbit anti-pERK [P-p44/42 MAPK(T202/Y204)], Cell Signaling Technology, 4370S: The antibody guarantee covers the use of the antibody for WB, IP, IHC, IF, and Flow cytometry applications. Species reactivity: Human, Mouse, Rat, Hamster, Monkey, Mink, D. melanogaster, Zebrafish, Bovine, Dog, Pig, S. cerevisiae

- Rabbit anti-pAkt, Cell Signaling Technology, 9271T: The antibody guarantee covers the use of the antibody for WB, IP, IHC, IF, and Flow cytometry applications. Species reactivity: Human, Mouse, Rat, Hamster, Monkey, D. melanogaster, Bovine, Dog

- Rabbit anti-FGFR1 (D8E4), Cell Signaling Technology, 9740S: FGFR1 Knockdown using two independent shRNAs led to a reduced fluorescence signal shown by immunofluorescent staining (Extended Data Fig. 8c). The antibody guarantee covers the use of the antibody for WB, IP, IHC, IF, and Flow cytometry applications. Species reactivity: Human, Mouse, Rat, Monkey

- Rabbit anti-HA-Tag (C29F4), Cell Signaling Technology, 3724S: The antibody guarantee covers the use of the antibody for WB, IP, IHC, IF, flow cytometry and ChIP applications. Species reactivity: All Species Expected

- Rat anti-HA-Tag (3F10), Roche, 12158167001: The antibody was validated by staining H9 hESCs expressing HA ETV4(Extended Data Fig. 4d). The antibody guarantee covers the use of the antibody for WB, IF and ELISA applications. Species reactivity: Human

- Rabbit anti-pMLC, Cell Signaling Technology, 3674S: The antibody guarantee covers the use of the antibody for WB applications. Species reactivity: Human, Mouse

- Goat anti-Nanog, R&D, AF1997: hESCs differetiation led to a reduced fluorescence signal shown by immunofluorescent staining (Data not included). The antibody guarantee covers the use of the antibody for WB and IF applications. Species reactivity: Human

- Goat anti-Brachyury, R&D, AF2085: The antibody guarantee covers the use of the antibody for WB, IHC, IF and ChIP applications. Species reactivity: Human, Mouse

- Mouse anti-EEA1, BD, 610456: The antibody guarantee covers the use of the antibody for WB, IHC, IF and IP applications. Species reactivity: Human, Rat, Chicken, Dog

- Rabbit anti-COP1, BETHYL, A300-894A: COP1 Knockdown using shRNA led to a reduced fluorescence signal shown by immunofluorescent staining (Extended Data Fig. 7c). The antibody guarantee covers the use of the antibody for WB and IP applications. Species reactivity: Human, Mouse

- Rabbit anti-paxillin, NOVUSBIO, NBP2-57097: The antibody guarantee covers the use of the antibody for WB, ICC and IF applications. Species reactivity: Mouse, Rat

-Mouse anti-ZO1, ThermoFisher Scientific, 33-9100 (monoclonal): The antibody guarantee covers the use of the antibody for WB, IHC, IF, ICC, Flow, ELISA and IP applications. Species reactivity: Human, Mouse, Dog, Rhesus monkey

- Mouse anti-E-Cad, Cell Signaling Technology, 14472S: The antibody guarantee covers the use of the antibody for WB, IP, IHC, IF and Flow cytometry applications. Species reactivity: Human, Mouse, Rat

- Rabbit anti-MMP14, NOVUSBIO, NBP2-67415: The antibody was validated by staining H9 hESCs expressing MMP14 (Extended Data Fig. 5k). The antibody guarantee covers the use of the antibody for WB, IF, IHC and IP applications. Species reactivity: Human, Mouse, Rat

- Rabbit anti-GFP, ThermoFisher Scientific, A11122: The antibody guarantee covers the use of the antibody for WB, IHC, IP and ChIP applications. Species reactivity: Tag

- Mouse anti-YAP, Santa Cruz, SC-101199: COP1 Knockdown using shRNA led to a reduced fluorescence signal shown by immunofluorescent staining (Extended Data Fig. 3i). The antibody guarantee covers the use of the antibody for WB, IP, IF, IHC and ELISA applications. Species reactivity: Human, Mouse, Rat

- Rabbit anti-YAP/TAZ, Cell Signaling Technology, 8418S: The antibody guarantee covers the use of the antibody for WB and IP applications. Species reactivity: Human, Mouse, Monkey

- mouse anti-β-actin, Santa Cruz, SC-47778: The antibody guarantee covers the use of the antibody for WB, IP, IF and ELISA. Species reactivity: mouse, rat, human, avian, bovine, canine, porcine, rabbit, Dictyostelium discoideum and Physarum polycephalum. Cross-reactivity may occur with all six known isoforms of Actin in higher vertebrates (including cytoplasmic β- and γ- Actin isoforms, skeletal, cardiac, and vascular α-Actin isoforms, and enteric γ-Actin isoform).

# Eukaryotic cell lines

| | |
|---|---|
| Cell line source(s) | H1 and H9 hESCs were purchased from WiCell.<br><br>HEK293T, ARPE19 and MCF-7 were purchased from ATCC. |
| Authentication | Cell lines were authenticated by short tandem repeat analysis and/or in vitro differentiation. |
| Mycoplasma contamination | Mycoplasma contamination was routinely checked and negative results were obtained. |
| Commonly misidentified lines<br>(See ICLAC register) | No commonly misidentified cell lines were used in the study. |

