## [Peer Review File · Nature Cell Biology]

Peer Review Information

Journal: Nature Cell Biology

Manuscript Title: ETV4 is a mechanical transducer linking cell crowding dynamics to lineage specification

Corresponding author name(s): Professor Jiwon Jang

Editorial Notes:

Reviewer Comments & Decisions:

Decision Letter, initial version:

*Please delete the link to your author homepage if you wish to forward this email to co-authors.

Dear Professor Jang,

Your manuscript, "ETV4 is a mechanical transducer linking cell crowding dynamics to lineage specification", has now been seen by 3 referees, who are experts in mechanobiology (referee 1); pluripotency and differentiation (referee 2); and mechanics and theoretical modelling (referee 3). As you will see from their comments (attached below) they find this work of potential interest, but have raised substantial concerns, which in our view would need to be addressed with considerable revisions before we can consider publication in Nature Cell Biology.

Nature Cell Biology editors discuss the referee reports in detail within the editorial team, including the chief editor, to identify key referee points that should be addressed with priority, and requests that are overruled as being beyond the scope of the current study. To guide the scope of the revisions, I have listed these points below. We are committed to providing a fair and constructive peer-review process, so please feel free to contact me if you would like to discuss any of the referee comments further.

I should stress that the referees' concerns point to unclear mechanoresponsive and mechanotransduction activity of ETV4, unclear mechanistic links, and potential confounding effects of cell crowding beyond mechanics which would need to be addressed with experiments and data, and reconsideration of the study for this journal and re-engagement of referees would depend on strength of these revisions.

In particular, it would be essential to:

- A) Experimentally address concerns about unclear effects of cell crowding on the ETV4 as a mechanotransducer as opposed to potential effects of integrin localization (reviewers #1 and #2). Please note that although Reviewer #1 suggests adding to discussion, we would require new experimental data to address these concerns.
- B) Experimentally address concerns about unclear effects of cell crowding on other factors like nutrients and signalling (Reviewer #2) or on cell geometry (Reviewers #2 and #3),
- C) Address concerns about unclear effects on pluripotency (Reviewer #2) with further experiments.
- D) Add experiments to further test the modelling predictions (Reviewer #3) as this would strengthen the study.
- E) Clarify methods and modelling (all Reviewers)
- E) All other referee concerns pertaining to strengthening existing data, providing controls, methodological details, clarifications and textual changes, should also be addressed.
- F) Finally please pay close attention to our guidelines on statistical and methodological reporting (listed below) as failure to do so may delay the reconsideration of the revised manuscript. In particular please provide:
 - a Supplementary Figure including unprocessed images of all gels/blots in the form of a multi-page pdf file. Please ensure that blots/gels are labeled and the sections presented in the figures are clearly indicated.
 - a Supplementary Table including all numerical source data in Excel format, with data for different figures provided as different sheets within a single Excel file. The file should include source data giving rise to graphical representations and statistical descriptions in the paper and for all instances where the figures present representative experiments of multiple independent repeats, the source data of all repeats should be provided.

We would be happy to consider a revised manuscript that would satisfactorily address these points, unless a similar paper is published elsewhere, or is accepted for publication in Nature Cell Biology in the meantime.

- ensure that it conforms to our format instructions and publication policies (see below and www.nature.com/nature/authors/).
- provide a point-by-point rebuttal to the full referee reports verbatim, as provided at the end of this letter.
- provide the completed Editorial Policy Checklist (found here <https://www.nature.com/authors/policies/Policy.pdf>), and Reporting Summary (found here <https://www.nature.com/authors/policies/ReportingSummary.pdf>). This is essential for reconsideration of the manuscript and these documents will be available to editors and referees in the event of peer review. For more information see <http://www.nature.com/authors/policies/availability.html> or contact me.

Nature Cell Biology is committed to improving transparency in authorship. As part of our efforts in this direction, we are now requesting that all authors identified as 'corresponding author' on published papers create and link their Open Researcher and Contributor Identifier (ORCID) with their account on the Manuscript Tracking System (MTS), prior to acceptance. ORCID helps the scientific community achieve unambiguous attribution of all scholarly contributions. You can create and link your ORCID from the home page of the MTS by clicking on 'Modify my Springer Nature account'. For more information please visit www.springernature.com/orcid.

[Redacted]

We would like to receive a revised submission within six months. We would be happy to consider a revision even after this timeframe, however if the resubmission deadline is missed and the paper is eventually published, the submission date will be the date when the revised manuscript was received.

We hope that you will find our referees' comments, and editorial guidance helpful. Please do not hesitate to contact me if there is anything you would like to discuss.

Best wishes,

Daryl Jason David

Daryl Jason Verzosa David, PhD

Senior Editor, Nature Cell Biology
Nature Portfolio

Heidelberger Platz 3, 14197 Berlin, Germany
Email: daryl.david@nature.com

ORCID: <https://orcid.org/0000-0002-9253-4805>

Reviewers' Comments:

Reviewer #1:

Remarks to the Author:

In this manuscript, Yang et al. report on a novel role of the transcription factor ETV4, which senses cell crowding and transduces it into lineage specification in human pluripotent stem cells. The authors first show that cell crowding, as encountered in iPSC colonies, inhibits ETV4 expression. This effect is also spatially regulated, following the patterns of crowding in cell colonies. Functionally, loss of ETV4 expression derepresses lineage specification into neuroectoderm through its transcriptional activity. Mechanistically, the authors show that increased cell spreading (that is, reduced crowding) increases integrin-mediated endocytosis of FGF receptors, which then increase ERK activity. In turn, ERK regulates ETV4. Overall, this work constitutes an impressive study, with a detailed and thorough characterization of a novel, major mechanism regulating lineage specification in stem cells. The authors use also a very wide array of techniques to study their system and verify their hypothesis. I therefore strongly support publication in Nature Cell Biology, although some issues should be addressed before that occurs. I list my points below.

- In figure 1, cell density is controlled by controlling the available area of matrigel-coated substrate, which is either unconstrained, or constrained to millimeter-sized patterns. However, resulting cell density will only be regulated by this if cells are left to proliferate long enough so that substrate area becomes a limiting factor for the smaller patterns. If this is the case or not is not clarified. For how long were cells left to grow on the substrates? Did they eventually occupy all of the patterns, or were they organized in separate colonies? This information is crucial to understand the setup and is not clear from either the text or methods.

- One of the central hypotheses of the theoretical model is that the effect of cell density is mediated by the availability of integrin ligands, which would be higher for highly spread cells. However, this hypothesis does not seem very plausible in my view. Since the early works on the topic, we know that integrin-mediated mechanotransduction relies of course on integrin ligation, but that the effects of cell spreading depend on cell shape itself, and not on the total number of available ligands (see for instance pioneering work by Chen et al, DOI: 10.1126/science.276.5317.1425). In the context of this work, endocytosis is a key step. This is likely to be regulated by membrane tension as discussed by the authors. In turn, membrane tension would increase in response to overall cell spreading, not necessarily to an increased number of integrin ligands. Given the already impressive amount of mechanistic detail provided by the authors, I think further exploring this aspect experimentally is not necessary. However, the authors should revise their theoretical model to consider the effects of cell shape/membrane tension rather than merely integrin ligation. This could be done by merely simplifying the model, and assuming directly an effect of cell shape rather than having to resort to integrin availability.

- Importantly, quantification and information on n numbers and repeats is missing in key figures. For example, panels 3a, 3b, 3g, 4c, 4f, 5j, show simply example images, with no quantification, no information on repeats, and no statistical analyses. Major conclusions are then derived from these panels. This is a major oversight which needs to be corrected thoroughly throughout the manuscript.

- Relatedly, the size of most features in figures (images, text, graphs) is extremely small and often unreadable/undecipherable. This needs to be corrected.

- From the text and methods, it is not clear if nuclear intensities (such as for instance in panel 2i) report simply the intensity, or the nuclear to cytoplasmic ratio. This should be clarified. Including quantifications on nuclear to cytoplasmic ratio would be useful particularly in the cases where ETV4 is compared to YAP, as these ratios are the standard way to assess YAP responses.

Reviewer #2:
Remarks to the Author:
Yang et al.

In this study, Yang and colleagues examine how cell crowding in culture relates to fate determination in human pluripotent stem cells (hPSC). The authors show that cell crowding results in a decrease in ETV4 expression, through modification of the integrin-cytoskeleton interaction and decreased FGFR endocytosis leading to changes in protein stability. They describe a model that relates crowding phenomena to ETV expression.

There is currently great interest in the use of hPSC to model events in early human development, and in the role of the physical environment in cell fate decisions. This study provides interesting novel data on the effects of cell crowding on lineage specification, and the role of ETV4 proteins in the response to crowding.

Two dimensional cultures have been used to explore patterning events that reflect some aspects of gastrulation. In this system, the laboratories of Brivanlou, Warmflash and others have shown a strong relationship between localized morphogen signaling and patterning. Here, Yang et al. attribute the control of patterning to cell density and downstream effects thereof. However, it is very difficult experimentally to separate out effects of "cell crowding" on multiple parameters related to position in the colony, including access to nutrients and oxygen (limited to apical aspect of the cell inside the colony in adherent cultures on plastic, which do not mimic conditions in the epiblast), local concentrations of morphogens, the differential distribution of receptors in cells on the outer and inner zones of the colony, and the extent of interaction of the cell with the substrate and extracellular matrix. All of these considerations complicate the interpretation of the experiments described here. For example, see Etoc et al. *Developmental Cell* 39: 302 2016 for an demonstration of how the integration of receptor distribution in different regions of the colony with spatially regulated morphogen distribution works to pattern the colony. These complications do not invalidate the authors' work but I think they oversimplify the situation by focusing on one transcription factor and one set of interactions. The authors should attempt to relate their findings to the extensive previous work done in this 2D system. Also, the authors should be careful to compare the cell densities used in their experiments with those of other authors using this system, and with what obtains in the epiblast in vitro, to convince readers that the phenomena described are not an artefact of particularly high density culture.

Specific points

1. P 2 Line 19 most would not argue that simple epithelium expansion of the epiblast results in cell fate determination; rather that it is a consequence largely of cell interactions between the epiblast cells and extraembryonic tissue.
2. P3 line 11 indicate what aspects of self-organization are not adequately explained by morphogen activity. What exactly are the deficiencies in the previous literature that the authors are trying to address?
3. P4 line 5-what evidence is there that contact inhibition of proliferation as observed classically in

fibroblasts also operates in hPSC culture

4. Page 4 line 11-limiting the adhesive area also restricts cell-matrix interactions in addition to density. How can the authors deconvolute these phenomena?

5. Figure 1 a and throughout-it is essential that the authors relate the cell density range that they study to those examined in earlier works on patterning in 2D, in routine culture, and in the primate post-implantation epiblast. It is my impression that for the most part, the authors mostly employ very high density cultures relative to these other models and the embryo.

6. Figure 1 and throughout-in addition to factors related to mechanical stress, crowding could well affect access to nutrients and therefore metabolism. Can the authors rule this out in their system? What is the metabolic status of cells on the outside relative to those in the interior?

7. Page 5 L13-the authors focused on PEA3 ETS domain factors, but this is just a subset of transcription factors identified in this analysis. How was this choice justified? Other factors listed in the table may play diverse roles in lineage specification.

8. Figure 1f-was cell size increased distal to the cut site as well, ETV4 is upregulated throughout

9. Extended data Fig 1 h-decreased cell size is again accompanied here by reduced area for cell substrate interaction

10. Figure 1 jkl-comparison with previous work is important. Matrigel and other hydrogels are considerably softer than cell culture plastic. Was plastic alone used in this experiment? Are the stiffness values cited here referring to plastic alone?

11. Page 7 L18-many previous studies have shown that during routine growth, colonies of hPSC do not in fact maintain equivalent expression of pluripotency factors throughout. Many workers have shown higher expression of pluripotency markers at the outer edge of maturing colonies. The authors need to account for this discrepancy.

12. Page 8 L7-describe the knockdown experiment and results clearly. "YAP immunostaining" was not validated by the knockdown.

13. Extended data Figure 2k- it is impossible to discern the quality of Ki67 staining from this micrograph. In fact the cells appear to have similar sized nuclei throughout the colony. This question of proliferative rates is important. Authors could easily do double label flow cytometry with ETV4 and EdU.

14. Page 9 l11-use of FGF in mesoderm differentiation is somewhat unconventional, would BMP alone do the same thing. Methods section refers to use of BMP4 only.

15. Page 11 L6-did the data of Nakanishi show elevated ETV4 in the NCAD positive fraction? Or are the authors simply inferring without justification that the outer colony cells in these two experiments are similar?

16. Figure 4-these micrographs do not seem to me to lend strong support to conclusions regarding cytoplasmic versus nuclear localization of the ERK signal.

17. Extended data Figure 5e does not seem to show expected nuclear localization of p-ERK anywhere

18. Page 13 line 3-what is the evidence that ERK signaling specifically affects ETV4 as against many other potential regulatory factors?

19. Page 14 L6-the FGF system is essential for maintenance of hPSC pluripotency and reducing its activity has diverse effects.

20. Page 15 L10 onwards-again the localization of the receptor in endosomes may relate to surface area available for binding, i.e. accessibility to ligand.

21. Page 16 L14-effects of FGF signaling on neural induction have been extensively investigated and are complex and stage dependent.

22. Page 17 l9-effects on integrin distribution could affect many aspects of cell behavior and response to a number of signals.

Reviewer #3:
Remarks to the Author:

Yang&al are presenting a study of patterning of human embryonic stem cells epithelia. They show that the transcription factor ETV4 is inhibited at the center of a growing hESC epithelial disc, and that this inhibition plays a role in later fate selection towards mesendoderm or neuroectoderm cell types. They propose that cell crowding explains the observed patterning behaviour: cells in the center of the colony have a smaller area which triggers ETV4 down regulation. They propose that a signalling cascade involving FGF, ERK signalling is the origin of the down regulation of ETV4 in the crowded region of the tissue.

The authors investigate the mechanism for symmetry breaking of a circular hESC tissue, and are proposing a mechanism based on cell crowding. I find that overall the study is interesting and convincing and sheds light on signalling pathways involved in symmetry breaking in a homogeneous cell population and cell fate specification.

Main comments:

- The authors make a convincing case for the relationship between cell crowding and down regulation of ETV4. It is less clear how this coupling occurs and whether ETV4 is really a « mechanical transducer ». The model and Fig. 6k seem to suggest a role for the cell area whose decrease leads to a total lower rate of FGF endocytosis. I am not sure if this effect can be called « mechanical », it rather seems a consequence of cell geometry. In other words, what is the evidence that a stress is sensed by the system? On the other hand, Fig. 6k shows an effect of actomyosin and cell-ECM interaction; supported by experiments showing that perturbation of actomyosin affects ETV4 and lineage specification. It is not clear to me if these effects occur through a change in cell area brought by perturbation of actomyosin localisation, or directly through intracellular signalling as suggested by the schematic of Fig. 6k. In other words, I am not sure convincing that interns and actomyosin play a signalling role, as opposed to affecting cell shape. Can this point be clarified?

- I appreciate that the authors aimed at recapitulate some of their findings with a simple model. I have a couple of comments nevertheless about the model and its relation to experimental data:

1) The authors rightly compare their model to population density profiles in Fig. 7b. It seems that it would be easy (and informative as to the explanation power of the model) to also measure profiles of ETV4 intensity and compare them to model predictions. Can the authors perform this comparison?

2) In that respect, I do not understand why the authors use the « integral fractional occupancy » θ as the model read-out, and not directly ETV4, which is what is shown in Fig. 7. Or if they think θ is more informative, can they provide with experimental measurements of θ ?

3) I understand that a central conclusion of the model is that a simple threshold of cell density explains whether ETV4 is on or off, at different times. If so, I think this conclusion could be shown more directly; for instance plotting ETV4 intensity as a function of cell density, for different regions of the tissue and different times, should reveal this relationship in a direct way?

4) It is not entirely clear that using a diffusion equation (Eq. 1) to describe the growth of the tissue is appropriate. It does not seem that Refs. 75 and 76 which the authors are citing are describing tissue density dynamics in this way. In principle, a diffusion equation applies to describe the concentration of non-interacting, randomly moving objects. Here, the tissue is cohesive and therefore cell-cell interactions likely dominate over random cell motion. This difficulty is reflected in the second boundary condition in Eq. 3, which imposes that the concentration vanishes at a fixed radius, $r=1$. I don't see a clear justification for this boundary condition (why would the cell concentration vanish at a fixed point in space?) and that also means that the authors are not showing the full solution of their model in Fig. 7b; as in their model the concentration of cells decreases smoothly to zero in space, in contrast to

experiments where it abruptly decreases to zero.

5) As far as I can see, there is no information on how the model parameters are obtained? The text says that they are « measured experimentally », but how this is done is not clarified. Also, Fig. 7h is rather cryptic and I don't see what it is bringing.

- There seems to be in the bottom-most experimental panel of Fig. 7f, a central patch of cells which are ETV4-positive. Can the authors comment on the origin of this patch?

- I have difficulties seeing that integrin is « cytoplasmic » in the center of the colony and less so at its periphery, in Fig. 6b. I rather see a strong signal close to the nuclei in the center, and a smaller signal at the periphery. Can the authors clarify this?

Minor comments:

- The figure panels are overall very small and difficult to read, some panels (e.g Fig. 6a) have impossibly small numbers. I would suggest that the authors improve figure readability.

- While it is clear in the context of the paper, the authors sometimes refer to « cell size » to refer to « cell apical area ». Because cell size could also refer to cell volume, I would suggest to clarify this possible ambiguity.

Methods should be written concisely, but should contain all elements necessary to allow interpretation and replication of the results. As a guideline, Methods sections typically do not exceed 3,000 words. The Methods should be divided into subsections listing reagents and techniques. When citing previous methods, accurate references should be provided and any alterations should be noted. Information must be provided about: antibody dilutions, company names, catalogue numbers and clone numbers for monoclonal antibodies; sequences of RNAi and cDNA probes/primers or company names and catalogue numbers if reagents are commercial; cell line names, sources and information on cell line identity and authentication. Animal studies and experiments involving human subjects must be reported in detail, identifying the committees approving the protocols. For studies involving human subjects/samples, a statement must be included confirming that informed consent was obtained. Statistical analyses and information on the reproducibility of experimental results should be provided in a section titled "Statistics and Reproducibility".

All Nature Cell Biology manuscripts submitted on or after March 21 2016 must include a Data availability statement at the end of the Methods section. For Springer Nature policies on data availability see <http://www.nature.com/authors/policies/availability.html>; for more information on this

particular policy see <http://www.nature.com/authors/policies/data/data-availability-statements-data-citations.pdf>. The Data availability statement should include:

- Accession codes for primary datasets (generated during the study under consideration and designated as "primary accessions") and secondary datasets (published datasets reanalysed during the study under consideration, designated as "referenced accessions"). For primary accessions data should be made public to coincide with publication of the manuscript. A list of data types for which submission to community-endorsed public repositories is mandated (including sequence, structure, microarray, deep sequencing data) can be found here <http://www.nature.com/authors/policies/availability.html#data>.
- Unique identifiers (accession codes, DOIs or other unique persistent identifier) and hyperlinks for datasets deposited in an approved repository, but for which data deposition is not mandated (see here for details <http://www.nature.com/sdata/data-policies/repositories>).
- At a minimum, please include a statement confirming that all relevant data are available from the authors, and/or are included with the manuscript (e.g. as source data or supplementary information), listing which data are included (e.g. by figure panels and data types) and mentioning any restrictions on availability.
- If a dataset has a Digital Object Identifier (DOI) as its unique identifier, we strongly encourage including this in the Reference list and citing the dataset in the Methods.

We recommend that you upload the step-by-step protocols used in this manuscript to the Protocol Exchange. More details can be found at www.nature.com/protocolexchange/about.

All imaging data should be accompanied by scale bars, which should be defined in the legend. Cropped images of gels/blots are acceptable, but need to be accompanied by size markers, and to retain visible background signal within the linear range (i.e. should not be saturated). The boundaries of panels with low background have to be demarked with black lines. Splicing of panels should only be considered if unavoidable, and must be clearly marked on the figure, and noted in the legend with a statement on whether the samples were obtained and processed simultaneously. Quantitative comparisons between samples on different gels/blots are discouraged; if this is unavoidable, it should only be performed for samples derived from the same experiment with gels/blots were processed in parallel, which needs to be stated in the legend.

Figures should be provided at approximately the size that they are to be printed at (single column is 86 mm, double column is 170 mm) and should not exceed an A4 page (8.5 x 11"). Reduction to the scale that will be used on the page is not necessary, but multi-panel figures should be sized so that the whole figure can be reduced by the same amount at the smallest size at which essential details in each panel are visible. In the interest of our colour-blind readers we ask that you avoid using red and green for contrast in figures. Replacing red with magenta and green with turquoise are two possible

colour-safe alternatives. Lines with widths of less than 1 point should be avoided. Sans serif typefaces, such as Helvetica (preferred) or Arial should be used. All text that forms part of a figure should be rewritable and removable.

SUPPLEMENTARY INFORMATION – Supplementary information is material directly relevant to the conclusion of a paper, but which cannot be included in the printed version in order to keep the manuscript concise and accessible to the general reader. Supplementary information is an integral part of a Nature Cell Biology publication and should be prepared and presented with as much care as the main display item, but it must not include non-essential data or text, which may be removed at the editor's discretion. All supplementary material is fully peer-reviewed and published online as part of the HTML version of the manuscript. Supplementary Figures and Supplementary Notes are

appended at the end of the main PDF of the published manuscript.

The total number of Supplementary Figures (not including the “unprocessed scans” Supplementary Figure) should not exceed the number of main display items (figures and/or tables (see our Guide to Authors and March 2012 editorial <http://www.nature.com/ncb/authors/submit/index.html#suppinfo>; <http://www.nature.com/ncb/journal/v14/n3/index.html#ed>). No restrictions apply to Supplementary Tables or Videos, but we advise authors to be selective in including supplemental data.

GUIDELINES FOR EXPERIMENTAL AND STATISTICAL REPORTING

REPORTING REQUIREMENTS – To improve the quality of methods and statistics reporting in our papers we have recently revised the reporting checklist we introduced in 2013. We are now asking all life sciences authors to complete two items: an Editorial Policy Checklist (found here <https://www.nature.com/authors/policies/Policy.pdf>) that verifies compliance with all required editorial policies and a reporting summary (found here <https://www.nature.com/authors/policies/ReportingSummary.pdf>) that collects information on experimental design and reagents. These documents are available to referees to aid the evaluation of the manuscript. Please note that these forms are dynamic ‘smart pdfs’ and must therefore be downloaded and completed in Adobe Reader. We will then flatten them for ease of use by the reviewers. If you would like to reference the guidance text as you complete the template, please access these flattened versions at <http://www.nature.com/authors/policies/availability.html>.

STATISTICS – Wherever statistics have been derived the legend needs to provide the n number (i.e. the sample size used to derive statistics) as a precise value (not a range), and define what this value represents. Error bars need to be defined in the legends (e.g. SD, SEM) together with a measure of centre (e.g. mean, median). Box plots need to be defined in terms of minima, maxima, centre, and percentiles. Ranges are more appropriate than standard errors for small data sets. Wherever statistical significance has been derived, precise p values need to be provided and the statistical test used needs to be stated in the legend. Statistics such as error bars must not be derived from $n < 3$. For sample sizes of $n < 5$ please plot the individual data points rather than providing bar graphs. Deriving statistics from technical replicate samples, rather than biological replicates is strongly discouraged. Wherever statistical significance has been derived, precise p values need to be provided and the

statistical test stated in the legend.

Author Rebuttal to Initial comments

Reviewers' Comments:

Reviewer #1:

Remarks to the Author:

In this manuscript, Yang et al. report on a novel role of the transcription factor ETV4, which senses cell crowding and transduces it into lineage specification in human pluripotent stem cells. The authors first show that cell crowding, as encountered in iPSC colonies, inhibits ETV4 expression. This effect is also spatially regulated, following the patterns of crowding in cell colonies. Functionally, loss of ETV4 expression derepresses lineage specification into neuroectoderm through its transcriptional activity. Mechanistically, the authors show that increased cell spreading (that is, reduced crowding) increases integrin-mediated endocytosis of FGF receptors, which then increase ERK activity. In turn, ERK regulates ETV4. Overall, this work constitutes an impressive study, with a detailed and thorough characterization of a novel, major mechanism regulating lineage specification in stem cells. The authors use also a very wide array of techniques to study their system and verify their hypothesis. I therefore strongly support publication in Nature Cell Biology, although some issues should be addressed before that occurs. I list my points below.

- We appreciate the reviewer for supporting our study. We have addressed all the issues by additional experiments.

- In figure 1, cell density is controlled by controlling the available area of matrigel-coated substrate, which is either unconstrained, or constrained to millimeter-sized patterns. However, resulting cell density will only be regulated by this if cells are left to proliferate long enough so that substrate area becomes a limiting factor for the smaller patterns. If this is the case or not is not clarified. For how long were cells left to grow on the substrates? Did they eventually occupy all of the patterns, or were they organized in separate colonies? This information is crucial to understand the setup and is not clear from either the text or methods.

A: hESCs were seeded on Matrigel micropatterns and left to grow for 2~3 days until they occupied the patterned area entirely. This point is clarified in the method (Page 26 Line 12) and in Fig. 1a.

- One of the central hypotheses of the theoretical model is that the effect of cell density is mediated by the availability of integrin ligands, which would be higher for highly spread cells. However, this hypothesis does not seem very plausible in my view. Since the early works on the topic, we know that integrin-mediated mechanotransduction relies of course on integrin ligation, but that the effects of cell spreading depend on cell shape itself, and not on the total number of available ligands (see

for instance pioneering work by Chen et al, DOI: 10.1126/science.276.5317.1425).

A: We concur that beyond cell surface area, cell shape plays a substantial role in integrin-mediated mechanotransduction. Although it is exceptionally challenging to experimentally tease apart the individual contributions of cell surface area and cell geometry in this context, we devised a strategy to tackle this issue. Leveraging the observation that ETV4 experiences a sharp inactivation at a specific cell density threshold, we conducted measurements of the Cell Shape Index (CSI) and cell surface area at the boundary of this transition. Notably, despite the significant alterations in cell surface area, both subsets of cells, distinguished as ETV4-high and ETV4-low, displayed remarkably similar CSIs (Extended Data Fig. 3a). This observation suggests that alterations in cell geometry have a limited impact on the regulation of ETV4 expression in the context of our study. Please see Page 8 Line 7 in the revised manuscript.

In the context of this work, endocytosis is a key step. This is likely to be regulated by membrane tension as discussed by the authors. In turn, membrane tension would increase in response to overall cell spreading, not necessarily to an increased number of integrin ligands. Given the already impressive amount of mechanistic detail provided by the authors, I think further exploring this aspect experimentally is not necessary. However, the authors should revise their theoretical model to consider the effects of cell shape/membrane tension rather than merely integrin ligation. This could be done by merely simplifying the model, and assuming directly an effect of cell shape rather than having to resort to integrin availability.

A: As the reviewer mentioned, the regulation of receptor endocytosis is orchestrated by multiple inputs, including integrin signaling, cell shape, and membrane tension, all converging on actin dynamics. To reinforce the pivotal role of actin in FGFR1 endocytosis, we validated our findings with blebbistatin, a myosin inhibitor, by other widely recognized actin dynamics inhibitors, YM (Y-27632 + ML-7) (Extended Data Fig. 9d,j,k,l). Furthermore, the activation of actomyosin by constitutively active RhoA was sufficient to reactivate FGFR1 endocytosis in crowded cells (Fig. 6i). These results collectively highlight the significance of actin dynamics in the process of FGFR1 endocytosis and its downstream signaling pathways.

Moreover, our new results unveiled that the inhibition of integrin signaling by a FAK inhibitor alone was sufficient to attenuate FGFR1 endocytosis (Extended Data Fig. 9c,d). While our results clearly establish the substantial involvement of integrin signaling and actin dynamics in receptor endocytosis, we acknowledge the potential contribution of cell shape and membrane tension. To address this, we have made a note of this possibility in the revised manuscript. Please see Page 19 Line 19.

- Importantly, quantification and information on n numbers and repeats is missing in key figures. For example, panels 3a, 3b, 3g, 4c, 4f, 5j, show simply example images, with no quantification, no information on repeats, and no statistical analyses. Major conclusions are then derived from these panels. This is a major oversight which needs to be corrected thoroughly throughout the manuscript.

A: Done as suggested. Please see our revised figures (Fig. 3a,b,g, 4f, 5j, 6j and Extended Data Fig. 2a,c,d,e,f,g, 3f, 4a, 6a,f, 7a, 8c,g,h,i,j).

- Relatedly, the size of most features in figures (images, text, graphs) is extremely small and often unreadable/undecipherable. This needs to be corrected.

A: Done as suggested. Please see our revised figures.

- From the text and methods, it is not clear if nuclear intensities (such as for instance in panel 2i) report simply the intensity, or the nuclear to cytoplasmic ratio. This should be clarified. Including quantifications on nuclear to cytoplasmic ratio would be useful particularly in the cases where ETV4 is compared to YAP, as these ratios are the standard way to assess YAP responses.

A: In Fig. 2i, we quantified the nuclear intensities for both ETV4 and YAP signals. In response to a valuable suggestion, we further extended our analysis to include the measurement of the nuclear-to-cytoplasmic ratio of YAP. We have subsequently validated the findings in Fig. 2i and presented these additional results in Extended Data Fig. 3f. To enhance clarity and comprehension, we have revised the figure legends accordingly.

Reviewer #2:

Remarks to the Author:

Yang et al.

In this study, Yang and colleagues examine how cell crowding in culture relates to fate determination in human pluripotent stem cells (hPSC). The authors show that cell crowding results in a decrease in ETV4 expression, through modification of the integrin-cytoskeleton interaction and decreased FGFR endocytosis leading to changes in protein stability. They describe a model that relates crowding phenomena to ETV expression.

There is currently great interest in the use of hPSC to model events in early human development,

and in the role of the physical environment in cell fate decisions. This study provides interesting novel data on the effects of cell crowding on lineage specification, and the role of ETV4 proteins in the response to crowding.

- We appreciate the reviewer for finding our work interesting and novel.

Two dimensional cultures have been used to explore patterning events that reflect some aspects of gastrulation. In this system, the laboratories of Brivanlou, Warmflash and others have shown a strong relationship between localized morphogen signaling and patterning. Here, Yang et al. attribute the control of patterning to cell density and downstream effects thereof. However, it is very difficult experimentally to separate out effects of "cell crowding" on multiple parameters related to position in the colony, including access to nutrients and oxygen (limited to apical aspect of the cell inside the colony in adherent cultures on plastic, which do not mimic conditions in the epiblast), local concentrations of morphogens, the differential distribution of receptors in cells on the outer and inner zones of the colony, and the extent of interaction of the cell with the substrate and extracellular matrix. All of these considerations complicate the interpretation of the experiments described here. For example, see Etoc et al. *Developmental Cell* 39: 302 2016 for an demonstration of how the integration of receptor distribution in different regions of the colony with spatially regulated morphogen distribution works to pattern the colony.

A: We agree that cellular density exerts a significant influence on various cellular processes. To elucidate this phenomenon, we have undertaken new experiments to examine the impact of cell density on the expression of genes related to glycolysis and hypoxia (Extended Data Fig. 3b), as well as the accessibility of oxygen and glucose (Extended Data Fig. 3c-e). Additionally, we have explored the effect of cell crowding on the expression of ETV4 in hESCs cultured within a transwell system (Extended Data Fig. 8f). This system enables the exposure of ligands to both the apical and basolateral sides of the cells. Please refer to our detailed responses below for further clarification on these specific points.

These complications do not invalidate the authors' work but I think they oversimplify the situation by focusing on one transcription factor and one set of interactions. The authors should attempt to relate their findings to the extensive previous work done in this 2D system.

A: As advised, we have contextualized our study by drawing connections to prior research conducted using 2D gastruloid systems, and we have incorporated this aspect into our discussion. Please see Page 23, Line 3.

Also, the authors should be careful to compare the cell densities used in their experiments with those of other authors using this system, and with what obtains in the epiblast in vitro, to convince readers that the phenomena described are not an artefact of particularly high density culture.

A: We have compared the cell densities used in our study to those from other 2D gastruloid systems. Please refer to our detailed response to the specific point below.

Specific points

1. P 2 Line 19 most would not argue that simple epithelium expansion of the epiblast results in cell fate determination; rather that it is a consequence largely of cell interactions between the epiblast cells and extraembryonic tissue.

A: We clarified the sentence in the revised manuscript. Please see Page 2 Line 19.

2. P3 line 11 indicate what aspects of self-organization are not adequately explained by morphogen activity. What exactly are the deficiencies in the previous literature that the authors are trying to address?

A: We acknowledge that the sentence may have been misleading, and we have made the revisions in the revised manuscript. Please see Page 3 Line 14.

3. P4 line 5-what evidence is there that contact inhibition of proliferation as observed classically in fibroblasts also operates in hPSC culture

A: In this study, we have discovered the varying sensitivities of ETV4 and YAP to changes in cell crowding within hESCs (Fig. 2i,j). These distinct mechanotransducers may play a crucial role in decoupling the processes of lineage specification and cell proliferation in response to alterations in cell density. However, we acknowledge that the original sentence may have lacked clarity and context, and we have opted to remove it in the revised manuscript. Please see Page 4 Line 5.

4. Page 4 line 11-limiting the adhesive area also restricts cell-matrix interactions in addition to density. How can the authors deconvolute these phenomena?

A: As the reviewer pointed out, increased cell density imposes limitations on the available cell surface area for cellular interactions with the extracellular matrix. Our data indicate that a pivotal mechanistic response governing ETV4 regulation is the inactivation of integrins due to diminished cell-matrix interactions. In this revised manuscript, we have undertaken additional experiments to bolster our findings, including the assessment of integrin inactivation induced by cell crowding through pFAK staining (an indicative marker of integrin activity) (Fig. 6c). Furthermore, we have probed the

involvement of integrin signaling in the regulation of ETV4 by employing a FAK inhibitor (Extended Data Fig. 9c-f). Together with the original data, these new results strongly support that a reduction in cell-matrix interactions, brought about by elevated cell density, triggers the inactivation of integrin signaling, ultimately leading to the downregulation of ETV4.

5. Figure 1 a and throughout-it is essential that the authors relate the cell density range that they study to those examined in earlier works on patterning in 2D, in routine culture, and in the primate post-implantation epiblast. It is my impression that for the most part, the authors mostly employ very high density cultures relative to these other models and the embryo.

A: In their 2D gastruloid models, Etoc et al. utilized cell densities ranging from 1,000 to 10,000 cells/mm², designating 7,000 cells/mm² as a high-density condition (ref. 6). Notably, they demonstrated that this density closely mirrors the cell density of the epiblast during the early gastrulation stage in human embryos (Carnegie stage 6b). In alignment with this precedent, our study explores a density range spanning from 2,000 to 10,000 cells/mm² with ETV4 inactivation occurring at the threshold of 5,000 cells/mm², underscoring the consistency of our density range with prior research. We clarified this point in the revised manuscript (Page 9 Line 11, Page 26 Line 13). Furthermore, we conducted additional experiments to clearly elucidate the impacts of various cell densities on integrin activity (as assessed by pFAK) and ETV4 expression (Fig. 2f, 6c). Consistent with our original results, sharp inactivation of integrin and ETV4 occurred at around the density of 5,000 cells/mm².

6. Figure 1 and throughout-in addition to factors related to mechanical stress, crowding could well affect access to nutrients and therefore metabolism. Can the authors rule this out in their system? What is the metabolic status of cells on the outside relative to those in the interior?

A: We appreciate the significance of this comment, recognizing the complexity of assessing all potential effects of cell crowding. To address this concern, we initially performed Gene Set Enrichment Analysis (GSEA) using RNA-seq data obtained from crowded hESCs. This analysis revealed no substantial alterations in the expression of genes associated with glycolysis and hypoxia (Extended Data Fig. 3b). To substantiate these bioinformatic findings, we conducted experiments employing the hypoxyprobe system and glucose uptake assays (Extended Data Fig. 3c-e). In light of the fact that hESCs form a monolayer epithelium, these collective results imply that cell crowding exerts limited influence on the accessibility of oxygen and nutrients in hESCs. Please see Page 8 Line 13 in the revised manuscript.

7. Page 5 L13-the authors focused on PEA3 ETS domain factors, but this is just a subset of transcription factors identified in this analysis. How was this choice justified? Other factors listed in the table may play diverse roles in lineage specification.

A: We acknowledge that other factors in Table S1 are likely involved in sensing mechanical cues and governing lineage specification in hESCs. Nevertheless, our primary focus within this study was directed toward ETV4 for several compelling reasons. Firstly, all three members of the PEA3 transcription factor family are featured in the list, and secondly, ETV4 exhibits high expression levels (FPKM=72.9) among the 40 transcription factors in the list (as illustrated in the accompanying graph). Most importantly, the ectopic expression of ETV4 was found to completely block cell crowding-induced neuroectoderm derivation (Extended Data Fig. 4i), underscoring the pivotal role of ETV4 in this context. We recognize that exploring the roles of these other factors in stem cell biology presents an exciting avenue for future research and investigation.

8. Figure 1f-was cell size increased distal to the cut site as well, ETV4 is upregulated throughout

A: Following the initial scratch, we observed a phenomenon wherein cells located near the cut site underwent a spreading process to occupy the empty space. As time progressed, this spreading area gradually expanded, paralleled by a proportional increase in the region where ETV4 upregulation was observed. In our revised manuscript, we have enhanced the clarity of the figure image to provide a more lucid representation of these dynamic processes (Fig. 1f). Additionally, we confirmed a reduction in cell density within the region where ETV4 was reactivated, substantiating our observations (See the data below).

9. Extended data Fig 1 h-decreased cell size is again accompanied here by reduced area for cell substrate interaction

A: Please see our response to Comment #4.

10. Figure 1 jkl-comparison with previous work is important. Matrigel and other hydrogels are considerably softer than cell culture plastic. Was plastic alone used in this experiment? Are the stiffness values cited here referring to plastic alone?

A: In both plastic and hydrogel plates, a thin Matrigel coating was applied to facilitate cell attachment. It's important to note that the stiffness values reported in our study represent the properties of the underlying plastic and hydrogel substrates. This is because the thin coating process involved incubating the plates with media containing diluted Matrigel (1% v/v), which does not significantly alter the intrinsic stiffness of the plastic and hydrogel materials. We have provided further clarification on this matter in the methods section. Please see Page 25 Line 20.

11. Page 7 L18-many previous studies have shown that during routine growth, colonies of hPSC do not in fact maintain equivalent expression of pluripotency factors throughout. Many workers have shown higher expression of pluripotency markers at the outer edge of maturing colonies. The authors need to account for this discrepancy.

A: To address this issue, we performed a quantitative analysis of OCT4, NANOG, and SOX2 expression, revealing no substantial differences between the peripheral and central regions of large hESC colonies (Extended Data Fig. 2d). This discrepancy may stem from technical variations inherent to cell culture methods. Furthermore, it is noteworthy that previous research has documented an outer region characterized by heightened expression of pluripotency markers, typically spanning 1

to 3 cell layers from the colony's edge. Importantly, this pattern does not align with the specific boundary where ETV4 inactivation was observed in our study.

12. Page 8 L7-describe the knockdown experiment and results clearly. "YAP immunostaining" was not validated by the knockdown.

A: As suggested, we conducted validation of YAP shRNAs by quantifying the mRNA levels of YAP and its target genes, CTGF and CYR61 (Extended Data Fig. 3h). We have also provided further clarification on this aspect in the revised manuscript. Please see Page 9 Line 4.

13. Extended data Figure 2k- it is impossible to discern the quality of Ki67 staining from this micrograph. In fact the cells appear to have similar sized nuclei throughout the colony. This question of proliferative rates is important. Authors could easily do double label flow cytometry with ETV4 and EdU.

A: As recommended, we have made improvements to the Ki67 staining image (Extended Data Fig. 3k) and additionally conducted an EdU incorporation assay. In line with the Ki67 staining results, our findings from the EdU incorporation assay confirm that ETV4-low cells located in the colony center exhibit a similar percentage of EdU incorporation when compared to ETV4-high cells found at the periphery of large hESC colonies (Extended Data Fig. 3l).

14. Page 9 l11-use of FGF in mesoderm differentiation is somewhat unconventional, would BMP alone do the same thing. Methods section refers to use of BMP4 only.

A: We have followed a well-established and widely-recognized protocol for directing human embryonic stem cells (hESCs) toward the mesendoderm lineage, which involves the combined use of FGF2 and BMP4 (Please see Table 2 in Ref.37). In line with previous studies that have successfully employed this approach for mesendoderm derivation, we have referenced these sources accordingly (Page 10 Line 10). Specifically, following the protocol outlined by Yu et al. (Ref. 38), we utilized mTeSR1 as the basal medium, which inherently contains FGF2, and supplemented it with BMP4 to induce mesendoderm differentiation. We have provided further clarification on this methodology in the Methods section of our revised manuscript. Please see Page 26 Line 2.

15. Page 11 L6-did the data of Nakanishi show elevated ETV4 in the NCAD positive fraction? Or are the authors simply inferring without justification that the outer colony cells in these two experiments

are similar?

A: We acknowledge the significance of this technical comment. In alignment with the post-translational regulation of ETV4 expression, as depicted in Fig. 4, our analysis revealed no significant difference in the RNA levels of ETV4 within the NCAD-positive cell population. However, an increase in the RNA levels of DUSP6, a well-established target gene of ETV4, was observed in these NCAD-positive cells (Extended Data Fig. 5e). This observation implies heightened ETV4 activity in this context. In fact, our subsequent experiments substantiated that DUSP6 expression is intricately controlled by ETV4 in hESCs (Extended Data Fig. 5d). Please see Page 12 Line 4.

16. Figure 4-these micrographs do not seem to me to lend strong support to conclusions regarding cytoplasmic versus nuclear localization of the ERK signal.

A: In contrast to the clear cytoplasmic localization of ERK-KTR in the peripheral region of large hESC colonies, we have observed an increased nuclear signal of ERK-KTR in the central area of these colonies. We have updated the figure images for improved clarity. Please see Fig. 4b,c.

17. Extended data Figure 5e does not seem to show expected nuclear localization of p-ERK anywhere

A: In this study, we employed two distinct approaches to assess ERK activity: ERK-KTR and phosphorylated ERK (pERK). In the ERK-KTR assay, the nuclear localization of the fluorescent reporter serves as an indicator of ERK inactivation. Conversely, in the case of pERK, the total levels of phosphorylated ERK reflect ERK activity, as ERK can undergo phosphorylation in both the cytosolic and nuclear compartments of cells. Notably, within the large hESC colonies, we observed elevated levels of pERK in both the cytosol and the nucleus of peripheral cells. To enhance clarity and provide a more detailed illustration, we have made updates to the figure representing these findings. Please see Extended Data Fig. 6f.

18. Page 13 line 3-what is the evidence that ERK signaling specifically affects ETV4 as against many other potential regulatory factors?

A: It is indeed true that ERK signaling can exert a broad influence on various regulatory factors. Nevertheless, our comprehensive dataset strongly supports the assertion that ETV4 serves as a pivotal mediator responsible for transducing the effects of ERK signaling on lineage specification. This conclusion is substantiated by several key findings: Firstly, inhibiting ERK activity led to a reduction in ETV4 expression (Fig. 4d) and a consequent derepression of neuroectoderm

differentiation (Fig. 4f)—a phenomenon mirrored when we specifically inhibited ETV4 (Fig. 3e). Most importantly, when we introduced ETV4 overexpression, it completely abolished the neuroectoderm derepression caused by ERK inhibition (Fig. 4g). These compelling results underscore the significance of ERK, with ETV4 as its major effector, in the context of lineage specification processes. We have clarified this point in the revised manuscript. Please see Page 14 Line 22.

19. Page 14 L6—the FGF system is essential for maintenance of hPSC pluripotency and reducing its activity has diverse effects.

A: Considering the pivotal role of FGF signaling in maintaining hPSC pluripotency, we adopted a strategic approach in our study. Specifically, we treated the cells with SU5402, an inhibitor targeting FGF receptors, for a short duration, limited to a maximum of 15 minutes (Fig. 5b). This short exposure was designed to minimize any potential indirect effects of FGFR inhibition. Remarkably, our results unveiled a substantial reduction in ERK activity even as early as 5 minutes following SU5402 treatment (Fig. 5b). These findings strongly support that FGF signaling directly regulates ERK activity in hESCs.

20. Page 15 L10 onwards—again the localization of the receptor in endosomes may relate to surface area available for binding, i.e. accessibility to ligand.

A: Since hESCs typically grow in a monolayer colony and FGF receptors are present on the apical side of crowded cells (Extended Data Fig. 8g), it is less likely that reduced surface area alone obstructs FGF receptors' accessibility to the ample FGF ligands present in stem cell media. Instead, we focused on exploring integrin-actomyosin signaling as a key regulator of receptor endocytosis. Our investigations revealed that inhibiting integrin signaling and actomyosin activity was sufficient to impede FGFR endocytosis within an hour after treatment (Fig. 6f-h, Extended Data Fig. 9c-l). Furthermore, actomyosin activation by constitutively active RhoA was able to at least partially reactivate FGFR endocytosis in crowded cells (Fig. 6i), underscoring the substantial role of actomyosin in receptor endocytosis. However, we acknowledge the potential influence of surface area on receptor endocytosis and have addressed this aspect in our revised manuscript. Please see Page 19 Line 19.

21. Page 16 L14—effects of FGF signaling on neural induction have been extensively investigated and are complex and stage dependent.

A: In accordance with the reviewer's feedback, it is evident that FGF signaling assumes a dual role in the context of neural differentiation. Specifically, previous studies highlight its suppressive function during the initial stages of neuroectoderm differentiation, while it transitions into a promoting role during later stages, particularly in driving neural stem cell derivation. Our results consistently demonstrate that inhibiting FGF signaling leads to the derepression of the early neuroectoderm fate, albeit at the cost of reduced mesendoderm derivation. We have cited the previous work in the revised manuscript. Please see Page 17 Line 16.

22. Page 17 19-effects on integrin distribution could affect many aspects of cell behavior and response to a number of signals.

A: We recognize that the relocalization of integrins due to cell crowding can broadly affect numerous cellular processes. Among these, we focused on actin regulation, as it plays a pivotal role in governing receptor endocytosis. We further clarified this point in the revised manuscript. Please see Page 19 Line 3.

Reviewer #3:

Remarks to the Author:

Yang&al are presenting a study of patterning of human embryonic stem cells epithelia. They show that the transcription factor ETV4 is inhibited at the center of a growing hESC epithelial disc, and that this inhibition plays a role in later fate selection towards mesendoderm or neuroectoderm cell types. They propose that cell crowding explains the observed patterning behaviour: cells in the center of the colony have a smaller area which triggers ETV4 down regulation. They propose that a signalling cascade involving FGF, ERK signalling is the origin of the down regulation of ETV4 in the crowded region of the tissue.

The authors investigate the mechanism for symmetry breaking of a circular hESC tissue, and are proposing a mechanism based on cell crowding. I find that overall the study is interesting and convincing and sheds light on signalling pathways involved in symmetry breaking in a homogeneous cell population and cell fate specification.

- We appreciate the reviewer for finding our work interesting and convincing.

Main comments:

- The authors make a convincing case for the relationship between cell crowding and down regulation of ETV4. It is less clear how this coupling occurs and whether ETV4 is really a « mechanical transducer ». The model and Fig. 6k seem to suggest a role for the cell area whose decrease leads to a total lower rate of FGF endocytosis. I am not sure if this effect can be called « mechanical », it rather seems a consequence of cell geometry. In other words, what is the evidence that a stress is sensed by the system? On the other hand, Fig. 6k shows an effect of actomyosin and cell-ECM interaction; supported by experiments showing that perturbation of actomyosin affects ETV4 and lineage specification. It is not clear to me if these effects occur through a change in cell area brought by perturbation of actomyosin localisation, or directly through intracellular signalling as suggested by the schematic of Fig. 6k. In other words, I am not sure convincing that integrins and actomyosin play a signalling role, as opposed to affecting cell shape. Can this point be clarified?

A: We highly value the significance of this feedback. In our study, we uncovered that the reduction in cell surface area due to cell crowding restricts interactions between integrins and the extracellular matrix (ECM), ultimately leading to the inactivation of ETV4 expression. To investigate the potential impact of altered cell geometry induced by cell crowding, we quantified the Cell Shape Index (CSI) at the boundary where ETV4 expression is sharply inactivated. Interestingly, despite substantial changes in cell surface area, both ETV4-high and ETV4-low cells exhibited similar CSIs, indicating that cell geometry has minimal effects on ETV4 expression in this context (Extended Data Fig. 3a).

To gain further insight into the signaling role of integrin, we assessed the levels of phosphorylated FAK (pFAK) as an indicative marker of integrin activity. Notably, the pFAK levels exhibited a sharp decline at a cell density of approximately 5000 cells/mm² (Fig. 6c), closely mirroring the pattern of ETV4 expression (Fig. 2f). Moreover, when integrin signaling was inhibited by a FAK inhibitor, we observed a significant reduction in FGFR1 endocytosis, ERK activity, and ETV4 expression within cells in a low density. (Extended Data Fig. 9c-f) These newly acquired findings strongly imply the pivotal role of integrin signaling in mediating the effects of cell crowding on ETV4 expression. Please see Page 18 Line 19.

- I appreciate that the authors aimed at recapitulate some of their findings with a simple model. I have a couple of comments nevertheless about the model and its relation to experimental data:

1) The authors rightly compare their model to population density profiles in Fig. 7b. It seems that it would be easy (and informative as to the explanation power of the model) to also measure profiles of ETV4 intensity and compare them to model predictions. Can the authors perform this comparison?

A: We appreciate the reviewer for the valuable suggestion. In the revised manuscript, we conducted measurements of ETV4 intensity in relation to varying cell densities within hESCs (Fig. 2f) and revised our model description accordingly (Fig. 7d). Please see Page 21 Line 14.

2) In that respect, I do not understand why the authors use the « integral fractional occupancy » θ as the model read-out, and not directly ETV4, which is what is shown in Fig. 7. Or if they think θ is more informative, can they provide with experimental measurements of θ ?

A: Following the suggestion, we conducted measurements of integrin activity using pFAK staining, a reliable marker indicative of integrin activity, under various density conditions (Fig. 6c, 7d). Please see Page 21 Line 14.

3) I understand that a central conclusion of the model is that a simple threshold of cell density explains whether ETV4 is on or off, at different times. If so, I think this conclusion could be shown more directly; for instance plotting ETV4 intensity as a function of cell density, for different regions of the tissue and different times, should reveal this relationship in a direct way?

A: Please see our response to Comment #1.

4) It is not entirely clear that using a diffusion equation (Eq. 1) to describe the growth of the tissue is appropriate. It does not seem that Refs. 75 and 76 which the authors are citing are describing tissue density dynamics in this way.

A: We have updated and replaced the references in the revised manuscript to ensure relevancy. Please see Page 20 Line 20 (Ref. 82).

In principle, a diffusion equation applies to describe the concentration of non-interacting, randomly moving objects. Here, the tissue is cohesive and therefore cell-cell interactions likely dominate over random cell motion. This difficulty is reflected in the second boundary condition in Eq. 3, which imposes that the concentration vanishes at a fixed radius, $r=1$. I don't see a clear justification for this boundary condition (why would the cell concentration vanish at a fixed point in space?) and that also means that the authors are not showing the full solution of their model in Fig. 7b; as in their model the concentration of cells decreases smoothly to zero in space, in contrast to experiments where it abruptly decreases to zero.

A: We acknowledge and appreciate the reviewer's consideration of the simplifying assumptions incorporated into our modeling approach. While it is indeed true that a more detailed model could

encompass additional factors such as cell-cell communication, nutrient availability, cell elasticity, chemotaxis, and other complex elements, we made a deliberate choice to employ a more straightforward model. This simplified approach allows us to effectively capture the dynamic changes in cell population density over time and establish a direct link between cell population density and ETV4 expression. We firmly believe that such a simplified model is more accessible for interpretation and requires fewer parameters to be experimentally measured.

It's worth noting that our utilization of a reaction-diffusion model with fixed boundary conditions finds precedence in prior research, particularly in modeling the growth of bacterial colonies (see Ref. 82). Nevertheless, we have taken the reviewer's valuable feedback into account, and in our revised manuscript, we have included a brief discussion of the limitations inherent in our modeling approach. Please see Page 31 Line 4.

5) As far as I can see, there is no information on how the model parameters are obtained? The text says that they are « measured experimentally », but how this is done is not clarified. Also, Fig. 7h is rather cryptic and I don't see what it is bringing.

A: Our model relies on three key experimental parameters: the cell division rate, the diffusion coefficient, and β . The cell division rate was chosen within a range of previously measured stem cell doubling rates. The diffusion coefficient and β were estimated by least square fitting to experimentally measured colony growth data and ETV4 profile. To provide further clarity regarding these aspects, we have made specific enhancements to the text in our revised manuscript (Page 30 Line 12, Page 32 Page 10).

In Fig. 7h, our intent is to illustrate the various states of ETV4 as a function of time and cell density. In essence, ETV4 switches to the "ON" state when cellular density falls below a critical threshold (set at 5000 cells/mm²). Notably, the critical density only becomes observable after a certain time point ($t > t_{cr}$), and the radius at which the ETV4 transition occurs varies dynamically over time. To ensure a comprehensive understanding of this concept, we have incorporated additional clarifications into the text of our revised manuscript. Please see Page 21 Line 23.

- There seems to be in the bottom-most experimental panel of Fig. 7f, a central patch of cells which are ETV4-positive. Can the authors comment on the origin of this patch?

A: We observed that certain large hESC colonies exhibited the presence of differentiated cells within their centers, concomitant with the reactivation of ETV4 expression. To provide a more accurate representation, we have incorporated an image of a colony devoid of differentiated cells into our

updated figure. Please see Fig. 7f in the revised manuscript.

- I have difficulties seeing that integrin is « cytoplasmic » in the center of the colony and less so at its periphery, in Fig. 6b. I rather see a strong signal close to the nuclei in the center, and a smaller signal at the periphery. Can the authors clarify this?

A: As the reviewer pointed out, cells in the center of large hESC colonies showed a perinuclear localization of integrin with loss of paxillin colocalization. We have made revisions to the images and the text in our manuscript to enhance clarity (Fig. 6b). Please see Page 18 Line 17.

Minor comments:

- The figure panels are overall very small and difficult to read, some panels (e.g Fig. 6a) have impossibly small numbers. I would suggest that the authors improve figure readability.

A: Revised as suggested.

- While it is clear in the context of the paper, the authors sometimes refer to « cell size » to refer to « cell apical area ». Because cell size could also refer to cell volume, I would suggest to clarify this possible ambiguity.

A: As recommended, we have used the term "cell surface area" in place of "cell size" throughout our revised manuscript.

Decision Letter, first revision:

Our ref: NCB-A50665A

17th January 2024

Dear Dr. Jang,

I apologize for the delay, and thank you for sending us your further proposed revision plans in response to the current reviewer comments.

Thank you for submitting your revised manuscript "ETV4 is a mechanical transducer linking cell crowding dynamics to lineage specification" (NCB-A50665A). It has now been seen by the original referees and their comments are below. The reviewers find that the paper has improved in revision, and therefore we'll be happy in principle to publish it in Nature Cell Biology, pending minor revisions to satisfy the referees' final requests and to comply with our editorial and formatting guidelines.

In particular, we'd ask for the following textual revisions:

- A) Clarify and discuss in the text potential hypothesis for the interaction between cell spreading, geometry, cell adhesion, and the cytoskeleton in your system (Reviewers #1 and #3)
- B) Further discuss apical area and potential consequences of cytoskeletal regulation (Reviewers #1 and #3)

The current version of your manuscript is in a PDF format. Please email us a copy of the file in an editable format (Microsoft Word or LaTeX)-- we can not proceed with PDFs at this stage.

Thank you again for your interest in Nature Cell Biology Please do not hesitate to contact me if you have any questions.

Sincerely,
Daryl

Daryl Jason Verzosa David, PhD

Senior Editor, Nature Cell Biology
Nature Portfolio
Advisory Editor, npj Biological Physics and Mechanics

Heidelberger Platz 3, 14197 Berlin, Germany
Email: daryl.david@nature.com
ORCID: <https://orcid.org/0000-0002-9253-4805>

Reviewer #1 (Remarks to the Author):

The authors have largely addressed my concerns and the manuscript is in my view almost ready for publication. As a remaining minor issue, I think my previous comment on cell shape was not fully understood, perhaps because I didn't explain it fully. In their mathematical model, the authors assume that the effects of spreading are mediated by integrin availability. I argued that the mechanical effects of cell spreading are known to be determined by the spreading itself, and not necessarily by integrin availability. I cited seminal work by the Ingber lab (DOI: 10.1126/science.276.5317.1425) where they show that through micropatterning approaches.

To address my comment, the authors have measured the cell shape index, which quantifies the "roundness" of cell spreading contact area, and found no differences. With this, they dismiss a role of cell shape. However, this misses the point: the role of cell shape in this context is precisely through cell spreading, not through the cell shape index. They also mention the role of integrin signalling, and of actomyosin activity and contractility, which they demonstrate. But all of these things are required for cell spreading, so they don't discard my point.

The only way to discriminate between a regulation via integrin availability, versus cell shape (spreading) itself, would be to engage in micropatterning experiments at the subcellular scale, and this is clearly beyond the scope of this paper and not necessary. However, all that I am saying is that a plausible hypothesis is that reduced cell spreading, by favouring a more rounded shape, would lead to the strengthening of the actin cortex, increased membrane tension, and impaired endocytosis. This is actually the mechanism demonstrated in this related paper: <https://doi.org/10.1016/j.stem.2020.10.018>. This still involves integrin signalling, still involves the cytoskeleton, still involves cell spreading, but simply acts through a mechanism that is not directly the amount of available integrins as hypothesized in the theoretical model.

To address my previous comments, the authors now state:

"Although our data demonstrate the critical role of actomyosin activity in receptor endocytosis, it is important to acknowledge that other cellular changes induced by cell crowding, such as the reduction in apical surface area for ligand interaction, cell geometry, and membrane tension, may also play a role in the regulation of receptor endocytosis."

In my view, membrane tension would not be "another cellular change" as stated by the authors, but a direct consequence of actomyosin activity, that perfectly fits with their data. I think mentioning this possibility and the relevant literature in the discussion would enrich the paper.

Reviewer #2 (Remarks to the Author):

The authors have provided a very thoughtful and thorough set of responses to all issues raised in my review. The additional data, and the revisions to the text, improve the precision and clarity of the study, and strengthen its conclusions. Although precisely how the role of cell crowding and mechanical dynamics integrates with canonical determinants of cell fate will be the subject of productive future study, this work has opened up new avenues of investigation.

Reviewer #3 (Remarks to the Author):

In their revision, Yang&al have answered many of my comments. I would suggest to further consider the points below.

- 1) I made a comment about mechanical stress and cell area which I think the authors misunderstood. The authors replied that cell "geometry" as measured by cell shape index is not linked with ETV4 expression. My point is rather that the authors, if I am correct, are never actually measuring mechanical stress, rather all their reasoning are based on geometrical measurements of cell apical area. I would suggest to make this point clear.
- 2) I also would suggest that the authors make a distinction between "cell apical area" and "cell surface area"; the latter for me would refer to the surface area of the entire cell.
- 3) in the table of parameters I think that r_0^* is given in mm and not in μm .

Decision Letter, final checks:

Our ref: NCB-A50665A

22nd February 2024

Dear Dr. Jang,

Thank you for your patience as we've prepared the guidelines for final submission of your Nature Cell Biology manuscript, "ETV4 is a mechanical transducer linking cell crowding dynamics to lineage specification" (NCB-A50665A). Please carefully follow the step-by-step instructions provided in the attached file, and add a response in each row of the table to indicate the changes that you have made. Ensuring that each point is addressed will help to ensure that your revised manuscript can be swiftly handed over to our production team.

In recognition of the time and expertise our reviewers provide to Nature Cell Biology's editorial process, we would like to formally acknowledge their contribution to the external peer review of your manuscript entitled "ETV4 is a mechanical transducer linking cell crowding dynamics to lineage specification". For those reviewers who give their assent, we will be publishing their names alongside the published article.

Nature Cell Biology offers a Transparent Peer Review option for new original research manuscripts submitted after December 1st, 2019. As part of this initiative, we encourage our authors to support

increased transparency into the peer review process by agreeing to have the reviewer comments, author rebuttal letters, and editorial decision letters published as a Supplementary item. When you submit your final files please clearly state in your cover letter whether or not you would like to participate in this initiative. Please note that failure to state your preference will result in delays in accepting your manuscript for publication.

Cover suggestions

COVER ARTWORK: We welcome submissions of artwork for consideration for our cover. For more information, please see our guide for cover artwork.

Nature Cell Biology has now transitioned to a unified Rights Collection system which will allow our Author Services team to quickly and easily collect the rights and permissions required to publish your work. Approximately 10 days after your paper is formally accepted, you will receive an email in providing you with a link to complete the grant of rights. If your paper is eligible for Open Access, our Author Services team will also be in touch regarding any additional information that may be required to arrange payment for your article.

Please note that *Nature Cell Biology* is a Transformative Journal (TJ). Authors may publish their research with us through the traditional subscription access route or make their paper immediately open access through payment of an article-processing charge (APC). Authors will not be required to make a final decision about access to their article until it has been accepted. Find out more about Transformative Journals

Please use the following link for uploading these materials:
[Redacted]

Best regards,

Kendra Donahue
Staff
Nature Cell Biology

On behalf of

Daryl Jason Verzosa David, PhD

Senior Editor, Nature Cell Biology
Nature Portfolio
Advisory Editor, npj Biological Physics and Mechanics

Heidelberger Platz 3, 14197 Berlin, Germany
Email: daryl.david@nature.com
ORCID: <https://orcid.org/0000-0002-9253-4805>

Reviewer #1:

Remarks to the Author:

The authors have largely addressed my concerns and the manuscript is in my view almost ready for publication. As a remaining minor issue, I think my previous comment on cell shape was not fully understood, perhaps because I didn't explain it fully. In their mathematical model, the authors assume that the effects of spreading are mediated by integrin availability. I argued that the mechanical effects of cell spreading are known to be determined by the spreading itself, and not necessarily by integrin availability. I cited seminal work by the Ingber lab (DOI: 10.1126/science.276.5317.1425) where they show that through micropatterning approaches.

To address my comment, the authors have measured the cell shape index, which quantifies the "roundness" of cell spreading contact area, and found no differences. With this, they dismiss a role of cell shape. However, this misses the point: the role of cell shape in this context is precisely through cell spreading, not through the cell shape index. They also mention the role of integrin signalling, and of actomyosin activity and contractility, which they demonstrate. But all of these things are required for cell spreading, so they don't discard my point.

The only way to discriminate between a regulation via integrin availability, versus cell shape (spreading) itself, would be to engage in micropatterning experiments at the subcellular scale, and this is clearly beyond the scope of this paper and not necessary. However, all that I am saying is that a plausible hypothesis is that reduced cell spreading, by favouring a more rounded shape, would lead to the strengthening of the actin cortex, increased membrane tension, and impaired endocytosis. This is actually the mechanism demonstrated in this related paper:

<https://doi.org/10.1016/j.stem.2020.10.018>. This still involves integrin signalling, still involves the cytoskeleton, still involves cell spreading, but simply acts through a mechanism that is not directly the amount of available integrins as hypothesized in the theoretical model.

To address my previous comments, the authors now state:

"Although our data demonstrate the critical role of actomyosin activity in receptor endocytosis, it is important to acknowledge that other cellular changes induced by cell crowding, such as the reduction in apical surface area for ligand interaction, cell geometry, and membrane tension, may also play a role in the regulation of receptor endocytosis."

In my view, membrane tension would not be "another cellular change" as stated by the authors, but a direct consequence of actomyosin activity, that perfectly fits with their data. I think mentioning this possibility and the relevant literature in the discussion would enrich the paper.

Reviewer #2:

Remarks to the Author:

The authors have provided a very thoughtful and thorough set of responses to all issues raised in my review. The additional data, and the revisions to the text, improve the precision and clarity of the study, and strengthen its conclusions. Although precisely how the role of cell crowding and mechanical dynamics integrates with canonical determinants of cell fate will be the subject of productive future study, this work has opened up new avenues of investigation.

Reviewer #3:

Remarks to the Author:

In their revision, Yang&al have answered many of my comments. I would suggest to further consider the points below.

1) I made a comment about mechanical stress and cell area which I think the authors misunderstood. The authors replied that cell "geometry" as measured by cell shape index is not linked with ETV4 expression. My point is rather that the authors, if I am correct, are never actually measuring mechanical stress, rather all their reasoning are based on geometrical measurements of cell apical area. I would suggest to make this point clear.

2) I also would suggest that the authors make a distinction between "cell apical area" and "cell surface area"; the latter for me would refer to the surface area of the entire cell.

3) in the table of parameters I think that r_0^* is given in mm and not in um.

Author Rebuttal, first revision:

Reviewers' Comments:

Reviewer #1:

Remarks to the Author:

The authors have largely addressed my concerns and the manuscript is in my view almost ready for publication. As a remaining minor issue, I think my previous comment on cell shape was not fully

understood, perhaps because I didn't explain it fully. In their mathematical model, the authors assume that the effects of spreading are mediated by integrin availability. I argued that the mechanical effects of cell spreading are known to be determined by the spreading itself, and not necessarily by integrin availability. I cited seminal work by the Ingber lab (DOI: 10.1126/science.276.5317.1425) where they show that through micropatterning approaches.

To address my comment, the authors have measured the cell shape index, which quantifies the “roundness” of cell spreading contact area, and found no differences. With this, they dismiss a role of cell shape. However, this misses the point: the role of cell shape in this context is precisely through cell spreading, not through the cell shape index. They also mention the role of integrin signalling, and of actomyosin activity and contractility, which they demonstrate. But all of these things are required for cell spreading, so they don't discard my point.

The only way to discriminate between a regulation via integrin availability, versus cell shape (spreading) itself, would be to engage in micropatterning experiments at the subcellular scale, and this is clearly beyond the scope of this paper and not necessary. However, all that I am saying is that a plausible hypothesis is that reduced cell spreading, by favouring a more rounded shape, would lead to the strengthening of the actin cortex, increased membrane tension, and impaired endocytosis. This is actually the mechanism demonstrated in this related paper:

<https://doi.org/10.1016/j.stem.2020.10.018>. This still involves integrin signalling, still involves the cytoskeleton, still involves cell spreading, but simply acts through a mechanism that is not directly the amount of available integrins as hypothesized in the theoretical model.

To address my previous comments, the authors now state:

“Although our data demonstrate the critical role of actomyosin activity in receptor endocytosis, it is important to acknowledge that other cellular changes induced by cell crowding, such as the reduction in apical surface area for ligand interaction, cell geometry, and membrane tension, may also play a role in the regulation of receptor endocytosis.”

In my view, membrane tension would not be “another cellular change” as stated by the authors, but a direct consequence of actomyosin activity, that perfectly fits with their data.

- We acknowledge that “membrane tension” directly results from actomyosin activity and should not be categorized as “another cellular change”. We revised this sentence in the revised manuscript. Please see Page 15 Line 20.

I think mentioning this possibility and the relevant literature in the discussion would enrich the paper.

- Our current study convincingly demonstrated that cell spreading impacts receptor endocytosis and downstream ETV4 expression through integrin signaling. We observed a sharp reduction in integrin signaling, indicated by decreased phospho-FAK levels, upon reduced cell spreading (resulting from

increased cell density) (Fig. 6c). Furthermore, by blocking integrin signaling, we successfully inhibited receptor endocytosis and ETV4 expression in cells exhibiting low cell density and high spreading (Extended Data Fig. 10d,f). These findings underscore the pivotal role of integrin signaling in mediating the relationship between cell spreading and the subsequent downstream pathway.

To establish a connection between cell spreading and integrin activity in our mathematical model, we employed integrin ligand availability influenced by cell-ECM adhesion area due to its well-established significance. Based on this straightforward mechanism, our model recapitulated the experimental observation of an ultrasensitive transition in integrin activity and ETV4 expression in response to gradual changes in cell density (Fig. 7d,g). This sharp transition occurred due to the cooperative kinetics of integrin-ligand binding. Consequently, we propose integrin ligand availability as a crucial mechanism governing the relationship between cell crowding dynamics and ETV4 transition.

However, as highlighted by the reviewer, we acknowledge the possibility that cell shape itself may contribute to regulating integrin signaling or the downstream pathway independently of integrin ligand availability. We have made a note of this possibility in the revised manuscript. Please see Page 17 Line 6.

Reviewer #2:

Remarks to the Author:

The authors have provided a very thoughtful and thorough set of responses to all issues raised in my review. The additional data, and the revisions to the text, improve the precision and clarity of the study, and strengthen its conclusions. Although precisely how the role of cell crowding and mechanical dynamics integrates with canonical determinants of cell fate will be the subject of productive future study, this work has opened up new avenues of investigation.

- We express our gratitude to the reviewer for recognizing our study as a gateway to exploring crucial biological questions.

Reviewer #3:

Remarks to the Author:

In their revision, Yang&al have answered many of my comments. I would suggest to further consider the points below.

1) I made a comment about mechanical stress and cell area which I think the authors misunderstood. The authors replied that cell "geometry" as measured by cell shape index is not linked with ETV4 expression. My point is rather that the authors, if I am correct, are never actually measuring mechanical stress, rather all their reasoning are based on geometrical measurements of cell apical area. I would suggest to make this point clear.

- The cellular response to mechanical stress is intricate, wherein stress fibers—contractile actomyosin bundles—function as reliable indicators of cellular mechanical stress levels. In our study, we consistently observed the dynamic regulation of pMLC+ stress fiber assembly within hESCs in response to mechanical cues such as cell density and substrate stiffness (Fig. 6e and Extended Data Fig. 1e,k). Additionally, we identified integrin signaling as a molecular mechanism that links cell density to the assembly of stress fibers (Extended Data Fig. 10g,h).

Actomyosin, beyond its role as a marker of mechanical stress, operates as vital cellular machinery involved in both sensing and mediating cellular responses to mechanical stimuli. Our study demonstrated the impact of actomyosin on receptor endocytosis and downstream pathways by directly manipulating it using myosin inhibitors (Fig. 6f-h, Extended Data Fig. 10d,k,l). Furthermore, we showed that increased actomyosin activity by constitutively-active RhoA was sufficient to re-activate receptor endocytosis in a high-density condition (Fig. 6i). This observation strongly implies the critical involvement of actomyosin in regulating density-dependent receptor endocytosis. As suggested by the reviewer, we clarified this aspect in the revised manuscript. Please see Page 4 Line 3 and Page 15 Line 17.

2) I also would suggest that the authors make a distinction between "cell apical area" and "cell surface area"; the latter for me would refer to the surface area of the entire cell.

- We acknowledge that the term "cell surface area" might lead to confusion. We provided clarification when it is initially introduced in the revised manuscript. Please see Page 3 Line 23.

3) in the table of parameters I think that r_0^* is given in mm and not in μm .

- The typo is corrected in the revised manuscript. Please see Supplementary Table 8.

Final Decision Letter:

Dear Dr Jang,

I am pleased to inform you that your manuscript, "ETV4 is a mechanical transducer linking cell crowding dynamics to lineage specification", has now been accepted for publication in Nature Cell Biology.

Please note that *Nature Cell Biology* is a Transformative Journal (TJ). Authors may publish their research with us through the traditional subscription access route or make their paper immediately open access through payment of an article-processing charge (APC). Authors will not be required to make a final decision about access to their article until it has been accepted. Find out more about Transformative Journals

To assist our authors in disseminating their research to the broader community, our SharedIt initiative provides you with a unique shareable link that will allow anyone (with or without a subscription) to read the published article. Recipients of the link with a subscription will also be able to download and

print the PDF.

If you have not already done so, we strongly recommend that you upload the step-by-step protocols used in this manuscript to the Protocol Exchange (www.nature.com/protocolexchange), an open online resource established by Nature Protocols that allows researchers to share their detailed experimental know-how. All uploaded protocols are made freely available, assigned DOIs for ease of citation and are fully searchable through nature.com. Protocols and Nature Portfolio journal papers in which they are used can be linked to one another, and this link is clearly and prominently visible in the online versions of both papers. Authors who performed the specific experiments can act as primary authors for the Protocol as they will be best placed to share the methodology details, but the Corresponding Author of the present research paper should be included as one of the authors. By uploading your Protocols to Protocol Exchange, you are enabling researchers to more readily reproduce or adapt the methodology you use, as well as increasing the visibility of your protocols and papers. You can also establish a dedicated page to collect your lab Protocols. Further information can be found at www.nature.com/protocolexchange/about

With kind regards,
Daryl

Daryl Jason Verzosa David, PhD

Senior Editor, Nature Cell Biology
Nature Portfolio
Advisory Editor, npj Biological Physics and Mechanics

Heidelberger Platz 3, 14197 Berlin, Germany
Email: daryl.david@nature.com
ORCID: <https://orcid.org/0000-0002-9253-4805>